# UNDERSTANDING THE GENERALIZATION OF ADAM IN LEARNING NEURAL NETWORKS WITH PROPER REGULARIZATION

**Difan Zou[1], Yuan Cao[2], Yuanzhi Li[3], Quanquan Gu[4]**

[1] Department of Computer Science & Institute of Data Science, The University of Hong Kong
`dzou@cs.hku.hk`
[2] Department of Statistics & Actuarial Science, The University of Hong Kong
`yuancao@hku.hk`
[3] Machine Learning Department, Carnegie Mellon University
`yuanzhil@andrew.cmu.edu`
[4] Department of Computer Science, University of California, Los Angeles
`qgu@cs.ucla.edu`

## ABSTRACT

Adaptive gradient methods such as Adam have gained increasing popularity in deep learning optimization. However, it has been observed in many deep learning applications such as image classification, Adam can converge to a different solution with a worse test error compared to (stochastic) gradient descent, even with a fine-tuned regularization. In this paper, we provide a theoretical explanation for this phenomenon: we show that in the nonconvex setting of learning over-parameterized two-layer convolutional neural networks starting from the same random initialization, for a class of data distributions (inspired from image data), Adam and gradient descent (GD) can converge to different global solutions of the training objective with provably different generalization errors, even with weight decay regularization. In contrast, we show that if the training objective is convex, and the weight decay regularization is employed, any optimization algorithms including Adam and GD will converge to the same solution if the training is successful. This suggests that the generalization gap between Adam and SGD in the presence of weight decay regularization is closely tied to the nonconvex landscape of deep learning optimization, which cannot be covered by the recent neural tangent kernel (NTK) based analysis.

## 1 INTRODUCTION

Adaptive gradient methods (Duchi et al., 2011; Hinton et al., 2012; Kingma & Ba, 2015; Reddi et al., 2018) such as Adam are very popular optimizers for training deep neural networks. By adjusting the learning rate coordinate-wisely based on historical gradient information, they are known to be able to automatically choose appropriate learning rates to achieve fast convergence in training. Because of this advantage, Adam and its variants are widely used in deep learning.

Despite their fast convergence, adaptive gradient methods have been observed to achieve worse generalization performance compared with gradient descent and stochastic gradient descent (SGD) (Wilson et al., 2017; Luo et al., 2019; Chen et al., 2020; Zhou et al., 2020) in many deep learning tasks such as image classification (we have done some simple deep learning experiments to justify this, the results are reported in Table 1). Even with explicit weight decay regularization, achieving good test error with adaptive gradient methods seems to be challenging. Moreover, we have also visualized the first layer of AlexNet trained by Adam and SGD in Figure 1, where

| Models | AlexNet | VGG-16 | ResNet-18 |
|--------|---------|--------|-----------|
| SGD | 75.22 | 93.25 | 94.62 |
| Adam | 73.08 | 92.19 | 92.93 |

Table 1: Test accuracy (%) comparison between Adam and SGD on the CIFAR-10 dataset.

we can also observe a clear difference between Adam and SGD: the model learned by Adam is more "noisy" than that learned by SGD.

Several recent works provided theoretical explanations of this generalization gap between Adam and GD by showing that Adam and GD have different implicit bias. Wilson et al. (2017); Agarwal et al. (2019) considered a setting of linear regression, and showed that Adam can fail when learning an overparameterized linear model on certain specifically designed data, while SGD can learn the linear model to achieve zero test error. This example in linear regression offers valuable insights into the difference between SGD and Adam. However, there is a gap between their theoretical results and the practical observations, since they consider a convex optimization setting, and the difference between Adam and SGD will no longer be observed when adding weight decay regularization. In fact, as we will show in this paper (Theorem 4.2), regularization can successfully correct the different implicit bias and push different algorithms to find the same solution, since the regularized training loss function of a convex model becomes strongly convex, which exhibits one unique global optimum. For this reason, we argue that the example in the convex setting cannot fully capture the differences between GD and Adam for training neural networks. More recently, Zhou et al. (2020) studied the expected escaping time of Adam and SGD from a local basin, and utilized this to explain the difference between SGD and Adam. However, their results do not take NN architecture into consideration, and do not provide an analysis of test errors either.

In this paper, we aim at answering the following question

*Why is there a generalization gap between Adam and gradient descent in learning neural networks, even with weight decay regularization?*

Specifically, we study Adam and GD for training neural networks with weight decay regularization on an image-like data model, and demonstrate the different behaviors of Adam and GD based on the notion of feature learning/noise memorization decomposition. Inspired by the experimental observation in Figure 1 where Adam tends to overfit the noise component of the data, we consider a model where the data are generated as a combination of feature and noise patches, and analyze the convergence and generalization of Adam and GD for training a two-layer convolutional neural network (CNN). The contributions of this paper are summarized as follows.

- We establish global convergence guarantees for Adam and GD with weight decay regularization. We show that, starting at the same random initialization, Adam and GD can both train a two-layer convolutional neural network to achieve zero training error after polynomially many iterations, despite the nonconvex optimization landscape.

- We further show that GD and Adam in fact converge to different global solutions with different generalization performance: when performed on the considered image-like data model, GD can achieve nearly zero test error, while the generalization performance of the model found by Adam is no better than a random guess. In particular, we show that the reason for this gap is due to the different training behaviors of Adam and GD: Adam is more likely to fit dense noises and output a model that is largely contributed by the noise patches; GD prefers to fit training data using their feature patch and finds a solution that is mainly composed by the true features.

- We also show that for convex settings with weight decay regularization, both Adam and gradient descent converge to the same solution and therefore have no test error difference. This suggests that the difference between Adam and GD cannot be fully explained by linear models or neural networks trained in the "almost convex" neural tangent kernel (NTK) regime (Jacot et al., 2018; Allen-Zhu et al., 2019b; Du et al., 2019a; Zou et al., 2019). It also demonstrates that the inferior generalization performance of Adam is closely tied to the nonconvex landscape of deep learning optimization, and cannot be solved by adding regularization.

## 2 RELATED WORK

In this section, we discuss the works that are closely related to our paper.

**Generalization gap between Adam and SGD.** The worse generalization of Adam compared with SGD has also been observed by some recent works and has motivated new variants of neural network training algorithms. Keskar & Socher (2017) proposed to switch between Adam and SGD to achieve better generalization. Merity et al. (2018) proposed a variant of the averaged stochastic gradient method to achieve good generalization performance for LSTM language models. Luo et al. (2019) proposed to use dynamic bounds on learning rates to achieve a smooth transition from adaptive methods to SGD to improve generalization. Our theoretical results for GD and Adam can also

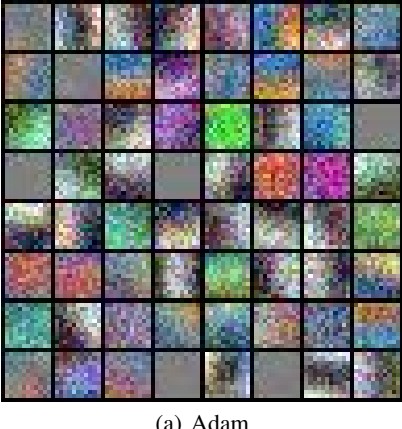 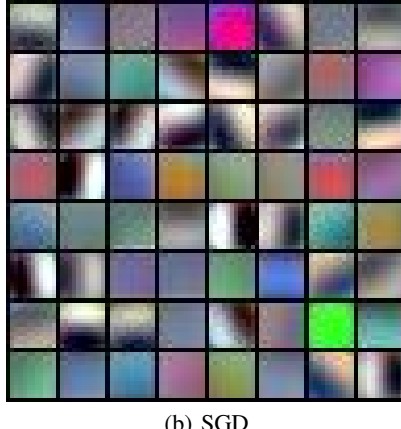

(a) Adam            (b) SGD

Figure 1: Visualization of the first layer of AlexNet trained by Adam and SGD on the CIFAR-10 dataset. Both algorithms are run for 100 epochs with weight decay regularization and standard data augmentations. Clearly, the model learned by Adam is more "noisy" than that learned by SGD, implying that Adam is more likely to overfit the noise in the training data.

provide theoretical insights into the effectiveness of these empirical studies.

**Optimization and generalization in deep learning.** Our work is also closely related to the recent line of work studying the optimization and generalization guarantees of neural networks in the neural tangent kernel (NTK) regime (Jacot et al., 2018) or lazy training regime (Chizat et al., 2019). In particular, recent works (Du et al., 2019b;a; Allen-Zhu et al., 2019b; Zou et al., 2019) showed that the optimization only happens within a small neighborhood region around the random initialization and proved the global convergence of GD and SGD when the neural network is sufficiently wide. Moreover, the generalization ability of GD/SGD has been further studied in the same setting (Allen-Zhu et al., 2019a; Arora et al., 2019a;b; Ji & Telgarsky, 2020; Chen et al., 2021), which suggests that wide neural network trained by GD/SGD can learn a low-dimensional function class. Moreover, Allen-Zhu & Li (2019); Bai & Lee (2019) initiated the study of learning neural networks beyond the NTK regime as it differs from the practical DNN training. Our analysis in this paper is also beyond NTK, and gives a detailed comparison between GD and Adam.

**Feature learning by neural networks.** This paper is also closely related to several recent works that studied how neural networks can learn features. Allen-Zhu & Li (2020a) showed that adversarial training purifies the learned features by removing certain "dense mixtures" in the hidden layer weights of the network. Allen-Zhu & Li (2020b) studied how ensemble and knowledge distillation work in deep learning when the data have "multi-view" features. Frei et al. (2022b) studied the feature learning for two-layer networks, and demonstrated its superior performance than linear models. Shen et al. (2022) explored the benefit of data augmentation by showing its ability to achieve more effective feature learning. This paper studies a different aspect of feature learning by Adam and GD, and shows that GD can learn the features while Adam may fail even with proper regularization.

## 3 PROBLEM SETUP AND PRELIMINARIES

We consider learning a CNN with Adam and GD based on $n$ independent training examples $\{(\mathbf{x}_i, y_i)\}_{i=1}^n$ generated from a data model $\mathcal{D}$. In the following. we first introduce our data model $\mathcal{D}$, and then explain our neural network model and the details of the training algorithms.

**Data model.** We consider a data model where the data inputs consist of feature and noise patches. Such a data model is motivated by image classification problems where the label of an image usually only depends on part of an image, and the other parts of the image showing random objects, or features that belong to other classes, can be considered as noises. When using CNN to fit the data, the convolution operation is applied to each patch of the data input separately. We claim that our data model is more practical than those considered in Wilson et al. (2017); Reddi et al. (2018), which are handcrafted for showing the failure of Adam in term of either convergence or generalization (detailed illustrations of the data models in these works are deferred to the appendix). For simplicity, we only consider the case where the data consists of one feature patch and one noise patch. However, our result can be easily extended to cover the setting where there are multiple feature/noise patches. The detailed definition of our data model is given in Definition 3.1 as follows.

**Definition 3.1.** Each data point $(\mathbf{x}, y)$ with $\mathbf{x} \in \mathbb{R}^{2d}$ and $y \in \{-1, 1\}$ is generated as follows: $\mathbf{x} = [\mathbf{x}_1^\top, \mathbf{x}_2^\top]^\top$, where one of $\mathbf{x}_1$ and $\mathbf{x}_2$ denotes the feature patch that consists of a feature vector

$y \cdot \mathbf{v}$, which is assumed to be 1-sparse, and the other one denotes the noise patch and consists of a noise vector $\boldsymbol{\xi}$. Without loss of generality, we assume $\mathbf{v} = [1, 0, \ldots, 0]^\top$. The noise vector $\boldsymbol{\xi}$ is generated according to the following process:

- Randomly select $s$ coordinates from $[d]\backslash\{1\}$ uniformly, denoted as a vector $\mathbf{s} \in \{0, 1\}^d$.
- Generate $\boldsymbol{\xi}$ from distribution $\mathcal{N}(\mathbf{0}, \sigma_p^2 \mathbf{I})$, and then mask off the first coordinate and other $d - s - 1$ coordinates, i.e., $\boldsymbol{\xi} = \boldsymbol{\xi} \odot \mathbf{s}$.
- Add feature noise to $\boldsymbol{\xi}$, i.e., $\boldsymbol{\xi} = \boldsymbol{\xi} - \alpha y \mathbf{v}$, where $0 < \alpha < 1$ is the strength of the feature noise.

In particular, throughout this paper we set $d = \Omega(n^4)$, $s = \Theta\left(\frac{d^{1/2}}{n^2}\right)$, $\sigma_p^2 = \Theta\left(\frac{1}{s \cdot \text{polylog}(n)}\right)$ and $\alpha = \Theta\left(\sigma_p \cdot \text{polylog}(n)\right)$.

The most natural way to think of our data model is to treat $\mathbf{x}$ as the output of some intermediate layer of a CNN. In literature, Papyan et al. (2017) pointed out that the outputs of an intermediate layer of a CNN are usually sparse. Yang (2019) also discussed the setting where the hidden nodes in such an intermediate layer are sampled independently. This motivates us to study sparse features and entry-wisely independent noises in our model. In this paper, we focus on the case where the feature vector $\mathbf{v}$ is 1-sparse and the noise vector is $s$-sparse for simplicity. However, these sparsity assumptions can be generalized to the settings where the feature and the noises are denser, as long as the sparsity gap between feature and noises exists.

Note that in Definition 3.1, each data input consists of two patches: a feature patch $y\mathbf{v}$ that is positively correlated with the label, and a noise patch $\boldsymbol{\xi}$ which contains the "feature noise" $-\alpha y \mathbf{v}$ as well as random Gaussian noises. Importantly, the feature noise $-\alpha y \mathbf{v}$ in the noise patch plays a pivotal role in both the training and test processes, which connects the noise overfitting in the training process and the inferior generalization ability in the test process.

Moreover, we would like to clarify that the data distribution considered in our paper is an **extreme case** where we assume there is only one feature vector and all data has a feature noise, since we believe this is the simplest model that captures the fundamental difference between Adam and SGD. With this data model, we aim to show why Adam and SGD perform differently. Our theoretical results and analysis techniques can also be extended to more practical settings where there are multiple feature vectors and multiple patches, each data can either contain a single feature or multiple features, together with pure random noise or feature noise.

**Two-layer CNN model.** We consider a two-layer CNN model $F$ using truncated polynomial activation function $\sigma(z) = (\max\{0, z\})^q$ and fix the weights of second layer to be all 1's, where $q \geq 3$. Given the data $(\mathbf{x}, y)$, the $j$-th output of the CNN can be formulated as

$$F_j(\mathbf{W}, \mathbf{x}) = \sum_{r=1}^m \left[\sigma(\langle \mathbf{w}_{j,r}, \mathbf{x}_1 \rangle) + \sigma(\langle \mathbf{w}_{j,r}, \mathbf{x}_2 \rangle)\right] = \sum_{r=1}^m \left[\sigma(\langle \mathbf{w}_{j,r}, y \cdot \mathbf{v} \rangle) + \sigma(\langle \mathbf{w}_{j,r}, \boldsymbol{\xi} \rangle)\right], \quad (3.1)$$

where $m$ is the width of the network, $\mathbf{w}_{j,r} \in \mathbb{R}^d$ denotes the $r$-th CNN filter, and $\mathbf{W}$ is the collection of model weights. For the ease of analysis, we set the output layer as all 1's. Our analyses and results can still be applied if we use random second layer weights.

Besides, the motivation of using polynomial ReLU activation function is to guarantee that the loss function is (locally) smooth and the amplification ability of pattern learning. It can be replaced by a smoothed ReLU activation function (e.g., the activation function used in Allen-Zhu & Li (2020b)). If we assume the input data distribution is Gaussian, we can also deal with ReLU activation function (Li et al., 2020). A set of similar smoothed ReLU-type activation functions have also been widely considered to study the generalization performance of two-layer neural networks from different aspects (Frei et al., 2022a; Cao et al., 2022; Shen et al., 2022; Chen et al., 2022). Moreover, we would like to emphasize that $\mathbf{x}_1$ and $\mathbf{x}_2$ denote two data patches, which are randomly assigned with feature vectors or noise vectors independently for each data point. The leaner has no knowledge about which one is the feature patch (or noise patch).

In this paper we assume the width of the network is polylogarithmic in the training sample size, i.e., $m = \text{polylog}(n)$. We assume $j \in \{-1, 1\}$ in order to make the logit index be consistent with the data label. Moreover, we assume that the each weight is initialized from a random draw of Gaussian random variable $\sim N(0, \sigma_0^2)$ with $\sigma_0 = \Theta\left(d^{-1/4}\right)$.

**Training objective.** Given the training data points $\{(\mathbf{x}_i, y_i)\}_{i=1,\ldots,n}$, we consider to learn the model parameter $\mathbf{W}$ by optimizing the empirical loss function with weight decay regularization

$$L(\mathbf{W}) = \frac{1}{n}\sum_{i=1}^{n} L_i(\mathbf{W}) + \frac{\lambda}{2}\|\mathbf{W}\|_F^2, \tag{3.2}$$

where $L_i(\mathbf{W}) = -\log\frac{e^{F_{y_i}(\mathbf{W}, \mathbf{x}_i)}}{\sum_{j\in\{-1,1\}} e^{F_j(\mathbf{W}, \mathbf{x}_i)}}$ denotes the individual loss for the data point $(\mathbf{x}_i, y_i)$ and $\lambda \geq 0$ is the regularization parameter. In particular, the regularization parameter can be arbitrary as long as it satisfies $\lambda \in (0, \lambda_0)$ with $\lambda_0 = \Theta\big(\frac{1}{d^{(q-1)/4}n\cdot\text{polylog}(n)}\big)$. We claim that the $\lambda_0$ is the largest feasible regularization parameter that the training process will not stuck at the origin point (recall that $L(\mathbf{W})$ admits zero gradient at $\mathbf{W} = \mathbf{0}$.)

**Training algorithms.** In this paper, we consider full-batch gradient descent and Adam[1]. In particular, starting from $\mathbf{W}^{(0)} = \{\mathbf{w}_{j,r}^{(0)}, j = \{\pm 1\}, r \in [m]\}$, the gradient descent update rule is

$$\mathbf{w}_{j,r}^{(t+1)} = \mathbf{w}_{j,r}^{(t)} - \eta \cdot \nabla_{\mathbf{w}_{j,r}} L(\mathbf{W}^{(t)}),$$

where $\eta$ is the learning rate. Meanwhile, Adam store historical gradient information in the momentum $\mathbf{m}^{(t)}$ and a vector $\mathbf{v}^{(t)}$ as follows

$$\mathbf{m}_{j,r}^{(t+1)} = \beta_1 \mathbf{m}_{j,r}^{(t)} + (1 - \beta_1) \cdot \nabla_{\mathbf{w}_{j,r}} L(\mathbf{W}^{(t)}), \tag{3.3}$$

$$\mathbf{v}_{j,r}^{(t+1)} = \beta_2 \mathbf{v}_{j,r}^{(t)} + (1 - \beta_2) \cdot [\nabla_{\mathbf{w}_{j,r}} L(\mathbf{W}^{(t)})]^2, \tag{3.4}$$

and entry-wisely adjusts the learning rate:

$$\mathbf{w}_{j,r}^{(t+1)} = \mathbf{w}_{j,r}^{(t)} - \eta \cdot \mathbf{m}_{j,r}^{(t)}/\sqrt{\mathbf{v}_{j,r}^{(t)}}, \tag{3.5}$$

where $\beta_1, \beta_2$ are the hyperparameters of Adam (a popular choice in practice is $\beta_1 = 0.9$, and $\beta_2 = 0.99$), which are considered as constants in our paper, and in (3.4) and (3.5), the square $(\cdot)^2$, square root $\sqrt{\cdot}$, and division $\cdot/\cdot$ all denote entry-wise calculations. We would like to clarify the original Adam paper (Kingma & Ba, 2015) considers to normalize the gradient $\mathbf{m}_{j,r}^{(t)}$ via $[\mathbf{v}_{j,r}^{(t)}+\epsilon]^{1/2}$, while the small bias term $\epsilon$ is ignored in our paper. In practice, tuning $\epsilon$ can help improve the generalization ability of Adam (Choi et al., 2019), as it allows to make a trade-off between the normalized gradient update and gradient update. We remark that considering tunable $\epsilon$ is beyond the focus of this paper. For the ease of analysis, we do not consider the initialization bias correction in the original Adam paper either and set $\mathbf{m}_{j,r}^{(0)} = \nabla_{\mathbf{w}_{j,r}} L(\mathbf{W}^{(0)})$ and $\mathbf{v}_{j,r}^{(0)} = [\nabla_{\mathbf{w}_{j,r}} L(\mathbf{W}^{(0)})]^2$.

## 4  MAIN RESULTS

In this section we will state the main theorems in this paper. We first provide the learning guarantees of Adam and Gradient descent for training a two-layer CNN model in the following theorem. Recall that in this setting the training objective is nonconvex.

**Theorem 4.1** (Nonconvex setting). Consider a two-layer CNN defined in (3.1) with $d = \Omega(n^4)$ and regularized training objective (3.2) with a regularization parameter $\lambda > 0$, suppose the network width is $m = \text{polylog}(n)$ and the data distribution follows Definition 3.1, then we have the following guarantees on the training and test errors for the models trained by Adam and Gradient descent:

1. Suppose we run **Adam** for $T = \frac{\text{poly}(n)}{\eta}$ iterations with $\eta = \frac{1}{\text{poly}(n)}$, then with probability at least $1 - O(n^{-1})$, we can find a NN model $\mathbf{W}_{\text{Adam}}^*$ such that $\|\nabla L(\mathbf{W}_{\text{Adam}}^*)\|_1 \leq \frac{1}{T\eta}$. Moreover, the model $\mathbf{W}_{\text{Adam}}^*$ also satisfies:

   - Training error is zero: $\frac{1}{n}\sum_{i=1}^{n} \mathbb{1}\big[F_{y_i}(\mathbf{W}_{\text{Adam}}^*, \mathbf{x}_i) \leq F_{-y_i}(\mathbf{W}_{\text{Adam}}^*, \mathbf{x}_i)\big] = 0$.
   - Test error is high: $\mathbb{P}_{(\mathbf{x},y)\sim\mathcal{D}}\big[F_y(\mathbf{W}_{\text{Adam}}^*, \mathbf{x}) \leq F_{-y}(\mathbf{W}_{\text{Adam}}^*, \mathbf{x})\big] \geq \frac{1}{2}$.

2. Suppose we run **gradient descent** for $T = \frac{\text{poly}(n)}{\eta}$ iterations with learning rate $\eta = \frac{1}{\text{poly}(n)}$, then with probability at least $1 - O(n^{-1})$, we can find a NN model $\mathbf{W}_{\text{GD}}^*$ such that $\|\nabla L(\mathbf{W}_{\text{GD}}^*)\|_F^2 \leq \frac{1}{T\eta}$. Moreover, the model $\mathbf{W}_{\text{GD}}^*$ also satisfies:

---

[1]Our theory can still hold for mini-batch stochastic gradient descent, which we will discuss in Appendix.

- Training error is zero: $\frac{1}{n} \sum_{i=1}^{n} \mathbb{1}\left[F_{y_i}(\mathbf{W}_{\mathrm{GD}}^*, \mathbf{x}_i) \leq F_{-y_i}(\mathbf{W}_{\mathrm{GD}}^*, \mathbf{x}_i)\right] = 0$.
- Test error is nearly zero: $\mathbb{P}_{(\mathbf{x}, y) \sim \mathcal{D}}\left[F_y(\mathbf{W}_{\mathrm{GD}}^*, \mathbf{x}) \leq F_{-y}(\mathbf{W}_{\mathrm{GD}}^*, \mathbf{x})\right] = \frac{1}{\mathrm{poly}(n)}$.

From the optimization perspective, Theorem 4.1 shows that both Adam and GD can be guaranteed to find a point with a very small gradient, which can also achieve zero classification error on the training data. Moreover, it can be seen that given the same iteration number $T$ and learning rate $\eta$, Adam can be guaranteed to find a point with up to $1/(T\eta)$ gradient norm in $\ell_1$ metric, while gradient descent can only be guaranteed to find a point with up to $1/\sqrt{T\eta}$ gradient norm in $\ell_2$ metric. More specifically, let $\epsilon$ be a sufficiently small quantity and ignoring other problem parameters, we can set $\eta = O(\epsilon)$ for Adam and $\eta = O(\epsilon^2)$ for GD, then Adam and GD will need $T = O(\epsilon^2)$ and $T = O(\epsilon^4)$ to find a first-order $\epsilon$-stationary point. This suggests that Adam could enjoy a faster convergence rate compared to SGD in the training process, which is consistent with the practice findings. We would also like to point out that there is no contradiction between our result and the recent work (Reddi et al., 2019) showing that Adam can fail to converge, as the counterexample in Reddi et al. (2019) is for the online version of Adam, while we study the full batch Adam.

In terms of the test performance, their generalization abilities are largely different, even with weight decay regularization. In particular, the output of gradient descent can generalize well and achieve nearly zero test error, while the output of Adam gives nearly $1/2$ test error. In fact, this gap is due to two major aspects of the training process: (1) At the early stage of training where weight decay exhibits negligible effect, Adam and GD behave very differently. In particular, Adam prefers the denser and thus tends to fit the noise vectors $\boldsymbol{\xi}$, gradient descent prefers the data patch of larger $\ell_2$ norm and thus will learn the feature patch; (2) At the late stage of training where the weight decay regularization cannot be ignored, both Adam and gradient descent will be enforced to converge to a *local minimum* of the regularized objective, which maintains the pattern learned in the early stage. Consequently, the model learned by Adam will be biased towards the noise patch to fit the feature noise vector $-\alpha y \mathbf{v}$, which is opposite in direction to the true feature vector and therefore leads to a test error no better than a random guess. More details about the training behaviors of Adam and GD are given in Section 5. Experimental justification are provided in Appendix.

Theorem 4.1 shows that when optimizing a nonconvex training objective, Adam and gradient descent will converge to different global solutions with different generalization errors, even with weight decay regularization. In comparison, the following theorem gives the learning guarantees of Adam and gradient descent when optimizing convex and smooth training objectives (e.g., linear model $F(\mathbf{w}, \mathbf{x}) = \mathbf{w}^\top \mathbf{x}$ with logistic loss).

**Theorem 4.2** (Convex setting). For any convex and smooth training objective with positive regularization parameter $\lambda$, suppose we run **Adam** and **gradient descent** for $T = \frac{\mathrm{poly}(n)}{\eta}$ iterations, then with probability at least $1 - n^{-1}$, the obtained parameters $\mathbf{W}_{\mathrm{Adam}}^*$ and $\mathbf{W}_{\mathrm{GD}}^*$ satisfy that $\|\nabla L(\mathbf{W}_{\mathrm{Adam}}^*)\|_1 \leq \frac{1}{T\eta}$ and $\|\nabla L(\mathbf{W}_{\mathrm{Adam}}^*)\|_2^2 \leq \frac{1}{T\eta}$ respectively. Moreover, let $F(\mathbf{W}, \mathbf{x}) \in \mathbb{R}$ be the output of the convex model with parameter $\mathbf{W}$ and input $\mathbf{x}$, it holds that:

- Training errors are the same, $\frac{1}{n} \sum_{i=1}^{n} \mathbb{1}\left[\mathrm{sgn}\left(F(\mathbf{W}_{\mathrm{Adam}}^*, \mathbf{x}_i)\right) \neq \mathrm{sgn}\left(F(\mathbf{W}_{\mathrm{GD}}^*, \mathbf{x}_i)\right)\right] = 0$.
- Test errors are nearly the same: $\mathbb{P}_{(\mathbf{x}, y) \sim \mathcal{D}}\left[\mathrm{sgn}\left(F(\mathbf{W}_{\mathrm{Adam}}^*, \mathbf{x}_i)\right) \neq \mathrm{sgn}\left(F(\mathbf{W}_{\mathrm{GD}}^*, \mathbf{x}_i)\right)\right] \leq \frac{1}{\mathrm{poly}(n)}$.

Theorem 4.2 shows that when optimizing a convex and smooth training objective (e.g., a linear model with logistic loss) with weight decay regularization, both Adam and gradient can converge to almost the same solution and enjoy very similar generalization performance. The proof will be relying on the strong convexity of the training objective and the convergence (to the first-order stationary) guarantee of Adam (Défossez et al., 2020) and GD. Combining this result and Theorem 4.1, it is clear that the inferior generalization performance is closely tied to the nonconvex landscape of deep learning, and cannot be understood by standard weight decay regularization.

## 5 PROOF OUTLINE OF THE MAIN RESULTS

In this section we provide the proof sketch of Theorem 4.1 and explain the different generalization abilities of the models found by gradient descent and Adam.

Before moving to the proof of main results, we first give the following lemma which shows that for data generated from the data distribution $\mathcal{D}$ in Definition 3.1, with high probability all noise vectors $\{\boldsymbol{\xi}_i\}_{i=1,\ldots,n}$ have nearly disjoint supports.

**Lemma 5.1.** Let $\{(\mathbf{x}_i, y_i)\}_{i=1,\dots,n}$ be the training dataset generated by Definition 3.1. Moreover, recall that $\mathbf{x}_i = [y_i \mathbf{v}^\top, \boldsymbol{\xi}_i^\top]^\top$ (or $\mathbf{x}_i = [\boldsymbol{\xi}_i^\top, y_i \mathbf{v}^\top]^\top$), let $\mathcal{B}_i = \mathrm{supp}(\boldsymbol{\xi}_i) \backslash \{1\}$ be the support of $\boldsymbol{\xi}_i$ except the first coordinate. Then with probability at least $1 - n^{-2}$, $\mathcal{B}_i \cap \mathcal{B}_j = \emptyset$ for all $i \neq j$.

This lemma implies that the optimization of each coordinate of the model parameter $\mathbf{W}$, except for the first one, is mostly determined by only one training data. Technically, this lemma can greatly simplify the analysis for Adam so that we can better illustrate its optimization behavior and explain the generalization performance gap between Adam and gradient descent.

**Proof outline.** For both Adam and gradient descent, we will show that the training process can be decomposed into two stages. In the first stage, which we call *pattern learning stage*, the weight decay regularization will be less important and can be ignored, while the algorithms tend to learn the pattern from the training data. In particular, we will show that in the pattern learning stage, the optimization algorithms have different *algorithmic bias*: Adam tends to fit the noise patch while gradient descent will mainly learn the feature patch. In the second stage, which we call it *regularization stage*, the effect of regularization cannot be neglected, which will regularize the algorithm to converge at some local stationary points. However, due to the nonconvex landscape of the training objective, the pattern learned in the first stage will remain unchanged, even when running an infinitely number of iterations.

## 5.1 PROOF SKETCH FOR ADAM

Recall that in each iteration of Adam, the model weight is updated by using a moving-averaged gradient, normalized by a moving average of the historical gradient squares. As pointed out in Balles & Hennig (2018); Bernstein et al. (2018), Adam behaves similarly to sign gradient descent (signGD) when using sufficiently small step size or the moving average parameters $\beta_1, \beta_2$ are nearly zero, which is also justified in our Lemma C.2. Specifically, we show that when considering constant $\beta_1$ and $\beta_2$, the Adam update on the coordinates with large gradient (e.g., $|\nabla L(\mathbf{W}^{(t)})[k]| > \eta$) can be well approximated by the signGD update (i.e., $\mathrm{sign}(\nabla L(\mathbf{W}^{(t)})[k])$). This motivates us to understand the optimization behavior of signGD and then extends it to Adam using their similarities. In particular, signGD updates the model parameter according to the following rule:

$$\mathbf{w}_{j,r}^{(t+1)} = \mathbf{w}_{j,r}^{(t+1)} - \eta \cdot \mathrm{sgn}(\nabla_{\mathbf{w}_{j,r}} L(\mathbf{W}^{(t)})).$$

Recall that each data has a feature patch and a noise patch. By Lemma 5.1 and the data distribution (see Definition 3.1), all noise vectors $\{\boldsymbol{\xi}_i\}_{i=1,\dots,n}$ are supported on disjoint coordinates except the first coordinate. For $\mathbf{x}_i$, let $\mathcal{B}_i$ denote its support excluding the first coordinate. In the subsequent analysis, we will always assume that those $\mathcal{B}_i$'s are disjoint, i.e., $\mathcal{B}_i \cap \mathcal{B}_j = \emptyset$ if $i \neq j$.

Next we will characterize two aspects of the training process: *feature learning* and *noise memorization*. Mathematically, we focus on two quantities: $\langle \mathbf{w}_{j,r}^{(t)}, j \cdot \mathbf{v} \rangle$ and $\langle \mathbf{w}_{y_i,r}^{(t)}, \boldsymbol{\xi}_i \rangle$. In particular, given the training data point $(\mathbf{x}_i, y_i)$ with $\mathbf{x}_i = [y_i \mathbf{v}^\top, \boldsymbol{\xi}_i^\top]^\top$, larger $\langle \mathbf{w}_{y_i,r}^{(t)}, y_i \cdot \mathbf{v} \rangle$ implies better feature learning and larger $\langle \mathbf{w}_{y_i,r}^{(t)}, \boldsymbol{\xi}_i \rangle$ represents better noise memorization. Then regarding the feature vector $\mathbf{v}$ that only has nonzero entry at the first coordinate, we have the following for signGD:

$$\langle \mathbf{w}_{j,r}^{(t+1)}, j\mathbf{v} \rangle = \langle \mathbf{w}_{j,r}^{(t)}, j\mathbf{v} \rangle - \eta \cdot \langle \mathrm{sgn}(\nabla_{\mathbf{w}_{j,r}} L(\mathbf{W}^{(t)})), j\mathbf{v} \rangle \tag{5.1}$$

$$= \langle \mathbf{w}_{j,r}^{(t)}, j\mathbf{v} \rangle + j\eta \cdot \mathrm{sgn}\left( \sum_{i=1}^n y_i \ell_{j,i}^{(t)} \big[ \sigma'(\langle \mathbf{w}_{j,r}^{(t)}, y_i \mathbf{v} \rangle) - \alpha \sigma'(\langle \mathbf{w}_{j,r}^{(t)}, \boldsymbol{\xi}_i \rangle) \big] - n\lambda \mathbf{w}_{j,r}^{(t)}[1] \right),$$

where $\ell_{j,i}^{(t)} := \mathbb{1}_{y_i = j} - \mathrm{logit}_j(F, \mathbf{x}_i)$ and $\mathrm{logit}_j(F, \mathbf{x}_i) = \frac{e^{F_j(\mathbf{W}, \mathbf{x}_i)}}{\sum_{k \in \{-1,1\}} e^{F_k(\mathbf{W}, \mathbf{x}_i)}}$. From (5.1) we can observe three terms in the signed gradient. Specifically, the first term represents the gradient over the feature patch, the second term stems from the feature noise term in the noise patch (see Definition 3.1), and the last term is the gradient of the weight decay regularization. On the other hand, the memorization of the noise vector $\boldsymbol{\xi}_i$ can be described as follows.

$$\langle \mathbf{w}_{y_i,r}^{(t+1)}, \boldsymbol{\xi}_i \rangle - \langle \mathbf{w}_{y_i,r}^{(t)}, \boldsymbol{\xi}_i \rangle = -\eta \cdot \langle \mathrm{sgn}(\nabla_{\mathbf{w}_{y_i,r}} L(\mathbf{W}^{(t)})), \boldsymbol{\xi}_i \rangle \tag{5.2}$$

$$= \eta \sum_{k \in \mathcal{B}_i \cup \{1\}} \mathrm{sgn}\left( \ell_{y_i,i}^{(t)} \sigma'(\langle \mathbf{w}_{y_i,r}^{(t)}, \boldsymbol{\xi}_i \rangle) \boldsymbol{\xi}_i[k] - n\lambda \mathbf{w}_{y_i,r}^{(t)}[k] \right) \cdot \boldsymbol{\xi}_i[k].$$

Throughout the proof, we will show that the training process of Adam can be decomposed into two stages: *pattern learning stage* and *regularization stage*. In the first stage, the algorithm learns the pattern of training data quickly, without being affected by the regularization term. In the second stage, the training data has already been correctly classified since the pattern has been well captured, the regularization will play an important role in the training process and guide the model to converge. **Stage I: Learning the pattern.** At the beginning of training, the neural network output is smaller than some constant for all data, and therefore all training data remain under-fitted and can provide large gradient for model training. We specify this stage of training as Stage I. In this stage, the effect of weight decay regularization can be ignored due to our choice of $\lambda$. We will show that in this stage the inner product $\langle \mathbf{w}_{y_i,r}^{(t)}, \boldsymbol{\xi}_i \rangle$ grows much faster than $\langle \mathbf{w}_{j,r}^{(t)}, j\mathbf{v} \rangle$ since feature learning only makes use of the first coordinate of the gradient, while noise memorization could take advantage of all the coordinates in $\mathcal{B}_i$ (see (5.2), note that $|\mathcal{B}_i| = s \gg 1$).

**Lemma 5.2** (General results in Stage I). Suppose the training data is generated according to Definition 3.1, assume $\lambda = o(\sigma_0^{q-2}\sigma_p/n)$ and $\eta = 1/\text{poly}(d)$, then for any $t \le T_0$ with $T_0 = \widetilde{O}\big(\frac{1}{\eta s \sigma_p}\big)$ and any $i \in [n]$,

$$\langle \mathbf{w}_{j,r}^{(t+1)}, j \cdot \mathbf{v} \rangle \le \langle \mathbf{w}_{j,r}^{(t)}, j \cdot \mathbf{v} \rangle + \Theta(\eta), \quad \langle \mathbf{w}_{y_i,r}^{(t+1)}, \boldsymbol{\xi}_i \rangle = \langle \mathbf{w}_{y_i,r}^{(t)}, \boldsymbol{\xi}_i \rangle + \widetilde{\Theta}(\eta s \sigma_p).$$

Since $\langle \mathbf{w}_{j,r}^{(t)}, \boldsymbol{\xi}_i \rangle$ enjoys a much faster increasing rate than that of $\langle \mathbf{w}_{j,r}^{(t)}, j \cdot \mathbf{v} \rangle$, after a certain number of iterations, the learning of noise patch will dominate the learning of feature patch (i.e., $\alpha\sigma'(\langle \mathbf{w}_{j,r}^{(t)}, \boldsymbol{\xi}_i \rangle) > \sigma'(\langle \mathbf{w}_{j,r}^{(t)}, y_i\mathbf{v} \rangle)$). Thus, by (5.1), the model will tend to fit the feature noise in the noise patch (i.e., $-\alpha y_i \mathbf{v}$), leading to a flipped feature learning phenomenon.

**Lemma 5.3** (Flipping the feature learning). Suppose the training data is generated according to Definition 3.1, $\alpha \ge \widetilde{\Theta}\big((s\sigma_p)^{1-q} \vee \sigma_0^{q-1}\big)$ and $\sigma_0 < \widetilde{O}\big((s\sigma_p)^{-1}\big)$, then for any $t \in [T_r, T_0]$ with $T_r = \widetilde{O}\big(\frac{\sigma_0}{\eta s \sigma_p \alpha^{1/(q-1)}}\big) \le T_0$,

$$\langle \mathbf{w}_{j,r}^{(t+1)}, j \cdot \mathbf{v} \rangle = \langle \mathbf{w}_{j,r}^{(t)}, j \cdot \mathbf{v} \rangle - \Theta(\eta).$$

Moreover, it holds that (1) $\mathbf{w}_{j,r}^{(T_0)}[1] = -\text{sgn}(j) \cdot \widetilde{\Omega}\big(\frac{1}{s\sigma_p}\big)$; (2) $\mathbf{w}_{j,r}^{(T_0)}[k] = \text{sgn}(\boldsymbol{\xi}_i[k]) \cdot \widetilde{\Omega}\big(\frac{1}{s\sigma_p}\big)$ or $\mathbf{w}_{j,r}^{(T_0)}[k] = \pm\widetilde{O}(\eta)$ for $k \in \mathcal{B}_i$ with $y_i = j$; and (3) $\mathbf{w}_{j,r}^{(T_0)}[k] = \pm\widetilde{O}(\eta)$ otherwise.

From Lemma 5.3 it can be observed that at the iteration $T_0$, the sign of the first coordinate of $\mathbf{w}_{j,r}^{(T_0)}$ is different from that of the true feature, i.e., $j \cdot \mathbf{v}$. This implies that at the end of the first training stage, the model is biased towards the noise patch to fit the feature noise.

**Stage II: Regularizing the model.** In this stage, as the neural network output becomes larger, part of training data starts to be well fitted and gives smaller gradient. As a consequence, the feature learning and noise memorization processes will be slowed down and the weight decay regularization term cannot be ignored. However, although weight decay regularization can prevent the model weight from being too large, it will maintain the pattern learned in Stage I and cannot push the model back to "forget" the noise and learn the feature and stops at some local stationary points. We summarize these results in the following lemma.

**Lemma 5.4** (Maintain the pattern). If $\alpha = O\big(s\sigma_p^2/n\big)$ and $\eta = o(\lambda)$, then let $r^* = \arg\max_{r\in[m]}\langle \mathbf{w}_{y_i,r}^{(t)}, \boldsymbol{\xi}_i \rangle$, for any $t \ge T_0$, $i \in [n]$, $j \in \{\pm 1\}$ and $r \in [m]$, it holds that

$$\langle \mathbf{w}_{y_i,r^*}^{(t)}, \boldsymbol{\xi}_i \rangle = \widetilde{\Theta}(1), \ \sum_{k\in\mathcal{B}_i}|\mathbf{w}_{y_i,r^*}^{(t)}[k]| \cdot |\boldsymbol{\xi}_i[k]| = \widetilde{\Theta}(1), \ \langle \mathbf{w}_{j,r}^{(t)}, \text{sgn}(j) \cdot \mathbf{v} \rangle \in [-o(1), O(\lambda^{-1}\eta)].$$

Lemma 5.4 shows that in the second stage, $\langle \mathbf{w}_{y_i,r}^{(t)}, \boldsymbol{\xi}_i \rangle$ will always be large while $\langle \mathbf{w}_{y_i,r}^{(t)}, y_i \cdot \mathbf{v} \rangle$ is still negative, or positive but extremely small. Next we will show that within polynomial steps, the algorithm can be guaranteed to find a point with small gradient.

**Lemma 5.5** (Convergence guarantee). If $\eta = O(d^{-1/2})$, then for any $t$ it holds that

$$L(\mathbf{W}^{(t+1)}) - L(\mathbf{W}^{(t)}) \le -\eta\|\nabla L(\mathbf{W}^{(t)})\|_1 + \widetilde{\Theta}(\eta^2 d).$$

Lemma 5.5 shows that we can pick a sufficiently small $\eta$ and $T = \text{poly}(n)/\eta$ to ensure that the algorithm can find a point with up to $O(1/(T\eta))$ in $\ell_1$ norm. Then we can show that given the results in Lemma 5.4, the formula of the algorithm output $\mathbf{W}^*$ can be precisely characterized, which we

can show that $\langle \mathbf{w}_{y_i,r}^*, y_i \cdot \mathbf{v}\rangle < 0$. This implies that the output model will be biased to fit the feature noise $-\alpha y\mathbf{v}$ but not the true one $\mathbf{v}$. Then when it comes to a fresh test example the model will fail to recognize its true feature. Also note that the noise in the test data is nearly independent of the noise in training data. Consequently, the model will not be able to identify the label of the test data and therefore cannot be better than a random guess.

## 5.2 PROOF SKETCH FOR GRADIENT DESCENT

Similar to the proof for Adam, we also decompose the entire training process into two stages.

**Stage I: Learning the pattern.** In this stage the gradient from training loss function is large and and the effect of regularization can be ignored. Unlike Adam that is sensitive to the sparsity of the feature vector or noise vector, gradient descent is more focusing on the $\ell_2$ norm of them, where the vector (which can be either feature or noise) with larger $\ell_2$ norm is more likely to be discovered and learned by GD. Note that the feature vector has a larger $\ell_2$ norm than the noise, we can show that gradient descent will learn the feature vector very quickly, while barely memorize the noise.

**Lemma 5.6.** Let $\Lambda_j^{(t)} = \max_{r \in [m]} \langle \mathbf{w}_{j,r}^{(t+1)}, j \cdot \mathbf{v}\rangle$, $\Gamma_{j,i}^{(t)} = \max_{r \in [m]} \langle \mathbf{w}_{j,r}^{(t)}, \boldsymbol{\xi}_i\rangle$, and $\Gamma_j^{(t)} = \max_{i:y_i=j} \Gamma_{j,i}^{(t)}$. Let $T_j$ be the iteration number that $\Lambda_j^{(t)}$ reaches $\Theta(1/m) = \widetilde{\Theta}(1)$, then we have $T_j = \widetilde{\Theta}(\sigma_0^{2-q})$ for all $j \in \{-1,1\}$. Moreover, let $T_0 = \max_j\{T_j\}$, then for all $t \leq T_0$ it holds that $\Gamma_j^{(t)} = \widetilde{O}(\sigma_0)$ for all $j \in \{-1,1\}$.

**Stage II: Regularizing the model.** Similar to Lemma 5.4, we show that in the second stage at which the impact of weight decay regularization cannot be ignored, the pattern of the training data learned in the first stage will remain unchanged.

**Lemma 5.7.** If $\eta \leq O(\sigma_0)$, it holds that $\Lambda_j^{(t)} = \widetilde{\Theta}(1)$ and $\Gamma_j^{(t)} = \widetilde{O}(\sigma_0)$ for all $t \geq \min_j T_j$.

The following lemma further shows that within polynomial steps, gradient descent is guaranteed to find a point with small gradient.

**Lemma 5.8.** If the learning rate satisfies $\eta = o(1)$, then for any $t \geq 0$ it holds that

$$L(\mathbf{W}^{(t+1)}) - L(\mathbf{W}^{(t)}) \leq -\frac{\eta}{2}\|\nabla L(\mathbf{W}^{(t)})\|_F^2.$$

Lemma 5.8 shows that we can pick a sufficiently small $\eta$ and $T = \text{poly}(n)/\eta$ to ensure that gradient descent can find a point with up to $O(1/(T\eta)^{1/2})$ in $\ell_2$ norm. By Lemma 5.7, it is clear that the output model of GD can well learn the feature vector while memorizing nearly nothing from the noise vectors, which can therefore achieve nearly zero test error.

**Experiments.** We perform experiments on synthetic data (generated according to Definition 3.1) to validate our theoretical findings: Adam performs stronger noise memorization than feature learning while GD performs stronger feature learning than noise memorization, when conducted on the data distribution constructed in Definition 3.1. We consider both the two-layer CNN model studied in this paper and a 5-layer CNN model for further justification. Experimental setup and results are deferred to Appendix A due to the page limit.

## 6 CONCLUDING REMARKS AND FUTURE WORK

In this paper, we study the generalization of Adam and compare it with gradient descent. We show that when training neural networks, Adam and GD starting from the same initialization can converge to different global solutions of the training objective with significantly different generalization errors, even with proper regularization. Our analysis reveals the fundamental difference between Adam and GD in learning features or noise, and demonstrates that this difference is closely tied to the nonconvex landscape of neural networks.

We would also like to remark several important research directions. First, our current result is for two-layer networks. Extending the results to deep networks could be an important next step, where we will not only look at the input data but also consider the output of each intermediate layer as "input". Second, our current data model is motivated by the image data (i.e., sparse feature and denser noise), where Adam has been observed to perform worse than SGD in terms of generalization. In fact, our theoretical analysis can lead to an opposite conclusion on the generalization comparison between Adam and GD if the noise is sparse and the feature is denser. Therefore, it would also be interesting to explore whether this is the case in other machine learning tasks such as natural language processing, where Adam is often observed to perform better than SGD.

ACKNOWLEDGEMENTS

We thank the anonymous reviewers and area chair for their helpful comments. YL is supported by the National Science Foundation CCF-2145703. QG is supported in part by the National Science Foundation CAREER Award 1906169, IIS-2008981 and the Sloan Research Fellowship.

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

# A    EXPERIMENTS

## A.1    EXPERIMENT DETAILS FOR FIGURE 1

The experiments in Figure 1 are performed by running (stochastic gradient) Adam and SGD in training AlexNet on the CIFAR-10 dataset. Specifically, the first layer of AlexNet is set as: kernel_size=11, stride=4, padding=2, in order to match the size of CIFAR-10 image. In terms of the input data, we use standard random crop and horizontal flip data augmentations. In terms of the model training, we set the batch size as $64$, the epoch number as $100$, the regularization parameter as $\lambda = 5 \times 10^{-4}$. Besides, we set the learning rate $\eta = 0.01$ for SGD and $\eta = 1 \times 10^{-5}$ for Adam, where $\beta_1$ and $\beta_2$ are set as their default values in pytorch.

## A.2    NUMERICAL EXPERIMENTS ON SYNTHETIC DATA

In this section we perform numerical experiments on the synthetic data generated according to Definition 3.1 to verify our main results. In particular, we set the problem dimension $d = 1000$, the training sample size $n = 200$ ($100$ positive examples and $100$ negative examples), feature vector $\mathbf{v} = [1, 0, \ldots, 0]^\top$, noise sparsity $s = 0.1d = 100$, standard deviation of noise $\sigma_p = 1/s^{1/2} = 0.1$, feature noise strength $\alpha = 0.2$, initialization scaling $\sigma_0 = 0.01$, regularization parameter $\lambda = 1 \times 10^{-5}$.

**Two-layer CNN model.**    We first consider the exact two-layer CNN model studied in the paper. We set the network width $m = 20$, activation function $\sigma(z) = \max\{0, z\}^3$, total iteration number $T = 1 \times 10^4$, and the learning rate $\eta = 5 \times 10^{-5}$ for Adam (default choices of $\beta_1$ and $\beta_2$ in pytorch), $\eta = 0.02$ for GD.

We first report the training error and test error achieved by the solutions found by SGD and Adam in Table 2, where the test error is calculated on a test dataset of size $10^4$. It is clear that both Adam and SGD can achieve zero training error, while they have entirely different results on the test data: SGD generalizes well and achieve zero test error; Adam generalizes worse than SGD and gives $> 0.5$ test error, which verifies our main result (Theorem 4.1).

| Algorithm | Adam | SGD |
|---|---|---|
| Training error | 0 | 0 |
| Test error | 0.884 | 0 |

Table 2: Training and test errors achieved by GD and Adam.

Moreover, we also calculate the inner products: $\max_r \langle \mathbf{w}_{1,r}, \mathbf{v} \rangle$ and $\min_i \max_r \langle \mathbf{w}_{1,r}, \boldsymbol{\xi}_i \rangle$, representing feature learning and noise memorization respectively, to verify our key lemmas. Here we only consider positive examples as the results for negative examples are similar. The results are reported in Figure 2. For Adam, from Figure 2(a), it can be seen that the algorithm will perform feature learning in the first few iterations and then entirely forget the feature (but fit feature noise), i.e., the feature learning is flipped, which verifies Lemma 5.3. In the meanwhile, the noise memorization happens in the entire training process and enjoys much faster rate than feature learning, which verifies Lemma 5.2. In addition, we can also observe that there are two stages for the increasing of $\min_i \max_r \langle \mathbf{w}_{1,r}, \boldsymbol{\xi}_i \rangle$: in the first stage $\min_i \max_r \langle \mathbf{w}_{1,r}, \boldsymbol{\xi}_i \rangle$ increases linearly, and in the second stage its increasing speed gradually slows down and $\min_i \max_r \langle \mathbf{w}_{1,r}, \boldsymbol{\xi}_i \rangle$ will remain in a constant order. This verifies Lemma 5.2 and Lemma 5.4. For GD, from Figure 2(b), it can be seen that the feature learning will dominate the noise memorization: feature learning will increase to a constant in the first stage and then remain in a constant order in the second stage; noise memorization will keep in a low level which is nearly the same as that at the initialization. This verifies Lemmas 5.6 and 5.7.

**5-Layer CNN model.**    We further perform numerical simulations for the deep neural network models. In particular, we consider a 5-layer CNN model: the first layer is exactly the same as the two-layer CNN model, followed by a 4-layer MLP with ReLU activation. The number of neurons is set

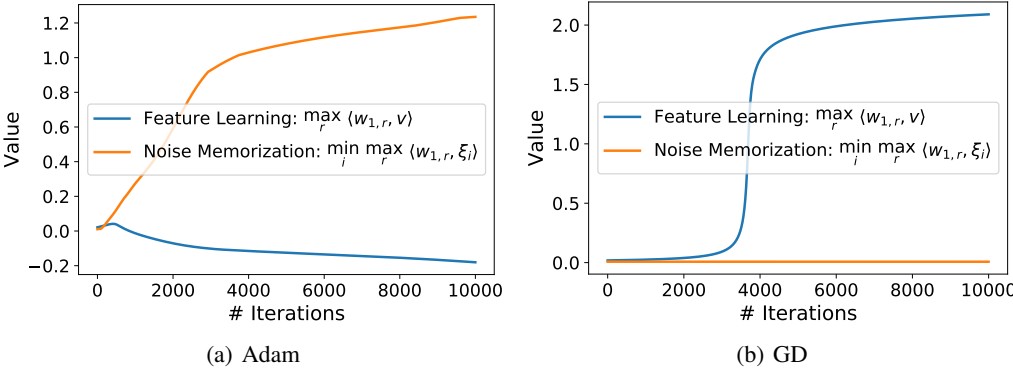

(a) Adam

(b) GD

Figure 2: Visualization of the feature learning $(\max_r \langle \mathbf{w}_{1,r}, \mathbf{v} \rangle)$ and noise memorization $(\min_i \max_r \langle \mathbf{w}_{1,r}, \boldsymbol{\xi}_i \rangle)$ for training the two-layer CNN model.

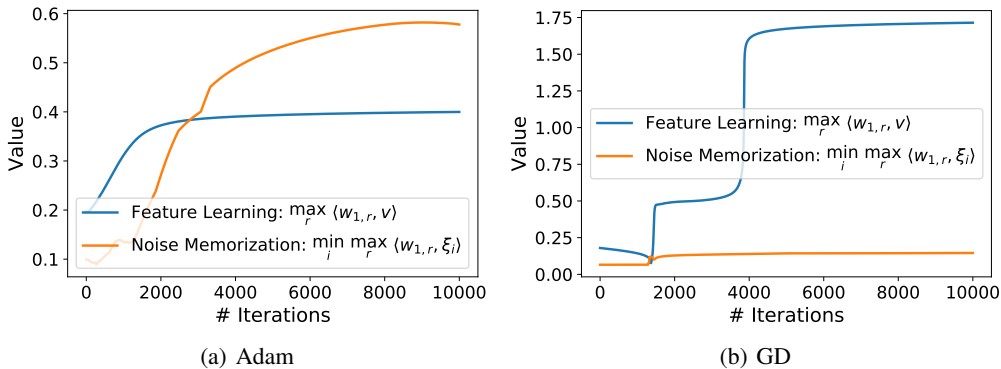

(a) Adam

(b) GD

Figure 3: Visualization of the feature learning $(\max_r \langle \mathbf{w}_{1,r}, \mathbf{v} \rangle)$ and noise memorization $(\min_i \max_r \langle \mathbf{w}_{1,r}, \boldsymbol{\xi}_i \rangle)$ for training the 5-layer CNN model.

as $m = 20$ for all layers. The total iteration number is $T = 1 \times 10^4$, the learning rate is $\eta = 5 \times 10^{-5}$ for Adam and $\eta = 0.1$ for GD.

We then show the feature learning and noise memorization of the first layer of this neural network model by calculating the inner products: $\max_r \langle \mathbf{w}_{1,r}, \mathbf{v} \rangle$ and $\min_i \max_r \langle \mathbf{w}_{1,r}, \boldsymbol{\xi}_i \rangle$, where $\mathbf{w}_{1,r}$ denotes the weights of the $r$-th neuron in the first layer. The results are shown in Figure 3. It can be observed that, when applied on the data distribution in Definition 3.1, Adam tends to perform stronger noise memorization than feature learning, while GD performs stronger feature learning and nearly negligible noise memorization. Moreover, we also visualize the first layer of the 5-layer CNN trained by Adam and GD in Figure 4. It can be seen that the CNN model found by Adam is clearly more "noisy" than that found by GD. This is consistent with our theoretical findings and empirical observation on the real-world dataset (i.e., Figure 1).

## B    EXTENSIONS TO MINI-BATCH STOCHASTIC GRADIENTS

One natural extension of our paper is proving the separation between mini-batch SGD (without replacement) and mini-batch Adam, which we believe is not difficult. In particular, let $\mathcal{I}_t$ of size $B$ be the set of indices of the mini-batch data used in the $t$-th iteration, the update rule of SGD is

$$\mathbf{w}_{j,r}^{(t+1)} = \mathbf{w}_{j,r}^{(t)} - \eta \cdot \frac{1}{B} \sum_{i \in \mathcal{I}_t} \nabla_{\mathbf{w}_{j,r}} L_i(\mathbf{W}^{(t)}) - \lambda \mathbf{w}_{j,r}^{(t)}.$$

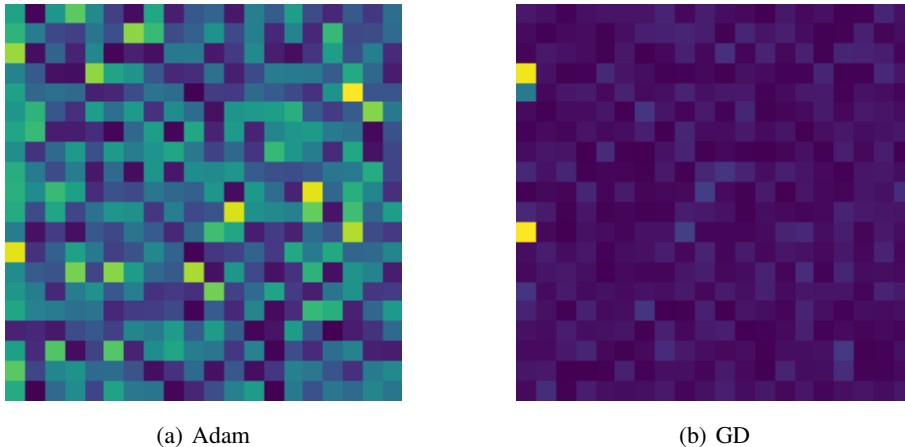

(a) Adam  (b) GD

Figure 4: Visualization of the first layer of the 5-layer CNN model on the synthetic dataset.

The update rule of mini-batch Adam is

$$\mathbf{m}_{j,r}^{(t+1)} = \beta_1 \mathbf{m}_{j,r}^{(t)} + (1-\beta_1) \cdot \left[ \frac{1}{B} \sum_{i \in \mathcal{I}_t} \nabla_{\mathbf{w}_{j,r}} L_i(\mathbf{W}^{(t)}) + \lambda \mathbf{w}_{j,r}^{(t)} \right],$$

$$\mathbf{v}_{j,r}^{(t+1)} = \beta_2 \mathbf{v}_{j,r}^{(t)} + (1-\beta_2) \cdot \left[ \frac{1}{B} \sum_{i \in \mathcal{I}_t} \nabla_{\mathbf{w}_{j,r}} L_i(\mathbf{W}^{(t)}) + \lambda \mathbf{w}_{j,r}^{(t)} \right]^2,$$

and

$$\mathbf{w}_{j,r}^{(t+1)} = \mathbf{w}_{j,r}^{(t)} - \eta \cdot \mathbf{m}_{j,r}^{(t)} / \sqrt{\mathbf{v}_{j,r}^{(t)} + \epsilon}.$$

Here the bias term is set as $\epsilon = \widetilde{\Theta}(\lambda \sigma_0)$. We claim that this parameter is introduced to guarantee that the regularization term will not dominate the training process when using stochastic gradients in Adam.

Then we will take a deeper look at the speeds of feature learning and noise learning for mini-batch SGD and Adam, where we focus on the period that $|\langle \mathbf{w}_{j,r}^{(t)}, \mathbf{v} \rangle|, |\langle \mathbf{w}_{j,r}^{(t)}, \boldsymbol{\xi}_i \rangle| = o(1)$ for all $j$, $i$, and $r$ (i.e., the pattern learning stage). This further implies that $|\ell_{j,i}^{(t)}| = 0.5 \pm o(1)$ for all $j$, $i$, and $t$. Thus in the following, we will assume that all $|\ell_{j,i}^{(t)}|$ has nearly the same quantity.

**Feature Learning.**   First, according to Definition 3.1, we know that the feature vector $\mathbf{v}$ and feature noise are the same for all data, which implies that the learning pattern of the feature coordinate will be largely the same as that of full-batch algorithms. In particular, for mini-batch Adam, we can show that the update of the first coordinate (i.e., feature coordinate) is similar to sign-GD when using sufficiently small learning rate $\eta = 1/\text{poly}(d)$ since all stochastic gradients $\nabla L_i(\mathbf{W}^{(t)})$ have the same component in this coordinate. Then using the fact that $|\ell_{j,i}^{(t)}|$'s are nearly the same for all $i$, we have

$$\langle \mathbf{w}_{j,r}^{(t+1)}, j\mathbf{v} \rangle \sim \langle \mathbf{w}_{j,r}^{(t)}, j\mathbf{v} \rangle + j\eta \cdot \text{sgn}\left( \sum_{i=1}^n y_i \ell_{j,i}^{(t)} \left[ \sigma'(\langle \mathbf{w}_{j,r}^{(t)}, y_i\mathbf{v} \rangle) - \alpha\sigma'(\langle \mathbf{w}_{j,r}^{(t)}, \boldsymbol{\xi}_i \rangle) \right] - n\lambda \mathbf{w}_{j,r}^{(t)}[1] \right).$$

which is the same as full-batch Adam (see (5.1)). For SGD, using the fact that $|\ell_{j,i}^{(t)}|$'s are nearly the same for all $i$, we can get that

$$\langle \mathbf{w}_{j,r}^{(t+1)}, j\mathbf{v} \rangle \sim (1 - \eta\lambda) \cdot \langle \mathbf{w}_{j,r}^{(t)}, j \cdot \mathbf{v} \rangle$$
$$+ \frac{\eta}{n} \cdot j \cdot \left( \sum_{i=1}^n y_i \ell_{j,i}^{(t)} \sigma'(\langle \mathbf{w}_{j,r}^{(t)}, y_i\mathbf{v} \rangle) - \alpha \sum_{i=1}^n y_i \ell_{j,i}^{(t)} \sigma'(\langle \mathbf{w}_{j,r}^{(t)}, \boldsymbol{\xi}_i \rangle) \right)$$

which is also the same as that of GD (see (C.28)).

**Noise Memorization.** Note that due to the normalization term $\mathbf{v}_{j,r}^{(t)}$ in the Adam update, all coordinates will be updated with nearly the same amount. Therefore, we only need to count the number of coordinates that are updated by full-batch Adam and mini-batch Adam.

Recall that we have shown that using mini-batch gradients will not affect the feature learning. However, the noise memorization will be slightly different, since in each iteration, full-batch Adam can update $\Theta(ns)$ coordinates while mini-batch Adam can only update $\widetilde{\Theta}(Bs)$ coordinates. To show this, we note that for any coordinate $k \neq 1$, the gradient momentum of full-batch Adam is

$$\mathbf{m}_{j,r}^{(t)}[k] \sim \sum_{\tau=0}^{\bar{\tau}} \beta_1^\tau (1 - \beta_1) \cdot \frac{1}{n} \sum_{i \in [n]} \left[ \nabla_{\mathbf{w}_{j,r}} L_i(\mathbf{W}^{(t-\tau)})[k] + \lambda \mathbf{w}_{j,r}^{(t-\tau)}[k] \right],$$

while for mini-batch Adam,

$$\mathbf{m}_{j,r}^{(t)}[k] \sim \sum_{\tau=0}^{\bar{\tau}} \beta_1^\tau (1 - \beta_1) \cdot \frac{1}{B} \sum_{i \in \mathcal{I}_{t-\tau}} \left[ \nabla_{\mathbf{w}_{j,r}} L_i(\mathbf{W}^{(t-\tau)})[k] + \lambda \mathbf{w}_{j,r}^{(t-\tau)}[k] \right],$$

where we only maintain the recent $\bar{\tau} = \mathrm{polylog}(n)$ gradients since for $\tau \leq t - \bar{\tau}$, the decaying terms $(\beta_1)^\tau \leq (\beta_i)^{\bar{\tau}}$ becomes negligible. Therefore, by comparing the above two equations and applying Definition 3.1, it is clear that for full-batch Adam can update all noise coordinates, i.e., $k \in \cup_{i \in [n]} \mathcal{B}_i$, which is of size $\Theta(ns)$. In contrast, mini-batch Adam can only update a subset of noise coordinates, i.e., $k \in \cup_{\tau \in [\bar{\tau}]} \cup_{i \in [\mathcal{I}_{t-\tau}]} \mathcal{B}_i$, which is of size $\bar{\tau} B s = \widetilde{\Theta}(Bs)$. This further implies that in each epoch (one pass of the data, $\Theta(n/B)$ steps), the noise coordinates in $\mathcal{B}_i$ will be updated by *mini-batch Adam* in at most $\bar{\tau} = \widetilde{\Theta}(1)$ steps, while within the same amount of iterations, the noise coordinates in $\mathcal{B}_i$ will be updated by *full-batch* Adam for $\Theta(n/B)$ steps, suggesting that mini-batch Adam admits a slower rate of noise memorization by a $\widetilde{\Theta}(n/B)$ factor.

For SGD, it is easy to show that the rate of noise memorization will still be nearly the same as that of GD. In particular, during each training epoch ($\Theta(n/B)$ steps), SGD will learn the noise vector $\boldsymbol{\xi}_i$ in only one step with the mini-batch gradient $\frac{1}{B} \nabla L_i(\mathbf{W}^{(\tau)})$ for some $\tau$ in this epoch, while within the same amount of steps, GD will learn the noise vector $\boldsymbol{\xi}_i$ in all $\Theta(n/B)$ steps but with strength $\frac{1}{n} \nabla L_i(\mathbf{W}^{(\tau)})$, giving the same total learning ability. This suggests that SGD admits a nearly the same rate of noise memorization compared to GD.

Overall, we are able to deliver the following lemmas that characterize the feature learning and noise memorization of SGD and stochastic gradient Adam in training Stage I.

**Lemma B.1** (SGD, Informal). Suppose the training data is generated according to Definition 3.1, then given proper configurations of $\lambda$ and $\epsilon$ and sufficiently small learning rate $\eta$, define $\Lambda_j^{(t)} = \max_{r \in [m]} \langle \mathbf{w}_{j,r}^{(t)}, j \cdot \mathbf{v} \rangle$ and $\Gamma_{j,i}^{(t)} = \max_{r \in [m]} \langle \mathbf{w}_{j,r}^{(t)}, \boldsymbol{\xi}_i \rangle$, for any $t_0$ satisfying $\Lambda_j^{(t_0)}, \Gamma_{j,i}^{(t_0)} = o(1/\mathrm{polylog}(n))$, we have the following one-epoch update of feature learning and noise memorization for SGD

$$\Lambda_j^{(t_0 + \lfloor n/B \rfloor)} \geq \Lambda_j^{(t_0)} + \eta \cdot \Theta\left( \frac{n}{B} \cdot (\Lambda_j^{(t_0)})^{q-1} \right)$$

$$\Gamma_j^{(t_0 + \lfloor n/B \rfloor)} \leq \Gamma_j^{(t_0)} + \eta \cdot \Theta\left( \frac{\eta s \sigma_p^2}{B} \cdot (\Gamma_j^{(t_0)})^{q-1} \right).$$

**Lemma B.2** (Stochastic gradient Adam, Informal). Suppose the training data is generated according to Definition 3.1, then given proper configurations of $\lambda$ and $\epsilon$, for any $t_0 \leq T_0$ with $T_0 = \widetilde{O}\left( \frac{n}{\eta B s \sigma_p} \right)$ and any $i \in [n]$, we have the following one-epoch update of feature learning and noise memorization

$$\langle \mathbf{w}_{j,r}^{(t_0 + \lfloor n/B \rfloor)}, j \cdot \mathbf{v} \rangle \leq \langle \mathbf{w}_{j,r}^{(t_0)}, j \cdot \mathbf{v} \rangle + \Theta(n\eta/B), \quad \langle \mathbf{w}_{y_i,r}^{(t_0 + \lfloor n/B \rfloor)}, \boldsymbol{\xi}_i \rangle = \langle \mathbf{w}_{y_i,r}^{(t_0)}, \boldsymbol{\xi}_i \rangle + \widetilde{\Theta}(\eta s \sigma_p).$$

To sum up, we have shown that (1) mini-batch SGD and mini-batch Adam will not change the learning speed of feature vector $\mathbf{v}$ compared to their full-batch counterparts, i.e., Lemma C.3 and (C.31) (needs to covert to $\lfloor n/B \rfloor$ iterations); (2) mini-batch Adam reduces the noise memorization speed of full-batch Adam by a $\widetilde{\Theta}(n/B)$ factor, while mini-batch SGD has nearly the same noise memorization speed compared to full-batch GD, by comparing to Lemma C.3 and (C.32)). Additionally,

recall that in our paper, the separation between Adam and GD is characterized by a $\mathrm{poly}(d)$ factor: the speed of feature learning in Adam and GD, and the rate of noise memorization in GD are both in the order of $O(\eta)$ (in each step), while the rate of noise memorization in Adam is proportional to the number of nonzero entries, which is in the order of $\eta \cdot \mathrm{poly}(d)$. Therefore, the separation between mini-batch SGD and mini-batch Adam in terms of the generalization error can still hold under a stronger over-parameterization condition $s\sigma_p = \Theta\big(d^{1/4}/(n\mathrm{polylog}(n))\big) = \omega(n/B)$ (in contrast, the over-parameterization condition for full-batch Adam is $s\sigma_p = \omega(1)$).

## C PROOF OF THEOREM 4.1: NONCONVEX CASE

In the beginning of the proof we first present the following useful lemma.

### C.1 PRELIMINARIES

We first recall the magnitude of all parameters:

$$d = \mathrm{poly}(n),\ \eta = \frac{1}{\mathrm{poly}(n)},\ s = \Theta\left(\frac{d^{1/2}}{n^2}\right),\ \sigma_p^2 = \Theta\left(\frac{1}{s \cdot \mathrm{polylog}(n)}\right),\ \sigma_0^2 = \Theta\left(\frac{1}{d^{1/2}}\right),$$

$$m = \mathrm{polylog}(n),\ \alpha = \Theta\big(\sigma_p \cdot \mathrm{polylog}(n)\big),\ \lambda = O\left(\frac{1}{d^{(q-1)/4}n \cdot \mathrm{polylog}(n)}\right).$$

Here $\mathrm{poly}(n)$ denotes a polynomial function of $n$ with degree of a sufficiently large constant, $\mathrm{poly}(n)$ denotes a polynomial function of $\log(n)$ with degree of a sufficiently large constant. Based on the parameter configuration, we claim that the following equations hold, which will be frequently used in the subsequent proof.

$$\lambda = o\left(\frac{\sigma_0^{q-2}\sigma_p}{n}\right),\ \alpha = \omega\big((s\sigma_p)^{1-q}\sigma_0^{q-1}\big),\ \sigma_0 = o\left(\frac{1}{s\sigma_p}\right),\ \alpha = o\left(\frac{s\sigma_p^2}{n}\right),\ \eta = o\big(\lambda\sigma_0^q\sigma_p^q\big).$$

**Lemma C.1** (Non-overlapping support). Let $\{(\mathbf{x}_i, y_i)\}_{i=1,\dots,n}$ be the training dataset sampled according to Definition 3.1. Moreover, let $\mathcal{B}_i = \mathrm{supp}(\boldsymbol{\xi}_i)\backslash\{1\}$ be the support of $\mathbf{x}_i$ except the first coordinate[2]. Then with probability at least $1 - n^{-2}$, $\mathcal{B}_i \cap \mathcal{B}_j = \emptyset$ for all $i, j \in [n]$.

*Proof of Lemma C.1.* For any fixed $k \in [n]$ and $j \in \mathrm{supp}(\boldsymbol{\xi}_k)\backslash\{1\}$, by the model assumption we have

$$\mathbb{P}\{(\boldsymbol{\xi}_i)_j \neq 0\} = s/(d-1),$$

for all $i \in [n]\backslash\{k\}$. Therefore by the fact that the data samples are independent, we have

$$\mathbb{P}(\exists i \in [n]\backslash\{k\} : (\xi_i)_j \neq 0) = 1 - [1 - s/(d-1)]^n.$$

Applying a union bound over all $k \in [n]$ and $j \in \mathrm{supp}(\boldsymbol{\xi}_k)\backslash\{1\}$, we obtain

$$\mathbb{P}(\exists k \in [n], j \in \mathrm{supp}(\boldsymbol{\xi}_k)\backslash\{1\}, i \in [n]\backslash\{k\} : (\boldsymbol{\xi}_i)_j \neq 0) \leq n \cdot s \cdot \{1 - [1 - s/(d-1)]^n\}. \quad \text{(C.1)}$$

By the data distribution assumption we have $s \leq \sqrt{d}/(2n^2)$, which clearly implies $s/(d-1) \leq 1/2$. Therefore we have

$$\begin{aligned}
n \cdot s \cdot [1 - (1 - s/d)^n] &= n \cdot s \cdot \{1 - \exp[n\log(1 - s/(d-1))]\} \\
&\leq n \cdot s \cdot [1 - \exp(n \cdot 2s/(d-1))] \\
&\leq n \cdot s \cdot [1 - \exp(n \cdot 4s/d)] \\
&\leq n \cdot s \cdot (4ns/d) \\
&= 4n^2 s^2/d \\
&\leq n^{-2},
\end{aligned}$$

where the first inequality follows by the inequalities $\log(1 - z) \geq -2z$ for $z \in [0, 1/2]$, the second inequality follows by $s/(d-1) \geq 2s/d$, the third inequality follows by the inequality $1 - \exp(-z) \leq z$ for $z \in \mathbb{R}$, and the last inequality follows by the assumption that $s \leq \sqrt{d}/(2n^2)$. Plugging the bound above into (C.1) finishes the proof.

$\square$

---

[2]Recall that all data inputs have nonzero first coordinate by Definition 3.1

## C.2 PROOF FOR ADAM

Before moving to the detailed proof, we first state the update rules of feature learning and noise memorization when the sign gradient is applied.

$$
\begin{aligned}
\langle \mathbf{w}_{j,r}^{(t+1)}, j\mathbf{v} \rangle &= \langle \mathbf{w}_{j,r}^{(t)}, j\mathbf{v} \rangle - \eta \cdot \langle \mathrm{sgn}(\nabla_{\mathbf{w}_{j,r}} L(\mathbf{W}^{(t)})), j\mathbf{v} \rangle \\
&= \langle \mathbf{w}_{j,r}^{(t)}, j\mathbf{v} \rangle + j\eta \cdot \mathrm{sgn}\bigg( \sum_{i=1}^{n} y_i \ell_{j,i}^{(t)} \big[ \sigma'(\langle \mathbf{w}_{j,r}^{(t)}, y_i\mathbf{v} \rangle) - \alpha\sigma'(\langle \mathbf{w}_{j,r}^{(t)}, \boldsymbol{\xi}_i \rangle) \big] - n\lambda\mathbf{w}_{j,r}^{(t)}[1] \bigg),
\end{aligned}
$$
(C.2)

where $\ell_{j,i}^{(t)} := \mathbb{1}_{y_i=j} - \mathrm{logit}_j(F, \mathbf{x}_i)$ and $\mathrm{logit}_j(F, \mathbf{x}_i) = \frac{e^{F_j(\mathbf{W}, \mathbf{x}_i)}}{\sum_{k \in \{-1,1\}} e^{F_k(\mathbf{W}, \mathbf{x}_i)}}$. From (C.2) we can observe three terms in the signed gradient. Specifically, the first term represents the gradient over the feature patch, the second term stems from the feature noise term in the noise patch (see Definition 3.1), and the last term is the gradient of the weight decay regularization. On the other hand, the memorization of the noise vector $\boldsymbol{\xi}_i$ can be described by the following update rule,

$$
\begin{aligned}
\langle \mathbf{w}_{y_i,r}^{(t+1)}, \boldsymbol{\xi}_i \rangle &= \langle \mathbf{w}_{y_i,r}^{(t)}, \boldsymbol{\xi}_i \rangle - \eta \cdot \langle \mathrm{sgn}(\nabla_{\mathbf{w}_{y_i,r}} L(\mathbf{W}^{(t)})), \boldsymbol{\xi}_i \rangle \\
&= \langle \mathbf{w}_{y_i,r}^{(t)}, \boldsymbol{\xi}_i \rangle + \eta \cdot \sum_{k \in \mathcal{B}_i} \Big\langle \mathrm{sgn}\Big( \ell_{y_i,i}^{(t)} \sigma'(\langle \mathbf{w}_{y_i,r}^{(t)}, \boldsymbol{\xi}_i \rangle) \boldsymbol{\xi}_i[k] - n\lambda\mathbf{w}_{y_i,r}^{(t)}[k] \Big), \boldsymbol{\xi}_i[k] \Big\rangle \\
&\quad - \alpha y_i \eta \cdot \mathrm{sgn}\bigg( \sum_{i=1}^{n} y_i \ell_{y_i,i}^{(t)} \big[ \sigma'(\langle \mathbf{w}_{y_i,r}^{(t)}, y_i\mathbf{v} \rangle) - \alpha\sigma'(\langle \mathbf{w}_{y_i,r}^{(t)}, \boldsymbol{\xi}_i \rangle) \big] - n\lambda\mathbf{w}_{y_i,r}^{(t)}[1] \bigg).
\end{aligned}
$$
(C.3)

In this subsection we first provide the following lemma that shows for most of the coordinate (with slightly large gradient), the Adam update is similar to signGD update (up to some constant factors). In the remaining proof for Adam, we will largely apply this lemma to get a signGD-like result for Adam (similar to the technical lemmas in Section 5). Besides, the proofs for all lemmas in Section 5 can be viewed as a simplified version of the proofs for technical lemmas for Adam, thus are omitted in the paper.

**Lemma C.2** (Closeness to SignGD). Recall the update rule of Adam, let $\mathbf{W}^{(t)}$ be the $t$-th iterate of the Adam algorithm. Suppose that $\langle \mathbf{w}_{j,r}^{(t)}, \mathbf{v} \rangle, \langle \mathbf{w}_{j,r}^{(t)}, \boldsymbol{\xi}_i \rangle = \widetilde{\Theta}(1)$ for all $j \in \{\pm1\}$ and $r \in [m]$. Then if $\beta_2 \geq \beta_1^2$, we have

- For all $k \in [d]$,

$$
\left| \frac{\mathbf{m}_{j,r}^{(t)}[k]}{\sqrt{\mathbf{v}_{j,r}^{(t)}[k]}} \right| \leq \Theta(1).
$$

- For every $k \notin \cup_{i=1}^{n} \mathcal{B}_i$ (including $k = 1$) we have either $|\nabla_{\mathbf{w}_{j,r}} L(\mathbf{W}^{(t)})[k]| \leq \widetilde{\Theta}(\eta)$ or

$$
\frac{\mathbf{m}_{j,r}^{(t)}[k]}{\sqrt{\mathbf{v}_{j,r}^{(t)}[k]}} = \mathrm{sgn}(\nabla_{\mathbf{w}_{j,r}} L(\mathbf{W}^{(t)})[k]) \cdot \Theta(1).
$$

- For every $k \in \mathcal{B}_i$, we have $|\nabla_{\mathbf{w}_{j,r}} L(\mathbf{W}^{(t)})[k]| \leq \widetilde{\Theta}(\eta n^{-1} s\sigma_p |\ell_{j,i}^{(t)}|) \leq \widetilde{\Theta}(\eta s\sigma_p)$ or

$$
\frac{\mathbf{m}_{j,r}^{(t)}[k]}{\sqrt{\mathbf{v}_{j,r}^{(t)}[k]}} = \mathrm{sgn}(\nabla_{\mathbf{w}_{j,r}} L(\mathbf{W}^{(t)})[k]) \cdot \Theta(1).
$$

*Proof.* First recall that the gradient $\nabla_{\mathbf{w}_{j,r}} L(\mathbf{W}^{(t)})$ can be calculated as

$$
\nabla_{\mathbf{w}_{j,r}} L(\mathbf{W}^{(t)}) = -\frac{1}{n}\bigg[ \sum_{i=1}^{n} y_i \ell_{j,i}^{(t)} \sigma'(\langle \mathbf{w}_{j,r}^{(t)}, y_i\mathbf{v} \rangle) \cdot \mathbf{v} + \sum_{i=1}^{n} \ell_{j,i}^{(t)} \cdot \sigma'(\langle \mathbf{w}_{j,r}^{(t)}, y_i\boldsymbol{\xi}_i \rangle) \cdot \boldsymbol{\xi}_i \bigg] + \lambda\mathbf{w}_{j,r}^{(t)}.
$$

More specifically, for the first coordinate of $\nabla_{\mathbf{w}_{j,r}} L(\mathbf{W}^{(t)})$, we have

$$\nabla_{\mathbf{w}_{j,r}} L(\mathbf{W}^{(t)})[1] = -\frac{1}{n}\left[\sum_{i=1}^{n} y_i \ell_{j,i}^{(t)} \sigma'(\langle \mathbf{w}_{j,r}^{(t)}, y_i \mathbf{v}\rangle) - \alpha \sum_{i=1}^{n} y_i \ell_{j,i}^{(t)} \cdot \sigma'(\langle \mathbf{w}_{j,r}^{(t)}, \boldsymbol{\xi}_i\rangle)\right] + \lambda \mathbf{w}_{j,r}^{(t)}[1]. \tag{C.4}$$

For any $k \in \mathcal{B}_i$, by Lemma C.1 we know that the gradient over this coordinate only depends on the training data $\boldsymbol{\xi}_i$, therefore, we have

$$\nabla_{\mathbf{w}_{j,r}} L(\mathbf{W}^{(t)})[k] = -\frac{1}{n}\ell_{j,i}^{(t)} \sigma'(\langle \mathbf{w}_{j,r}^{(t)}, \boldsymbol{\xi}_i\rangle)\boldsymbol{\xi}_i[k] + \lambda \mathbf{w}_{j,r}^{(t)}[k]. \tag{C.5}$$

For the remaining coordinates, we have

$$\nabla_{\mathbf{w}_{j,r}} L(\mathbf{W}^{(t)})[k] = \lambda \mathbf{w}_{j,r}^{(t)}[k]. \tag{C.6}$$

Now let us focus on the moving averaged gradient $\mathbf{m}_{j,r}^{(t)}$ and squared gradient $\mathbf{v}_{j,r}^{(t)}$. We first show that for all $k \in [d]$, it holds that

$$\frac{\left|\mathbf{m}_{j,r}^{(t)}[k]\right|}{\sqrt{\mathbf{v}_{j,r}^{(t)}[k]}} \le \Theta(1). \tag{C.7}$$

By the update rule of $\mathbf{m}_{j,r}^{(t)}$, we have

$$\mathbf{m}_{j,r}^{(t)}[k] = \beta_1 \mathbf{m}_{j,r}^{(t-1)}[k] + (1 - \beta_1) \cdot \nabla_{\mathbf{w}_{j,r}} L(\mathbf{W}^{(t)})[k]$$

$$= \sum_{\tau=0}^{t} \beta_1^{\tau}(1 - \beta_1) \cdot \nabla_{\mathbf{w}_{j,r}} L(\mathbf{W}^{(t-\tau)})[k].$$

Similarly, we also have

$$\mathbf{v}_{j,r}^{(t)}[k] = \sum_{\tau=0}^{t} \beta_2^{\tau}(1 - \beta_2) \cdot \nabla_{\mathbf{w}_{j,r}} L(\mathbf{W}^{(t-\tau)})[k]^2.$$

Then by Cauchy-Schwartz inequality we have

$$\left(\mathbf{m}_{j,r}^{(t)}[k]\right)^2 \le \left(\sum_{\tau=0}^{t} \frac{[\beta_1^{\tau}(1 - \beta_1)]^2}{\alpha_{\tau}^2} \cdot \nabla_{\mathbf{w}_{j,r}} L(\mathbf{W}^{(t-\tau)})[k]^2\right) \cdot \left(\sum_{\tau=0}^{t} \alpha_{\tau}^2\right).$$

Let $\alpha_{\tau}^2 = \frac{[\beta_1^{\tau}(1-\beta_1)]^2}{\beta_2^{\tau}(1-\beta_2)}$, which forms an exponentially decaying sequence if $\beta_2 \ge \beta_1^2$. Therefore, we have $\sum_{\tau=0}^{t} \alpha_{\tau}^2 = \Theta(1)$ and the above inequality implies that

$$\left(\mathbf{m}_{j,r}^{(t)}[k]\right)^2 \le \mathbf{v}_{j,r}^{(t)}[k] \cdot \Theta(1),$$

which proves (C.7).

Now we are going to prove the main argument of this lemma. Note that $\mathbf{m}_{j,r}^{(t)}$, which is a weighted average of all historical gradients, where the weights decay exponentially fast, then we can take on a threshold $\bar{\tau} = \text{polylog}(\eta^{-1})$ such that $\sum_{\tau=\bar{\tau}}^{t} \beta_1^{\tau}(1 - \beta_1) = \frac{1}{\text{poly}(\eta^{-1})}$. Then for each $k \in [d]$ we have

$$\mathbf{m}_{j,r}^{(t)}[k] = \sum_{\tau=0}^{\bar{\tau}} \beta_1^{\tau}(1 - \beta_1) \cdot \nabla_{\mathbf{w}_{j,r}} L(\mathbf{W}^{(t-\tau)})[k] + \sum_{\tau=\bar{\tau}}^{t} \beta_1^{\tau}(1 - \beta_1) \cdot \nabla_{\mathbf{w}_{j,r}} L(\mathbf{W}^{(t-\tau)})[k]$$

$$= \sum_{\tau=0}^{\bar{\tau}} \beta_1^{\tau}(1 - \beta_1) \cdot \nabla_{\mathbf{w}_{j,r}} L(\mathbf{W}^{(t-\tau)})[k] \pm \frac{1}{\text{poly}(\eta^{-1})},$$

where in the last equality we use the fact that $|\nabla_{\mathbf{w}_{j,r}} L(\mathbf{W}^{(t-\tau)})[k]| = \widetilde{O}(1)$ for all $k \in [d]$. Similarly, we can also have the following on $\mathbf{v}_{j,r}^{(t)}$,

$$\mathbf{v}_{j,r}^{(t)}[k] = \sum_{\tau=0}^{\bar{\tau}} \beta_2^\tau (1 - \beta_2) \cdot \nabla_{\mathbf{w}_{j,r}} L(\mathbf{W}^{(t-\tau)})[k]^2 \pm \frac{1}{\text{poly}(\eta^{-1})}.$$

Here we slightly abuse the notation by using the same $\bar{\tau}$. Then we have

$$\frac{\mathbf{m}_{j,r}^{(t)}[k]}{\sqrt{\mathbf{v}_{j,r}^{(t)}[k]}} = \frac{\sum_{\tau=0}^{\bar{\tau}} \beta_1^\tau (1 - \beta_1) \cdot \nabla_{\mathbf{w}_{j,r}} L(\mathbf{W}^{(t-\tau)})[k] \pm \frac{1}{\text{poly}(\eta^{-1})}}{\sqrt{\sum_{\tau=\bar{\tau}}^{\bar{\tau}} \beta_2^\tau (1 - \beta_2) \cdot \nabla_{\mathbf{w}_{j,r}} L(\mathbf{W}^{(t-\tau)})[k]^2 \pm \frac{1}{\text{poly}(\eta^{-1})}}}.$$

In order to prove the main argument of this lemma, the key is to show that within $\bar{\tau}$ iterations, the gradient $\nabla_{\mathbf{w}_{j,r}} L(\mathbf{W}^{(t)})[k]$ barely changes. In particular, by (C.7), we have the update of each coordinate in one step is at most $\Theta(\eta)$. This implies that

$$\left| \langle \mathbf{w}_{j,r}^{(t)}, \mathbf{v} \rangle - \langle \mathbf{w}_{j,r}^{(\tau)}, \mathbf{v} \rangle \right| \leq \Theta(\eta\bar{\tau}),$$
$$\left| \langle \mathbf{w}_{j,r}^{(t)}, \boldsymbol{\xi}_i \rangle - \langle \mathbf{w}_{j,r}^{(\tau)}, \boldsymbol{\xi}_i \rangle \right| \leq \Theta(\eta\bar{\tau} s \sigma_p),$$
$$\left| \mathbf{w}_{j,r}^{(t)}[k] - \mathbf{w}_{j,r}^{(\tau)}[k] \right| \leq \Theta(\eta\bar{\tau}).$$

Then applying the fact that $|\langle \mathbf{w}_{j,r}^{(\tau)}, \mathbf{v} \rangle| \leq \widetilde{\Theta}(1)$ and $|\langle \mathbf{w}_{j,r}^{(\tau)}, \boldsymbol{\xi}_i \rangle| \leq \widetilde{\Theta}(1)$, we further have

$$\left| F_j(\mathbf{W}^{(\tau)}, \mathbf{x}_i) - F_j(\mathbf{W}^{(t)}, \mathbf{x}_i) \right| \leq \Theta(m\eta\bar{\tau} s \sigma_p) = \widetilde{\Theta}(\eta\bar{\tau} s \sigma_p),$$

where we use the fact that $m = \widetilde{\Theta}(1)$ and $s\sigma_p = \omega(1)$. Then it holds that

$$\ell_{j,i}^{(\tau)} = \frac{e^{F_j(\mathbf{W}^{(\tau)}, \mathbf{x}_i)}}{\sum_{k \in \{-1,1\}} e^{F_k(\mathbf{W}^{(\tau)}, \mathbf{x}_i)}}$$
$$\leq \frac{e^{F_j(\mathbf{W}^{(t)}, \mathbf{x}_i) + \widetilde{\Theta}(\eta\bar{\tau} s \sigma_p)}}{e^{F_j(\mathbf{W}^{(t)}, \mathbf{x}_i) + \widetilde{\Theta}(\eta\bar{\tau} s \sigma_p)} + e^{F_{-j}(\mathbf{W}^{(t)}, \mathbf{x}_i) - \widetilde{\Theta}(\eta\bar{\tau} s \sigma_p)}}$$
$$= \text{sgn}(\ell_{j,i}^{(t)}) \cdot \Theta(|\ell_{j,i}^{(t)}|),$$

where we use the fact that $\widetilde{\Theta}(\eta\bar{\tau} s \sigma_p) = o(1)$. Similarly, we can also show that $\ell_{j,i}^{(\tau)} \geq \text{sgn}(\ell_{j,i}^{(t)}) \cdot \Theta(|\ell_{j,i}^{(t)}|)$, which further implies

$$\ell_{j,i}^{(\tau)} = \text{sgn}(\ell_{j,i}^{(t)}) \cdot \Theta(|\ell_{j,i}^{(t)}|)$$

for all $\tau \in [t - \bar{\tau}, t]$. Note that $|\ell_{j,i}^{(\tau)}| \leq 1$, then it holds that

$$\ell_{j,i}^{(\tau)} \sigma'(\langle \mathbf{w}_{j,r}^{(\tau)}, \mathbf{v} \rangle) = \text{sgn}(\ell_{j,i}^{(t)}) \cdot \Theta(|\ell_{j,i}^{(t)}|) \cdot \sigma'(\langle \mathbf{w}_{j,r}^{(\tau)}, \mathbf{v} \rangle)$$
$$\leq \text{sgn}(\ell_{j,i}^{(t)}) \cdot \Theta(|\ell_{j,i}^{(t)}|) \cdot \sigma'(\langle \mathbf{w}_{j,r}^{(t)}, \mathbf{v} \rangle) + \Theta(|\ell_{j,i}^{(t)}|) \cdot \widetilde{\Theta}(\eta\bar{\tau}).$$

We can also similarly derive the following

$$\ell_{j,i}^{(\tau)} \sigma'(\langle \mathbf{w}_{j,r}^{(\tau)}, \mathbf{v} \rangle) \geq \text{sgn}(\ell_{j,i}^{(t)}) \cdot \Theta(|\ell_{j,i}^{(t)}|) \cdot \sigma'(\langle \mathbf{w}_{j,r}^{(t)}, \mathbf{v} \rangle) - \Theta(|\ell_{j,i}^{(t)}|) \cdot \widetilde{\Theta}(\eta\bar{\tau}),$$
$$\ell_{j,i}^{(\tau)} \sigma'(\langle \mathbf{w}_{j,r}^{(\tau)}, \boldsymbol{\xi}_i \rangle) \leq \text{sgn}(\ell_{j,i}^{(t)}) \cdot \Theta(|\ell_{j,i}^{(t)}|) \cdot \sigma'(\langle \mathbf{w}_{j,r}^{(t)}, \boldsymbol{\xi}_i \rangle) + \Theta(|\ell_{j,i}^{(t)}|) \cdot \widetilde{\Theta}(\eta\bar{\tau} s \sigma_p),$$
$$\ell_{j,i}^{(\tau)} \sigma'(\langle \mathbf{w}_{j,r}^{(\tau)}, \boldsymbol{\xi}_i \rangle) \geq \text{sgn}(\ell_{j,i}^{(t)}) \cdot \Theta(|\ell_{j,i}^{(t)}|) \cdot \sigma'(\langle \mathbf{w}_{j,r}^{(t)}, \boldsymbol{\xi}_i \rangle) - \Theta(|\ell_{j,i}^{(t)}|) \cdot \widetilde{\Theta}(\eta\bar{\tau} s \sigma_p).$$

Combining the above results, applying (C.4), (C.5), and (C.6), we can show that for the first coordinate, we have

$$\nabla_{\mathbf{w}_{j,r}} L(\mathbf{W}^{(\tau)})[1] = \Theta\left( \nabla_{\mathbf{w}_{j,r}} L(\mathbf{W}^{(t)})[1] \right) \pm \Theta\left( \frac{1}{n} \sum_{i=1}^{n} |\ell_{j,i}^{(t)}| \right) \cdot \widetilde{O}(\eta\bar{\tau}) \pm \Theta(\lambda\eta\bar{\tau});$$

for any $k \in \mathcal{B}_i$, we have

$$\nabla_{\mathbf{w}_{j,r}} L(\mathbf{W}^{(\tau)})[k] = \Theta\Big(\nabla_{\mathbf{w}_{j,r}} L(\mathbf{W}^{(t)})[k]\Big) \pm \Theta\Big(\frac{|\ell_{j,i}^{(t)}|}{n}\Big) \cdot \widetilde{O}(\eta\bar{\tau}s\sigma_p) \pm \Theta(\lambda\eta\bar{\tau});$$

and for remaining coordinates, we have

$$\nabla_{\mathbf{w}_{j,r}} L(\mathbf{W}^{(\tau)})[k] = \Theta\Big(\nabla_{\mathbf{w}_{j,r}} L(\mathbf{W}^{(t)})[k]\Big) \pm \Theta(\lambda\eta\widetilde{\tau}).$$

Now we can plug the above results into the formula of $\mathbf{m}_{j,r}^{(t)}$ and $\mathbf{v}_{j,r}^{(t)}$. Using the fact that $\bar{\tau} = \widetilde{\Theta}(1)$, $\lambda = o(1)$, and $|\ell_{j,i}^{(t)}| \le 1$, we have for all $k = 1$ or $k \notin \mathcal{B}_i$ for any $i$,

$$\frac{\mathbf{m}_{j,r}^{(t)}[k]}{\sqrt{\mathbf{v}_{j,r}^{(t)}[k]}} = \frac{\nabla_{\mathbf{w}_{j,r}} L(\mathbf{W}^{(t)})[k] \pm \widetilde{\Theta}(\eta)}{\Theta\Big(|\nabla_{\mathbf{w}_{j,r}} L(\mathbf{W}^{(t)})[k]|\Big) \pm \widetilde{\Theta}(\eta))}.$$

For $k \in \mathcal{B}_i$ we have

$$\frac{\mathbf{m}_{j,r}^{(t)}[k]}{\sqrt{\mathbf{v}_{j,r}^{(t)}[k]}} = \frac{\nabla_{\mathbf{w}_{j,r}} L(\mathbf{W}^{(t)})[k] \pm \widetilde{\Theta}\Big(\frac{\eta s\sigma_p|\ell_{j,i}^{(t)}|}{n}\Big) \pm \widetilde{\Theta}(\lambda\eta)}{\Theta\Big(|\nabla_{\mathbf{w}_{j,r}} L(\mathbf{W}^{(t)})[k]|\Big) \pm \widetilde{\Theta}\Big(\frac{\eta s\sigma_p|\ell_{j,i}^{(t)}|}{n}\Big) \pm \widetilde{\Theta}(\lambda\eta)}.$$

Then, we can conclude that for all $k = 1$ or $k \notin \mathcal{B}_i$ for any $i$, we have either $|\nabla_{\mathbf{w}_{j,r}} L(\mathbf{W}^{(t)})[k]| \le \widetilde{\Theta}(\eta)$ or

$$\frac{\mathbf{m}_{j,r}^{(t)}[k]}{\sqrt{\mathbf{v}_{j,r}^{(t)}[k]}} = \mathrm{sgn}\big(\nabla_{\mathbf{w}_{j,r}} L(\mathbf{W}^{(t)})[k]\big) \cdot \Theta(1).$$

For any $k \in \mathcal{B}_i$, we have either $|\nabla_{\mathbf{w}_{j,r}} L(\mathbf{W}^{(t)})[k]| \le \widetilde{\Theta}\big(\eta n^{-1} s\sigma_p|\ell_{j,i}^{(t)}| + \lambda\eta\big)$ or

$$\frac{\mathbf{m}_{j,r}^{(t)}[k]}{\sqrt{\mathbf{v}_{j,r}^{(t)}[k]}} = \mathrm{sgn}\big(\nabla_{\mathbf{w}_{j,r}} L(\mathbf{W}^{(t)})[k]\big) \cdot \Theta(1).$$

This completes the proof.

$\square$

**Lemma C.3** (Lemma 5.2, restated). Suppose the training data is generated according to Definition 3.1, assume $\lambda = o(\sigma_0^{q-2}\sigma_p/n)$ and $\eta = 1/\mathrm{poly}(d)$, then for any $t \le T_0$ with $T_0 = \widetilde{O}\big(\frac{1}{\eta s\sigma_p}\big)$ and any $i \in [n]$,

$$\langle \mathbf{w}_{j,r}^{(t+1)}, j \cdot \mathbf{v} \rangle \le \langle \mathbf{w}_{j,r}^{(t)}, j \cdot \mathbf{v} \rangle + \Theta(\eta),$$
$$\langle \mathbf{w}_{y_i,r}^{(t+1)}, \boldsymbol{\xi}_i \rangle = \langle \mathbf{w}_{y_i,r}^{(t)}, \boldsymbol{\xi}_i \rangle + \widetilde{\Theta}(\eta s\sigma_p).$$

*Proof.* At the initialization, we have

$$|\langle \mathbf{w}_{j,r}^{(0)}, \mathbf{v} \rangle| = \widetilde{\Theta}(\sigma_0), \quad |\langle \mathbf{w}_{j,r}^{(0)}, \boldsymbol{\xi}_i \rangle| = \widetilde{\Theta}(s^{1/2}\sigma_p\sigma_0 + \alpha) = \widetilde{\Theta}(s^{1/2}\sigma_p\sigma_0), \quad \mathbf{w}_{j,r}^{(0)}[k] = \widetilde{\Theta}(\sigma_0),$$

which also imply that $|\ell_{j,i}^{(0)}| = \Theta(1)$. Then recalling that $\lambda = o(\sigma_0^{q-2}\sigma_p/n)$, $\alpha = o(1)$, $s^{1/2}\sigma_p = \widetilde{O}(1)$, we have

$$\mathrm{sgn}\Bigg( \sum_{i=1}^{n} y_i \ell_{j,i}^{(0)} \sigma'(\langle \mathbf{w}_{j,r}^{(0)}, y_i \mathbf{v} \rangle) - \alpha \sum_{i=1}^{n} y_i \ell_{j,i}^{(0)} \sigma'(\langle \mathbf{w}_{j,r}^{(0)}, \boldsymbol{\xi}_i \rangle) - n\lambda \mathbf{w}_{j,r}^{(0)}[1] \Bigg)$$
$$= \mathrm{sgn}\big[ j \cdot \widetilde{\Theta}(n\sigma_0^{q-1}) - j \cdot \widetilde{\Theta}(\alpha n(s^{1/2}\sigma_p\sigma_0)^{q-1}) \pm o(\sigma_0^{q-1}\sigma_p) \big]$$
$$= \mathrm{sgn}(j).$$

Since $\mathbf{v}$ is 1-sparse, then by Lemma C.2, we have

$$\langle \mathbf{w}_{j,r}^{(1)}, j \cdot \mathbf{v} \rangle \leq \langle \mathbf{w}_{j,r}^{(0)}, j \cdot \mathbf{v} \rangle - \eta \left\langle \mathbf{m}_{j,r}^{(0)} / \sqrt{\mathbf{v}_{j,r}^{(0)}}, j \cdot \mathbf{v} \right\rangle \leq \langle \mathbf{w}_{j,r}^{(0)}, j \cdot \mathbf{v} \rangle + \Theta(\eta).$$

Now suppose that the first inequality holds for iterations $0, \ldots, t-1$. Then we have

$$\langle \mathbf{w}_{j,r}^{(t)}, j \cdot \mathbf{v} \rangle \leq \Theta(\eta \cdot T_0) \leq O(\sigma_0).$$

Besides, note that $\ell_{j,i}^{(t)} = \mathbb{1}_{j=y_i} - \text{logit}_j(F^{(t)}, \mathbf{x}_i)$, we have

$$\text{sgn}\big(y_i \ell_{j,i}^{(t)}\big) = \text{sgn}(j),$$

where we recall that $j \in \{-1, 1\}$. Therefore, given that $\lambda = o(\sigma_0^{q-2} \sigma_p / n)$, $\alpha = o(1)$, $s^{1/2}\sigma_p = \widetilde{O}(1)$, and assume $\ell_{j,i}^{(t)} = \Theta(1)$ (which will be verified later),

$$\text{sgn}\Bigg( \sum_{i=1}^n y_i \ell_{j,i}^{(t)} \sigma'(\langle \mathbf{w}_{j,r}^{(t)}, y_i \mathbf{v} \rangle) - \alpha \sum_{i=1}^n y_i \ell_{j,i}^{(t)} \sigma'(\langle \mathbf{w}_{j,r}^{(t)}, \boldsymbol{\xi}_i \rangle) - n\lambda \mathbf{w}_{j,r}^{(t)}[1] \Bigg)$$
$$= \text{sgn}\big[ j \cdot \widetilde{\Theta}(n\sigma_0^{q-1}) - j \cdot \widetilde{\Theta}(\alpha n(s^{1/2}\sigma_p\sigma_0)^{q-1}) \pm o\big(\sigma_0^{q-1}\sigma_p\big) \big]$$
$$= \text{sgn}(j).$$

Since $\mathbf{v}$ is 1-sparse, then by Lemma C.2, the following inequality naturally holds,

$$\langle \mathbf{w}_{j,r}^{(t+1)}, j \cdot \mathbf{v} \rangle \leq \langle \mathbf{w}_{j,r}^{(t)}, j \cdot \mathbf{v} \rangle - \eta \left\langle \mathbf{m}_{j,r}^{(t)} / \sqrt{\mathbf{v}_{j,r}^{(t)}}, j \cdot \mathbf{v} \right\rangle \leq \langle \mathbf{w}_{j,r}^{(t)}, j \cdot \mathbf{v} \rangle + \Theta(\eta).$$

Additionally, in terms of the memorization of noise, we first consider the iterate in the initialization. By the condition that $\eta = o(1/d) = o(1/(s\sigma_p))$ and note that for a sufficiently large fraction of $k \in \mathcal{B}_i$ (e.g., 0.99), we have $|\boldsymbol{\xi}_i[k]| \geq \widetilde{\Theta}(\sigma_p) \geq \widetilde{\Theta}(\eta n^{-1} s\sigma_p |\ell_{j,i}^{(0)}|)$ and thus

$$\text{sgn}\big(\nabla_{\mathbf{w}_{y_i,r}} L(\mathbf{W}^{(0)})[k]\big) = \text{sgn}\Big( \ell_{y_i,i}^{(0)} \sigma'(\langle \mathbf{w}_{y_i,r}^{(0)}, \boldsymbol{\xi}_i \rangle) \boldsymbol{\xi}_i[k] - n\lambda \mathbf{w}_{y_i,r}^{(0)}[k] \Big)$$
$$= -\text{sgn}\big[ \widetilde{\Theta}\big((d^{1/2}\sigma_p\sigma_0)^{q-1}\sigma_p \cdot \text{sgn}(\boldsymbol{\xi}_i[k])\big) \pm o(\sigma_0^{q-1}\sigma_p) \big] = -\text{sgn}(\boldsymbol{\xi}_i[k]).$$
$$\text{(C.8)}$$

Therefore, by Lemma C.2 we have the following according to (C.3),

$$\langle \mathbf{w}_{y_i,r}^{(1)}, \boldsymbol{\xi}_i \rangle = \langle \mathbf{w}_{y_i,r}^{(0)}, \boldsymbol{\xi}_i \rangle - \eta \left\langle \mathbf{m}_{j,r}^{(0)} / \sqrt{\mathbf{v}_{y_i,r}^{(0)}}, \boldsymbol{\xi}_i \right\rangle$$
$$\geq \langle \mathbf{w}_{y_i,r}^{(0)}, \boldsymbol{\xi}_i \rangle + \Theta(\eta) \cdot \sum_{k \in \mathcal{B}_i} \langle \text{sgn}(\boldsymbol{\xi}_i[k]), \boldsymbol{\xi}_i[k] \rangle - O(\eta s\sigma_p) - O(\eta\alpha)$$
$$= \langle \mathbf{w}_{y_i,r}^{(0)}, \boldsymbol{\xi}_i \rangle + \widetilde{\Theta}(\eta s\sigma_p),$$

where in the first inequality the term $O(\eta s\sigma_p)$ represents the coordinates that $|\boldsymbol{\xi}_i[k]| \leq O(\sigma_p)$ (so that we cannot use the sign information of $\nabla_{y_i,r} L(\mathbf{W}^{(0)})$ but directly bound it by $\Theta(1)$) and the last inequality is due to the fact that $|\mathcal{B}_i| \geq s - 1$ and $\alpha = o(1)$. For general $t$, we will consider the following induction hypothesis:

$$\langle \mathbf{w}_{y_i,r}^{(t+1)}, \boldsymbol{\xi}_i \rangle = \langle \mathbf{w}_{y_i,r}^{(t)}, \boldsymbol{\xi}_i \rangle + \widetilde{\Theta}(\eta s\sigma_p), \tag{C.9}$$

which has already been verified for $t = 0$. By Hypothesis (C.9), the following holds at time $t$,

$$\langle \mathbf{w}_{y_i,r}^{(t)}, \boldsymbol{\xi}_i \rangle = \langle \mathbf{w}_{y_i,r}^{(0)}, \boldsymbol{\xi}_i \rangle + \widetilde{\Theta}(t\eta s\sigma_p) = \widetilde{\Theta}(s^{1/2}\sigma_p\sigma_0 + t\eta s\sigma_p).$$

In the meanwhile, we have the following upper bound for $|\mathbf{w}_{j,r}^{(t)}[k]|$,

$$|\mathbf{w}_{j,r}^{(t)}[k]| \leq |\mathbf{w}_{j,r}^{(t-1)}[k]| + \eta |\text{sign}(\nabla_{\mathbf{w}_{j,r}} L(\mathbf{W}^{(t-1)}))| \leq |\mathbf{w}_{j,r}^{(0)}[k]| + t\eta = \widetilde{\Theta}(\sigma_0 + t\eta). \tag{C.10}$$

Besides, it is also easy to verify that for any $t \leq T_0 = \widetilde{\Theta}\left(\frac{1}{s\sigma_p \eta m}\right) = \widetilde{\Theta}\left(\frac{1}{s\sigma_p \eta}\right)$, we have $\langle \mathbf{w}_{y_i,r}^{(t)}, \boldsymbol{\xi}_i \rangle, \langle \mathbf{w}_{y_i,r}^{(t)}, j \cdot \mathbf{v} \rangle < \Theta(1/m)$ and thus $|\ell_{j,i}^{(t)}| = \Theta(1)$. Then similar to (C.8), we have

$$\text{sgn}\left(\nabla_{\mathbf{w}_{y_i,r}} L(\mathbf{W}^{(t)})[k]\right)$$
$$= \text{sgn}\left(\ell_{y_i,i}^{(t)} \sigma'(\langle \mathbf{w}_{y_i,r}^{(t)}, \boldsymbol{\xi}_i \rangle)\boldsymbol{\xi}_i[k] - n\lambda \mathbf{w}_{y_i,r}^{(t)}[k]\right)$$
$$= -\text{sgn}\left(\widetilde{\Theta}\left[(s^{1/2}\sigma_p\sigma_0 + t\eta s\sigma_p)^{q-1}\sigma_p \cdot \text{sgn}(\boldsymbol{\xi}_i[k]) \pm o\left(\sigma_0^{q-2}\sigma_p \cdot (\sigma_0 + t\eta)\right)\right]\right)$$
$$= -\text{sgn}(\boldsymbol{\xi}_i[k]). \tag{C.11}$$

This further implies that

$$\langle \mathbf{w}_{y_i,r}^{(t+1)}, \boldsymbol{\xi}_i \rangle \geq \langle \mathbf{w}_{y_i,r}^{(t)}, \boldsymbol{\xi}_i \rangle - \Theta(\eta) \cdot \sum_{k \in \mathcal{B}_i} \langle \text{sgn}(\nabla_{\mathbf{w}_{y_i,r}} L(\mathbf{W}^{(t)})[k]), \boldsymbol{\xi}_i[k] \rangle - O(\eta^2 s^2 \sigma_p^2) - O(\eta\alpha)$$
$$= \langle \mathbf{w}_{y_i,r}^{(t)}, \boldsymbol{\xi}_i \rangle + \widetilde{\Theta}(\eta s\sigma_p),$$

where the term $-O(\eta^2 s^2 \sigma_p^2)$ is contributed by the gradient coordinates that are smaller than $\Theta(\eta s\sigma_p)$. This verifies Hypothesis (C.9) at time $t$ and thus completes the proof. $\square$

From Lemma C.3, note that $s\sigma_p = \omega(1)$, then it can be seen that $\langle \mathbf{w}_{j,r}^{(t)}, j \cdot \mathbf{v} \rangle$ increases much faster than $\langle \mathbf{w}_{j,r}^{(t)}, j \cdot \mathbf{v} \rangle$. By looking at the update rule of $\langle \mathbf{w}_{j,r}^{(t)}, j \cdot \mathbf{v} \rangle$ (see (C.2)), it will keeps increasing only when, roughly speaking, $\sigma'(\langle \mathbf{w}_{j,r}^{(t)}, j \cdot \mathbf{v} \rangle) > \alpha\sigma'(\langle \mathbf{w}_{j,r}^{(t)}, \boldsymbol{\xi}_i \rangle)$. Since $\langle \mathbf{w}_{j,r}^{(t)}, \boldsymbol{\xi}_i \rangle$ increases much faster than $\langle \mathbf{w}_{j,r}^{(t)}, j \cdot \mathbf{v} \rangle$, it can be anticipated after a certain number of iterations, $\langle \mathbf{w}_{j,r}^{(t)}, j \cdot \mathbf{v} \rangle$ will start to decrease. In the following lemma, we provide an upper bound on the iteration number such that this decreasing occurs.

**Lemma C.4** (Lemma C.4, restated). Suppose the training data is generated according to Definition 3.1, $\alpha \geq \widetilde{\Theta}\left((s\sigma_p)^{1-q} \vee \sigma_0^{q-1}\right)$ and $\sigma_0 < \widetilde{O}\left((s\sigma_p)^{-1}\right)$, then for any $t \in [T_r, T_0]$ with $T_r = \widetilde{O}\left(\frac{\sigma_0}{\eta s\sigma_p \alpha^{1/(q-1)}}\right) \leq T_0$,

$$\langle \mathbf{w}_{j,r}^{(t+1)}, j \cdot \mathbf{v} \rangle = \langle \mathbf{w}_{j,r}^{(t)}, j \cdot \mathbf{v} \rangle - \Theta(\eta).$$

Moreover, it holds that

$$\mathbf{w}_{j,r}^{(T_0)}[k] = \begin{cases} -\text{sgn}(j) \cdot \widetilde{\Omega}\left(\frac{1}{s\sigma_p}\right), & k = 1, \\ \text{sgn}(\boldsymbol{\xi}_i[k]) \cdot \widetilde{\Omega}\left(\frac{1}{s\sigma_p}\right) \text{ or } \pm\widetilde{O}(\eta), & k \in \mathcal{B}_i, \text{ with } y_i = j, \\ \pm\widetilde{O}(\eta), & \text{otherwise.} \end{cases}$$

*Proof.* Recall from Lemma C.3 that for any $t \leq T_0$ we have

$$\langle \mathbf{w}_{j,r}^{(t+1)}, j \cdot \mathbf{v} \rangle \leq \langle \mathbf{w}_{j,r}^{(t)}, j \cdot \mathbf{v} \rangle + \Theta(\eta) \leq \langle \mathbf{w}_{j,r}^{(0)}, j \cdot \mathbf{v} \rangle + \Theta(t\eta),$$
$$\langle \mathbf{w}_{y_s,r}^{(t+1)}, \boldsymbol{\xi}_s \rangle = \langle \mathbf{w}_{y_s,r}^{(t)}, \boldsymbol{\xi}_s \rangle + \widetilde{\Theta}(\eta s\sigma_p) \leq \langle \mathbf{w}_{y_s,r}^{(0)}, \boldsymbol{\xi}_s \rangle + \widetilde{\Theta}(t\eta s\sigma_p).$$

Besides, by Lemma C.2 we also have $|\mathbf{w}_{j,r}^{(t)}[k]| \leq |\mathbf{w}_{j,r}^{(0)}[k]| + O(t\eta)$. Then it can be verified that for some $T_r = \widetilde{O}\left(\frac{\sigma_0}{\eta s\sigma_p \alpha^{1/(q-1)}}\right)$, we have for all $i \in [n]$ and $t \in [T_r, T_0]$

$$\alpha\sigma'(\langle \mathbf{w}_{y_i,r}^{(t)}, \boldsymbol{\xi}_i \rangle) \geq C \cdot \left[\sigma'(\langle \mathbf{w}_{j,r}^{(t)}, j \cdot \mathbf{v} \rangle) + \lambda n|\mathbf{w}_{j,r}^{(t)}[1]|\right]$$

for some constant $C$. This further implies that

$$\text{sgn}\left(\nabla_{\mathbf{w}_{j,r}} L(\mathbf{W}^{(t)})[1]\right)$$
$$= -\text{sgn}\left(\sum_{i=1}^{n} y_i \ell_{j,i}^{(t)} \sigma'(\langle \mathbf{w}_{j,r}^{(t)}, y_i \mathbf{v} \rangle) - \alpha \sum_{i=1}^{n} y_i \ell_{j,i}^{(t)} \sigma'(\langle \mathbf{w}_{j,r}^{(t)}, \boldsymbol{\xi}_i \rangle) - n\lambda \mathbf{w}_{j,r}^{(t)}[1]\right)$$
$$= -\text{sgn}\left[-\alpha \sum_{i=1}^{n} y_i \ell_{j,i}^{(t)} \sigma'(\langle \mathbf{w}_{j,r}^{(t)}, \boldsymbol{\xi}_i \rangle)\right]$$
$$= \text{sgn}(j),$$

where we use the fact that $\text{sgn}(y_i \ell_{j,i}^{(t)}) = \text{sgn}(j)$ for all $i \in [n]$. Then by Lemma C.2 and (C.2), we have for all $t \in [T_r, T_0]$,

$$\langle \mathbf{w}_{j,r}^{(t+1)}, j \cdot \mathbf{v} \rangle = \langle \mathbf{w}_{j,r}^{(t)}, j \cdot \mathbf{v} \rangle - \Theta(\eta) \cdot \text{sgn}(j) \cdot \text{sgn}\big( \nabla_{\mathbf{w}_{j,r}} L(\mathbf{W}^{(t)})[1] \big) = \langle \mathbf{w}_{j,r}^{(t)}, j \cdot \mathbf{v} \rangle - \Theta(\eta).$$

Then at iteration $T_0$, for the first coordinate we have

$$\mathbf{w}_{j,r}^{(T_0)}[1] = \mathbf{w}_{j,r}^{(0)}[1] + \text{sgn}(j) \cdot \Theta(T_r \eta) - \text{sgn}(j) \cdot \Theta((T_0 - T_r)\eta) \geq -\text{sgn}(j) \cdot \widetilde{\Omega}\bigg( \frac{1}{s\sigma_p} \bigg)$$

For any $k \in \mathcal{B}_i$ with $y_i = j$, we have either the coordinate will increase at a rate of $\Theta(1)$ or fall into 0. As a consequence we have either $\mathbf{w}_{j,r}^{(T_0)}[k] \in [-\widetilde{\Theta}(\eta), \widetilde{\Theta}(\eta)]$ or

$$\mathbf{w}_{j,r}^{(T_0)}[k] = \mathbf{w}_{j,r}^{(0)}[k] + \text{sgn}(\boldsymbol{\xi}_i[k]) \cdot \Theta(T_0 \eta) \geq \text{sgn}(\boldsymbol{\xi}_i[k]) \cdot \widetilde{\Omega}\bigg( \frac{1}{s\sigma_p} \bigg).$$

For the remaining coordinate, its update will be determined by the regularization term, which will finally fall into the region around zero since we have $T_0 \eta = \omega(\sigma_0)$. By Lemma C.2 it is clear that $\mathbf{w}_{j,r}^{(T_0)}[k] \in [-\widetilde{\Theta}(\eta), \widetilde{\Theta}(\eta)]$. □

**Lemma C.5** (Lemma 5.4, restated). If $\alpha = O\big( \frac{s\sigma_p^2}{n} \big)$ and $\eta = o(\lambda)$, then let $r^* = \arg\max_{r \in [m]} \langle \mathbf{w}_{y_i, r}^{(t)}, \boldsymbol{\xi}_i \rangle$, for any $t \geq T_0$, $i \in [n]$, $j \in [2]$ and $r \in [m]$, it holds that

$$\langle \mathbf{w}_{y_i, r^*}^{(t)}, \boldsymbol{\xi}_i \rangle = \widetilde{\Theta}(1), \quad \sum_{k \in \mathcal{B}_i} |\mathbf{w}_{y_i, r^*}^{(t)}[k]| \cdot |\boldsymbol{\xi}_i[k]| = \widetilde{\Theta}(1),$$

$$\forall r \in [m], \quad \langle \mathbf{w}_{j,r}^{(t)}, \text{sgn}(j) \cdot \mathbf{v} \rangle \in [-\widetilde{O}\big( \frac{n\alpha}{s\sigma_p^2} \big), O(\lambda^{-1}\eta)].$$

*Proof.* The proof will be relying on the following three induction hypothesis:

$$\langle \mathbf{w}_{y_i, r^*}^{(t)}, \boldsymbol{\xi}_i \rangle = \widetilde{\Omega}(1), \tag{C.12}$$

$$\sum_{k \in \mathcal{B}_i} |\mathbf{w}_{y_i, r^*}^{(t+1)}[k]| \cdot |\boldsymbol{\xi}_i[k]| = \widetilde{\Theta}(1), \tag{C.13}$$

$$\forall r \in [m], \langle \mathbf{w}_{j,r}^{(t)}, \text{sgn}(j) \cdot \mathbf{v} \rangle \in \Big[ -\widetilde{O}\big( \frac{n\alpha}{s\sigma_p^2} \big), O(\lambda^{-1}\eta) \Big], \tag{C.14}$$

which we assume they hold for all $\tau \leq t$ and $r \in [m]$, $i \in [n]$, and $j \in [2]$. It is clear that all hypothesis hold when $t = T_0$ according to Lemma C.4.

**Verifying Hypothesis (C.12).** We first verify Hypothesis (C.12). Recall that the update rule for $\langle \mathbf{w}_{y_i, r}^{(t)}, \boldsymbol{\xi}_i \rangle$ is given as follows,

$$\langle \mathbf{w}_{y_i, r}^{(t+1)}, \boldsymbol{\xi}_i \rangle$$

$$= \langle \mathbf{w}_{y_i, r}^{(t)}, \boldsymbol{\xi}_i \rangle - \eta \cdot \langle \mathbf{m}_{y_i, r}^{(t)} / \sqrt{\mathbf{v}_{y_i, r}^{(t)}}, \boldsymbol{\xi}_i \rangle$$

$$\geq \langle \mathbf{w}_{y_i, r}^{(t)}, \boldsymbol{\xi}_i \rangle - \Theta(\eta) \cdot \langle \text{sgn}\big( \nabla_{\mathbf{w}_{y_i, r}} L(\mathbf{W}^{(t)}) \big), \boldsymbol{\xi}_i \rangle - \widetilde{\Theta}(\eta^2 s^2 \sigma_p^2)$$

$$= \langle \mathbf{w}_{y_i, r}^{(t)}, \boldsymbol{\xi}_i \rangle + \Theta(\eta) \cdot \sum_{k \in \mathcal{B}_i} \Big\langle \text{sgn}\Big( \ell_{y_i, i}^{(t)} \sigma'(\langle \mathbf{w}_{y_i, r}^{(t)}, \boldsymbol{\xi}_i \rangle) \boldsymbol{\xi}_i[k] - n\lambda \mathbf{w}_{y_i, r}^{(t)}[k] \Big), \boldsymbol{\xi}_i[k] \Big\rangle$$

$$- \alpha y_i \Theta(\eta) \cdot \text{sgn}\Big( \sum_{i=1}^{n} y_i \ell_{j,i}^{(t)} \sigma'(\langle \mathbf{w}_{j,r}^{(t)}, y_i \mathbf{v} \rangle) - \alpha \sum_{i=1}^{n} y_i \ell_{j,i}^{(t)} \sigma'(\langle \mathbf{w}_{j,r}^{(t)}, \boldsymbol{\xi}_i \rangle) - n\lambda \mathbf{w}_{j,r}^{(t)}[1] \Big)$$

$$- \widetilde{\Theta}(\eta^2 s^2 \sigma_p^2). \tag{C.15}$$

Note that for any $a$ and $b$ we have $\text{sgn}(a - b) \cdot a \geq |a| - 2|b|$. Then it follows that

$$\sum_{k \in \mathcal{B}_i} \Big\langle \text{sgn}\Big( \ell_{y_i, i}^{(t)} \sigma'(\langle \mathbf{w}_{y_i, r}^{(t)}, \boldsymbol{\xi}_i \rangle) \boldsymbol{\xi}_i[k] - n\lambda \mathbf{w}_{y_i, r}^{(t)}[k] \Big), \boldsymbol{\xi}_i[k] \Big\rangle \geq \sum_{k \in \mathcal{B}_i} \bigg( |\boldsymbol{\xi}_i[k]| - \frac{2n\lambda |\mathbf{w}_{y_i, r}^{(t)}[k]|}{\ell_{y_i, i}^{(t)} \sigma'(\langle \mathbf{w}_{y_i}^{(t)}, \boldsymbol{\xi}_i \rangle)} \bigg)$$

$$\geq \widetilde{\Theta}(s\sigma_p) - \widetilde{\Theta}\bigg( \frac{n\lambda}{\ell_{y_i, i}^{(t)} \sigma_p} \bigg),$$

where the last inequality follows from Hypothesis (C.12) and (C.13). Further recall that $\lambda = o(\sigma_0^{q-2}\sigma_p/n)$, plugging the above inequality to (C.15) gives

$$
\langle \mathbf{w}_{y_i,r}^{(t+1)}, \boldsymbol{\xi}_i \rangle \geq \langle \mathbf{w}_{y_i,r}^{(t)}, \boldsymbol{\xi}_i \rangle + \widetilde{\Theta}(\eta s\sigma_p) - \widetilde{\Theta}\left( \frac{\eta n\lambda}{\ell_{y_i,i}^{(t)}\sigma_p} \right) - \widetilde{\Theta}(\eta^2 s^2 \sigma_p^2)
$$

$$
\geq \langle \mathbf{w}_{y_i,r}^{(t)}, \boldsymbol{\xi}_i \rangle + \widetilde{\Theta}(\eta s\sigma_p) - \Theta(\alpha\eta) - \widetilde{\Theta}\left( \frac{\eta \sigma_0^{q-2}}{\ell_{y_i,i}^{(t)}} \right). \tag{C.16}
$$

Then it is clear that $\langle \mathbf{w}_{y_i,r}^{(t)}, \boldsymbol{\xi}_i \rangle$ will increase by $\widetilde{\Theta}(\eta s\sigma_p)$ if $\ell_{y_i,i}^{(t)}$ is larger than some constant of order $\widetilde{\Omega}(\frac{n\lambda}{s\sigma_p^2}) = \widetilde{\Omega}(\frac{\sigma_0^{q-2}}{s\sigma_p})$. We will first show that as soon as there is a iterate $\mathbf{W}^{(\tau)}$ satisfying $\ell_{y_i,i}^{(\tau)} \leq \widetilde{O}(\frac{n\lambda}{s\sigma_p^2})$ for some $\tau \leq t$, then it must hold that $\ell_{y_i,i}^{(\tau')}$ will also be smaller than some constant in the order of $\widetilde{O}(\frac{n\lambda}{s\sigma_p^2})$ for all $\tau' \in [\tau, t+1]$. To prove this, we first note that if $\ell_{y_i,i}^{(t)}$ reaches some constant in the order of $\widetilde{O}(\frac{n\lambda}{s\sigma_p^2})$, we have for all $r \in [m]$ by (C.16)

$$
\langle \mathbf{w}_{y_i,r}^{(t+1)}, \boldsymbol{\xi}_i \rangle \geq \langle \mathbf{w}_{y_i,r}^{(t)}, \boldsymbol{\xi}_i \rangle + \widetilde{\Theta}(\eta s\sigma_p),
$$
$$
\langle \mathbf{w}_{-y_i,r}^{(t+1)}, \boldsymbol{\xi}_i \rangle \leq \langle \mathbf{w}_{-y_i,r}^{(t)}, \boldsymbol{\xi}_i \rangle + O(\alpha\eta),
$$
$$
|\langle \mathbf{w}_{j,r}^{(t+1)}, \mathbf{v} \rangle| \leq |\langle \mathbf{w}_{j,r}^{(t)}, \mathbf{v} \rangle| + O(\eta). \tag{C.17}
$$

Therefore, we have

$$
\ell_{y_i,i}^{(t+1)} = \frac{e^{F_{-y_i}(\mathbf{W}^{(t+1)}, \mathbf{x}_i)}}{\sum_{j \in \{-1,1\}} e^{F_j(\mathbf{W}^{(t+1)}, \mathbf{x}_i)}}
$$

$$
= \frac{1}{1 + \exp\left[ \sum_{r=1}^m \left[ \sigma(\langle \mathbf{w}_{y_i,r}^{(t+1)}, \mathbf{v} \rangle) + \sigma(\langle \mathbf{w}_{y_i,r}^{(t+1)}, \boldsymbol{\xi}_i \rangle) - \sigma(\langle \mathbf{w}_{-y_i,r}^{(t+1)}, \mathbf{v} \rangle) - \sigma(\langle \mathbf{w}_{-y_i,r}^{(t+1)}, \boldsymbol{\xi}_i \rangle) \right] \right]}
$$

$$
\leq \frac{1}{1 + \exp\left[ \sum_{r=1}^m \left[ \sigma(\langle \mathbf{w}_{y_i,r}^{(t)}, \mathbf{v} \rangle) + \sigma(\langle \mathbf{w}_{y_i,r}^{(t)}, \boldsymbol{\xi}_i \rangle) - \sigma(\langle \mathbf{w}_{-y_i,r}^{(t)}, \mathbf{v} \rangle) - \sigma(\langle \mathbf{w}_{-y_i,r}^{(t)}, \boldsymbol{\xi}_i \rangle) \right] + \widetilde{\Theta}(\eta s\sigma_p^2) \right]}
$$

$$
\leq \frac{1}{1 + \exp\left[ \sum_{r=1}^m \left[ \sigma(\langle \mathbf{w}_{y_i,r}^{(t)}, \mathbf{v} \rangle) + \sigma(\langle \mathbf{w}_{y_i,r}^{(t)}, \boldsymbol{\xi}_i \rangle) \right] - \sigma(\langle \mathbf{w}_{-y_i,r}^{(t)}, \mathbf{v} \rangle) - \sigma(\langle \mathbf{w}_{-y_i,r}^{(t)}, \boldsymbol{\xi}_i \rangle) \right]\right]}
$$

$$
= \ell_{y_i,i}^{(t)},
$$

where inequality follows from (C.17). Therefore, this implies that as long as $\ell_{y_i,i}^{(t)}$ is larger than some constant $b = \widetilde{O}(\frac{n\lambda}{s\sigma_p^2})$, then the adam algorithm will prevent it from further increasing. Besides, since $m\eta\sigma_p^2 = o(1)$, then we must have $\ell_{y_i,i}^{(t+1)} \in [0.5\ell_{y_i,i}^{(t)}, 2\ell_{y_i,i}^{(t)}]$. As a consequence, we can deduce that $\ell_{y_i,i}^{(t)}$ cannot be larger than $2b$, since otherwise there must exists a iterate $\mathbf{W}^{(\tau)}$ with $\tau \leq t$ such that $\ell_{y_i,i}^{(\tau)} \in [b, 2b]$ and $\ell_{y_i,i}^{(\tau+1)} \geq \ell_{y_i,i}^{(\tau)}$, which contradicts the fact that $\ell_{y_i,i}^{(\tau)}$ should decreases if $\ell_{y_i,i}^{(\tau)} \geq b$. Therefore, we can claim that if $\ell_{y_i,i}^{(\tau)} \leq b = \widetilde{O}(\frac{n\lambda}{s\sigma_p^2})$ for some $\tau \leq t$, then we have

$$
\ell_{y_i,i}^{(\tau')} \leq \widetilde{O}\left( \frac{n\lambda}{s\sigma_p^2} \right) \tag{C.18}
$$

for all $\tau' \in [\tau, t+1]$. Then further note that

$$
2\ell_{y_i,i}^{(t+1)} \geq \ell_{y_i,i}^{(t)} = \frac{e^{F_{-y_i}(\mathbf{W}^{(t)}, \mathbf{x}_i)}}{\sum_{j \in \{-1,1\}} e^{F_j(\mathbf{W}^{(t)}, \mathbf{x}_i)}}
$$

$$
\geq \exp\left( -\sum_{r=1}^m \left[ \sigma(\langle \mathbf{w}_{y_i,r}^{(t)}, y_i\mathbf{v} \rangle) + \sigma(\langle \mathbf{w}_{y_i,r}^{(t)}, \boldsymbol{\xi}_i \rangle) \right] \right)
$$

$$
\geq \exp\left( -\Theta\left( m \max_{r \in [m]} \sigma(\langle \mathbf{w}_{y_i,r}^{(t)}, \boldsymbol{\xi}_i \rangle) \right) \right), \tag{C.19}
$$

where in the last inequality we use Hypothesis (C.14). Then by the fact that $\ell_{y_i,i}^{(t+1)} \leq \widetilde{O}\left(\frac{n\lambda}{s\sigma_p^2}\right) = o(1)$ and $m = \widetilde{\Theta}(1)$, it is clear that $\exp\left(-\Theta\left(m \max_{r \in [m]} \sigma(\langle \mathbf{w}_{y_i,r}^{(t+1)}, \boldsymbol{\xi}_i \rangle)\right)\right) = o(1)$ so that $\max_{r \in [m]} \langle \mathbf{w}_{y_i,r}^{(t+1)}, \boldsymbol{\xi}_i \rangle = \widetilde{\Omega}(1)$. This verifies Hypothesis (C.12).

**Verifying Hypothesis (C.13).** Now we will verify Hypothesis (C.13). First, note that we have already shown that $\langle \mathbf{w}_{y_i,r^*}^{(t+1)}, \boldsymbol{\xi}_i \rangle = \widetilde{\Omega}(1)$ so it holds that

$$\sum_{k \in \mathcal{B}_i} |\mathbf{w}_{y_i,r^*}^{(t+1)}[k]| \cdot |\boldsymbol{\xi}_i[k]| + \alpha |\mathbf{w}_{y_i,r^*}^{(t+1)}[1]| \geq \langle \mathbf{w}_{y_i,r^*}^{(t+1)}, \boldsymbol{\xi}_i \rangle = \widetilde{\Omega}(1).$$

By Hypothesis (C.14), we have $|\mathbf{w}_{y_i,r^*}^{(t+1)}[1]| \leq |\mathbf{w}_{y_i,r^*}^{(t)}[1]| + \eta = o(1)$. Besides, since each coordinate in $\boldsymbol{\xi}_i$ is a Gaussian random variable, then $\max_{k \in \mathcal{B}_i} |\boldsymbol{\xi}_i[k]| = \widetilde{O}(\sigma_p)$. This immediately implies that

$$\sum_{k \in \mathcal{B}_i} |\mathbf{w}_{y_i,r^*}^{(t+1)}[k]| \cdot |\boldsymbol{\xi}_i[k]| = \widetilde{\Omega}(1).$$

Then we will prove the upper bound of $\sum_{k \in \mathcal{B}_i} |\mathbf{w}_{y_i,r}^{(t+1)}[k]| \cdot |\boldsymbol{\xi}_i[k]|$. Recall that by Lemma C.2, for any $k \in \mathcal{B}_i$ such that $\nabla_{\mathbf{w}_{y_i,r}} L(\mathbf{W}^{(t)})[k] \geq \widetilde{\Theta}(n^{-1}\eta s \sigma_p \ell_{y_i,i}^{(t)})$, we have

$$\mathbf{w}_{y_i,r}^{(t+1)}[k] = \mathbf{w}_{y_i,r}^{(t)}[k] + \Theta(\eta) \cdot \mathrm{sgn}\left(\ell_{y_i,i}^{(t)} \sigma'(\langle \mathbf{w}_{y_i,r}^{(t)}, \boldsymbol{\xi}_i \rangle) \boldsymbol{\xi}_i[k] - n\lambda \mathbf{w}_{y_i,r}^{(t)}[k]\right).$$

Note that by Lemma C.4, for every $k \in \mathcal{B}_i$, we have either $\mathbf{w}_{y_i,r}^{(T_0)}[k] = \mathrm{sgn}(\boldsymbol{\xi}_i[k]) \cdot \widetilde{\Theta}\left(\frac{1}{s\sigma_p}\right)$ or $|\mathbf{w}_{y_i,r}^{(T_0)}[k]| \leq \eta$. Then during the training process after $T_0$, we have either $\mathrm{sgn}(\mathbf{w}_{y_i,r}^{(t)}[k]) = \mathrm{sgn}(\boldsymbol{\xi}_i[k])$ or $\mathrm{sgn}(\boldsymbol{\xi}_i[k]) \cdot \mathbf{w}_{y_i,r}^{(t)} \geq -\widetilde{O}(\eta)$ since if for some iteration number $t'$ that we have $\mathrm{sgn}(\mathbf{w}_{y_i,r}^{(t')}[k]) = -\mathrm{sgn}(\boldsymbol{\xi}_i[k])$ but $\mathrm{sgn}(\mathbf{w}_{y_i,r}^{(t'-1)}[k]) = \mathrm{sgn}(\boldsymbol{\xi}_i[k])$, then after $\bar{\tau} = \widetilde{O}(1)$ steps (see the proof of Lemma C.2 for the definition of $\bar{\tau}$) in the constant number of steps the gradient will must be in the same direction of $\boldsymbol{\xi}_i[k]$, which will push $\mathbf{w}_{y_i,r}[k]$ back to zero or become positive along the direction of $\boldsymbol{\xi}_i[k]$. Therefore, based on this property we have the following regarding the inner product $\langle \mathbf{w}_{y_i,r}^{(t)}, \boldsymbol{\xi}_i \rangle$,

$$\begin{aligned}
\langle \mathbf{w}_{y_i,r}^{(t)}, \boldsymbol{\xi}_i \rangle &= \sum_{k \in \mathcal{B}_i \cup \{1\}} \mathbf{w}_{y_i,r}^{(t)}[k] \cdot \boldsymbol{\xi}_i[k] \\
&\geq \sum_{k \in \mathcal{B}_i \cup \{1\}} |\mathbf{w}_{y_i,r}^{(t)}[k]| \cdot |\boldsymbol{\xi}_i[k]| - \widetilde{O}(\eta) \cdot \sum_{k \in \mathcal{B}_i \cup \{1\}} |\boldsymbol{\xi}_i[k]| \\
&= \sum_{k \in \mathcal{B}_i \cup \{1\}} |\mathbf{w}_{y_i,r}^{(t)}[k]| \cdot |\boldsymbol{\xi}_i[k]| - \widetilde{O}(\eta s \sigma_p),
\end{aligned}$$

where the second inequality follows from the fact that the entry $\mathbf{w}_{y_i,r}^{(t)}[k]$ that has different sign of $\boldsymbol{\xi}_i[k]$ satisfies $|\mathbf{w}_{y_i,r}^{(t)}[k]| \leq \widetilde{O}(\eta)$. Then let $B_i^{(t)} = \sum_{j \in \mathcal{B}_i \cup \{1\}} |\mathbf{w}_{y_i,r}^{(t)}[k] \cdot \mathbb{1}(|\mathbf{w}_{y_i,r}^{(t)}[k]| \geq \widetilde{O}(\eta))| \cdot |\boldsymbol{\xi}_i[k]|$, which satisfies $B_i^{(T_0)} = \widetilde{\Theta}(1)$ by Lemma C.4. Then assume $B_i^{(t)}$ keeps increasing and reaches some value in the order of $\Theta\left(\log(dn\eta^{-1})\right)$, it holds that according to the inequality above

$$\langle \mathbf{w}_{y_i,r}^{(t)}, \boldsymbol{\xi}_i \rangle = \Theta\left(\log(dn\eta^{-1})\right) - \widetilde{\Theta}(\eta s \sigma_p) = \Theta\left(\log(dn\eta^{-1})\right),$$

where we use the condition that $\eta = O\left((s\sigma_p)^{-1}\right)$. Then by Hypothesis (C.12) and (C.14) we know that $|\langle \mathbf{w}_{j,r}^{(t)}, \mathbf{v} \rangle| = o(1)$, $\langle \mathbf{w}_{y_i,r^*}^{(t)}, \boldsymbol{\xi}_i \rangle = \widetilde{\Omega}(1)$, and $|\langle \mathbf{w}_{-y_i,r^*}^{(t)}, \boldsymbol{\xi}_i \rangle| = \widetilde{O}(d\eta) + \alpha |\langle \mathbf{w}_{-y_i,r^*}^{(t)}, \mathbf{v} \rangle| = o(1)$ then similar to (C.19), it holds that

$$\ell_{y_i,i}^{(t)} = \frac{e^{F_{-y_i}(\mathbf{W}^{(t)}, \mathbf{x}_i)}}{\sum_{j \in \{-1,1\}} e^{F_j(\mathbf{W}^{(t)}, \mathbf{x}_i)}} \leq \exp\left(-\Theta\left(\sigma(\langle \mathbf{w}_{y_i,r^*}^{(t)}, \boldsymbol{\xi}_i \rangle)\right)\right) \leq \mathrm{poly}(d^{-1}, n^{-1}, \eta).$$

Therefore, at this time we have for all $k \in \mathcal{B}_i$,

$$\ell_{y_i,i}^{(t)} \sigma'(\langle \mathbf{w}_{y_i,r}^{(t)}, \boldsymbol{\xi}_i \rangle) \boldsymbol{\xi}_i[k] \leq \mathrm{poly}(d^{-1}, n^{-1}, \eta) \cdot \Theta\left(\log^{q-1}(dn\eta^{-1})\right) \cdot \widetilde{\Theta}(\sigma_p) \leq n\lambda\eta.$$

Then for all $|\mathbf{w}_{y_i,r}^{(t)}[k]| \geq \widetilde{O}(\eta)$, the sign of the gradient satisfies

$$\text{sgn}\big(\nabla_{\mathbf{w}_{y_i,r}}L(\mathbf{W}^{(t)})[k]\big) = -\text{sgn}\bigg(\ell_{y_i,i}^{(t)}\sigma'(\langle\mathbf{w}_{y_i,r}^{(t)},\boldsymbol{\xi}_i\rangle)\boldsymbol{\xi}_i[k] - n\lambda\mathbf{w}_{y_i,r}^{(t)}[k]\bigg)$$
$$= \text{sgn}(n\lambda\eta - \mathbf{w}_{y_i,r}^{(t)}[k])$$
$$= \text{sgn}(\mathbf{w}_{y_i,r}^{(t)}[k]).$$

Then note that $|\nabla_{\mathbf{w}_{y_i,r}}L(\mathbf{W}^{(t)})[k]| = \Theta(|\lambda\mathbf{w}_{y_i,r}^{(t)}[k]|) \geq \Theta\big(n^{-1}\eta s\sigma_p\ell_{y_i,i}^{(t)} + \lambda\eta\big)$, by the update rule of $\mathbf{w}_{y_i,r}^{(t)}[k]$ and Lemma C.2, we know the sign gradient will dominate the update process. Then we have $|\mathbf{w}_{y_i,r}^{(t+1)}[k]| = |\mathbf{w}_{y_i,r}^{(t)}[k] - \Theta(\eta) \cdot \text{sgn}(\mathbf{w}_{y_i,r}^{(t)}[k])| \leq |\mathbf{w}_{y_i,r}^{(t)}[k]|$, which implies that $\big|\mathbf{w}_{y_i,r}^{(t)}[k] \cdot \mathbb{1}(|\mathbf{w}_{y_i,r}^{(t)}[k]| \geq \widetilde{O}(\eta))\big|$ decreases so that $B_i^{(t)}$ also decreases. Therefore, we can conclude that $B_i^{(t)}$ will not exceed $\Theta\big(\log(dn\eta^{-1})\big)$. Then combining the results for all $i \in [n]$ gives

$$\sum_{k\in\mathcal{B}_i}|\mathbf{w}_{y_i,r^*}^{(t)}[k]| \cdot |\boldsymbol{\xi}_i[k]| \leq B_i^{(t)} + \widetilde{O}(s\eta\sigma_p) \leq \Theta\big(\log(dn\eta^{-1})\big) + O(1) = \widetilde{\Theta}(1),$$

where in the first inequality we again use the condition that $\eta = o(1/d) = o\big((s\sigma_p)^{-1}\big)$. This verifies Hypothesis (C.13). Notably, this also implies that $\langle\mathbf{w}_{y_i,r^*}^{(t)},\boldsymbol{\xi}_i\rangle = \max_{r\in[m]}\langle\mathbf{w}_{y_i,r}^{(t)},\boldsymbol{\xi}_i\rangle \leq \widetilde{\Theta}(1)$.

**Verifying Hypothesis (C.14).** In order to verify Hypothesis (C.14), let us first recall the update rule of $\langle\mathbf{w}_{j,r}^{(t)},\mathbf{v}\rangle$:

$$\langle\mathbf{w}_{j,r}^{(t+1)},\mathbf{v}\rangle = \langle\mathbf{w}_{j,r}^{(t)},\mathbf{v}\rangle - \eta\bigg\langle\frac{\mathbf{m}_{j,r}^{(t)}}{\sqrt{\mathbf{v}_{j,r}^{(t)}}},\mathbf{v}\bigg\rangle.$$

Then by Lemma C.2, we know that if $|\nabla_{\mathbf{w}_{j,r}}L(\mathbf{W}^{(t)})[1]| \leq \widetilde{\Theta}(\eta)$, then $|\mathbf{m}_{j,r}^{(t)}/\sqrt{\mathbf{v}_{j,r}^{(t)}}| \leq \Theta(1)$ and otherwise

$$\bigg\langle\frac{\mathbf{m}_{j,r}^{(t)}}{\sqrt{\mathbf{v}_{j,r}^{(t)}}},\mathbf{v}\bigg\rangle = -\text{sgn}\bigg(\sum_{i=1}^{n}y_i\ell_{j,i}^{(t)}\sigma'(\langle\mathbf{w}_{j,r}^{(t)},y_i\mathbf{v}\rangle) - \alpha\sum_{i=1}^{n}y_i\ell_{j,i}^{(t)}\sigma'(\langle\mathbf{w}_{j,r}^{(t)},\boldsymbol{\xi}_i\rangle) - n\lambda\mathbf{w}_{j,r}^{(t)}[1]\bigg) \cdot \Theta(1).$$

Without loss of generality we assume $j = 1$, then by Lemma C.4 we know that $\mathbf{w}_{1,r}^{(T_0)}[1] = -\widetilde{\Omega}\big(\frac{1}{s\sigma_p}\big)$. In the remaining proof, we will show that either $\mathbf{w}_{1,r}^{(t+1)}[1] \in [0,\widetilde{\Theta}(\lambda^{-1}\eta)]$ or $\mathbf{w}_{1,r}^{(t+1)}[1] \in \big[-\widetilde{O}\big(\frac{n\alpha}{s\sigma_p^2}\big),0\big)$.

First we will show that $\mathbf{w}_{1,r}^{(t+1)}[1] \in [0,\widetilde{\Theta}(\lambda^{-1}\eta)]$ for all $r$. Note that in the beginning of this stage, we have $\mathbf{w}_{1,r}^{(T_0)}[1] < 0$. In order to make the sign of $\mathbf{w}_{1,r}^{(t)}[1]$ flip, we must have, in some iteration $t' \leq t$ that satisfies $\mathbf{w}_{1,r}^{(t')}[1] \in [0,\widetilde{\Theta}(\lambda^{-1}\eta)]$, therefore

$$-n\nabla_{\mathbf{w}_{1,r}}L(\mathbf{W}^{(t')})[1] = \sum_{i=1}^{n}y_i\ell_{j,i}^{(t')}\sigma'(\langle\mathbf{w}_{j,r}^{(t')},y_i\mathbf{v}\rangle) - \alpha\sum_{i=1}^{n}y_i\ell_{j,i}^{(t')}\sigma'(\langle\mathbf{w}_{j,r}^{(t')},\boldsymbol{\xi}_i\rangle) - n\lambda\mathbf{w}_{j,r}^{(t')}[1]$$
$$\leq n\big[(\mathbf{w}_{j,r}^{(t')}[1])^{q-2} - \lambda\big] \cdot \mathbf{w}_{j,r}^{(t')}[1] \leq -\widetilde{\Theta}(n\eta) \leq 0,$$

where the second inequality holds since $\eta = o(\lambda^{(q-1)/(q-2)})$. Note that $|\nabla_{\mathbf{w}_{1,r}}L(\mathbf{W}^{(t')})[1]| \geq \widetilde{\Theta}(\eta)$, then by Lemma C.2 we know that Adam is similar to sign gradient descent and thus $\mathbf{w}_{1,r}^{(t'+1)}[1] = \mathbf{w}_{1,r}^{(t')}[1] - \Theta(\eta)$ which starts to decrease. This implies that if $\mathbf{w}_{1,r}^{(t+1)}[1]$ is positive, then it cannot exceed $\widetilde{\Theta}(\lambda^{-1}\eta) = o(1)$.

Then we can prove that if $\mathbf{w}_{1,r}^{(t+1)}[1]$ is negative, then $|\mathbf{w}_{1,r}^{(t+1)}[1]| = \widetilde{O}\left(\frac{n\alpha}{s\sigma_p^2}\right)$. In this case we have for all $t' \leq t$,

$$
\begin{aligned}
-n\nabla_{\mathbf{w}_{1,r}^{(t)}} L(\mathbf{W}^{(t')})[1] &= \sum_{i=1}^{n} y_i \ell_{1,i}^{(t')} \sigma'(\langle \mathbf{w}_{1,r}^{(t')}, y_i\mathbf{v}\rangle) - \alpha \sum_{i=1}^{n} y_i \ell_{1,i}^{(t')} \sigma'(\langle \mathbf{w}_{1,r}^{(t')}, \boldsymbol{\xi}_i\rangle) - n\lambda\mathbf{w}_{1,r}^{(t')}[1] \\
&\geq -\sum_{i:y_i=1} |\ell_{1,i}^{(t')}| \cdot \widetilde{\Theta}(\alpha) + n\lambda|\mathbf{w}_{1,r}^{(t')}[1]| + \sum_{i:y_i=-1} |\ell_{1,i}^{(t')}| \cdot |\mathbf{w}_{1,r}^{(t')}[1]|^{q-1}, \\
&\geq -\sum_{i:y_i=1} |\ell_{1,i}^{(t')}| \cdot \widetilde{\Theta}(\alpha) + n\lambda|\mathbf{w}_{1,r}^{(t')}[1]|,
\end{aligned}
$$

where in the inequality we use Hypothesis (C.13) and (C.14) to get that

$$
\langle \mathbf{w}_{y_i,r}^{(t')}, \boldsymbol{\xi}_i\rangle \leq \sum_{k\in\mathcal{B}_i} |\mathbf{w}_{y_i,r}^{(t')}[k]| \cdot \max_{k\in\mathcal{B}_i} |\boldsymbol{\xi}_i[k]| + \alpha|\langle \mathbf{w}_{y_i,r}^{(t')}, \mathbf{v}\rangle| = \widetilde{\Theta}(1).
$$

Recall from (C.18) that we have $|\ell_{j,i}^{(t')}| = \widetilde{O}\left(\frac{n\lambda}{s\sigma_p^2}\right)$, therefore we have if $\mathbf{w}_{j,r}^{(t')}[1]$ is smaller than some value in the order of $-\widetilde{\Theta}\left(\frac{n\alpha}{s\sigma_p^2}\right) \cdot \mathrm{polylog}(d)$, then

$$
-n\nabla_{\mathbf{w}_{1,r}^{(t)}} L(\mathbf{W}^{(t')})[1] \geq -\widetilde{\Theta}\left(\frac{\alpha n^2 \lambda}{s\sigma_p^2}\right) + \widetilde{\Theta}\left(\frac{n\lambda \cdot n\alpha}{s\sigma_p^2}\right) \cdot \mathrm{polylog}(d) \geq \widetilde{\Theta}(n\eta),
$$

which by Lemma C.2 implies that $\mathbf{w}_{j,r}^{(t')}[1]$ will increase. Therefore, we can conclude that $\mathbf{w}^{(t+1)} \in \left[-\widetilde{O}\left(\frac{n\alpha}{s\sigma_p^2}\right), 0\right)$ in this case, which verifies Hypothesis (C.14). $\qquad\square$

**Lemma C.6** (Lemma 5.5, restated). If the step size satisfies $\eta = O(d^{-1/2})$, then for any $t$ it holds that

$$
L(\mathbf{W}^{(t+1)}) - L(\mathbf{W}^{(t)}) \leq -\eta\|\nabla L(\mathbf{W}^{(t)})\|_1 + \widetilde{\Theta}(\eta^2 d).
$$

*Proof.* Let $\Delta F_{j,i} = F_j(\mathbf{W}^{(t+1)}, \mathbf{x}_i) - F_j(\mathbf{W}^{(t)}, \mathbf{x}_i)$. Then regarding the loss function

$$
L_i(\mathbf{W}) = -\log\frac{e^{F_{y_i}(\mathbf{W},\mathbf{x}_i)}}{\sum_j e^{F_j(\mathbf{W},\mathbf{x}_i)}} = -F_{y_i}(\mathbf{W},\mathbf{x}_i) + \log\left(\sum_j e^{F_j(\mathbf{W},\mathbf{x}_i)}\right).
$$

It is clear that the function $L_i(\mathbf{W})$ is 1-smooth with respect to the vector $[F_{-1}(\mathbf{W}, \mathbf{x}_i), F_1(\mathbf{W}, \mathbf{x}_i)]$. Then based on the definition of $\Delta F_{j,i}$, we have

$$
L_i(\mathbf{W}^{(t+1)}) - L_i(\mathbf{W}^{(t)}) \leq \sum_j \frac{\partial L_i(\mathbf{W}^{(t)})}{\partial F_j(\mathbf{W}^{(t)}, \mathbf{x}_i)} \cdot \Delta F_{j,i} + \sum_j (\Delta F_{j,i})^2. \qquad \text{(C.20)}
$$

Moreover, note that

$$
F_j(\mathbf{W}^{(t)}, \mathbf{x}_i) = \sum_{r=1}^{m} \left[\sigma(\langle \mathbf{w}_{j,r}^{(t)}, y_i\mathbf{v}\rangle) + \sigma(\langle \mathbf{w}_{j,r}^{(t)}, \boldsymbol{\xi}_i\rangle)\right].
$$

By the results that $\langle \mathbf{w}_{j,r}^{(t)}, \mathbf{v}\rangle \leq \widetilde{\Theta}(1)$ and $\langle \mathbf{w}_{j,r}^{(t)}, \boldsymbol{\xi}\rangle \leq \widetilde{\Theta}(1)$, for any $\eta = O(d^{-1/2})$, we have

$$
\langle \mathbf{w}_{j,r}^{(t+1)}, \mathbf{v}\rangle \leq \langle \mathbf{w}_{j,r}^{(t)}, \mathbf{v}\rangle + \eta \leq \widetilde{\Theta}(1), \quad \langle \mathbf{w}_{j,r}^{(t+1)}, \boldsymbol{\xi}_i\rangle \leq \langle \mathbf{w}_{j,r}^{(t)}, \boldsymbol{\xi}_i\rangle + \widetilde{\Theta}(\eta s^{1/2}) \leq \widetilde{\Theta}(1),
$$

which implies that the smoothness parameter of the functions $\sigma(\langle \mathbf{w}_{j,r}^{(t)}, y_i\mathbf{v}\rangle)$ and $\sigma(\langle \mathbf{w}_{j,r}^{(t)}, \boldsymbol{\xi}_i\rangle)$ are at most $\widetilde{\Theta}(1)$ for any $\mathbf{w}$ in the path between $\mathbf{w}_{j,r}^{(t)}$ and $\mathbf{w}_{j,r}^{(t+1)}$. Then we can apply first Taylor expansion on $\sigma(\langle \mathbf{w}_{j,r}^{(t)}, y_i\mathbf{v}\rangle)$ and $\sigma(\langle \mathbf{w}_{j,r}^{(t)}, \boldsymbol{\xi}_i\rangle)$ and bound the second-order error as follows,

$$
\begin{aligned}
&\left|\sigma(\langle \mathbf{w}_{j,r}^{(t+1)}, y_i\mathbf{v}\rangle) - \sigma(\langle \mathbf{w}_{j,r}^{(t)}, y_i\mathbf{v}\rangle) - \langle\nabla_{\mathbf{w}_{j,r}}\sigma(\langle \mathbf{w}_{j,r}^{(t)}, y_i\mathbf{v}\rangle), \mathbf{w}_{j,r}^{(t+1)} - \mathbf{w}_{j,r}^{(t)}\rangle\right| \\
&\leq \widetilde{\Theta}\left(\|\mathbf{w}_{j,r}^{(t+1)} - \mathbf{w}_{j,r}^{(t)}\|_2^2\right) = \widetilde{\Theta}(\eta^2 d), \qquad\qquad\qquad\qquad\qquad\qquad\qquad\qquad\text{(C.21)}
\end{aligned}
$$

where the last inequality is due to Lemma C.2 that

$$[\mathbf{w}_{j,r}^{(t+1)} - \mathbf{w}_{j,r}^{(t)}]^2 = \eta^2 \left\| \frac{\mathbf{m}_{j,r}^{(t)}}{\sqrt{\mathbf{v}_{j,r}^{(t)}}} \right\|_2^2 \leq \Theta(\eta^2 d).$$

Similarly, we can also show that

$$\left| \sigma(\langle \mathbf{w}_{j,r}^{(t+1)}, \boldsymbol{\xi}_i \rangle) - \sigma(\langle \mathbf{w}_{j,r}^{(t)}, \boldsymbol{\xi}_i \rangle) - \langle \nabla_{\mathbf{w}_{j,r}} \sigma(\langle \mathbf{w}_{j,r}^{(t)}, \boldsymbol{\xi}_i \rangle), \mathbf{w}_{j,r}^{(t+1)} - \mathbf{w}_{j,r}^{(t)} \rangle \right| \leq \Theta(\eta^2 d). \quad \text{(C.22)}$$

Combining the above bounds on the second-order errors, we have

$$\left| \Delta F_{j,i} - \langle \nabla_{\mathbf{W}} F_j(\mathbf{W}^{(t)}, \mathbf{x}_i), \mathbf{W}^{(t+1)} - \mathbf{W}^{(t)} \rangle \right| \leq \widetilde{\Theta}(m\eta^2 d) = \widetilde{\Theta}(\eta^2 d), \quad \text{(C.23)}$$

where the last equation is due to our assumption that $m = \widetilde{\Theta}(1)$. Besides, by (C.21) and (C.22) the convexity property of the function $\sigma(x)$, we also have

$$\begin{aligned}
\left| \sigma(\langle \mathbf{w}_{j,r}^{(t+1)}, y_i \mathbf{v} \rangle) - \sigma(\langle \mathbf{w}_{j,r}^{(t)}, y_i \mathbf{v} \rangle) \right| &\leq \left| \langle \nabla_{\mathbf{w}_{j,r}} \sigma(\langle \mathbf{w}_{j,r}^{(t)}, y_i \mathbf{v} \rangle), \mathbf{w}_{j,r}^{(t+1)} - \mathbf{w}_{j,r}^{(t)} \rangle \right| + \widetilde{\Theta}(\eta^2 d) \\
&= \widetilde{\Theta}\big( \eta |\sigma'(\langle \mathbf{w}_{j,r}^{(t+1)}, y_i \mathbf{v} \rangle)| \cdot \|\mathbf{v}\|_1 \big) + \widetilde{\Theta}(\eta^2 d) \\
&= \widetilde{\Theta}(\eta + \eta^2 d); \\
\left| \sigma(\langle \mathbf{w}_{j,r}^{(t+1)}, \boldsymbol{\xi}_i \rangle) - \sigma(\langle \mathbf{w}_{j,r}^{(t)}, \boldsymbol{\xi}_i \rangle) \right| &\leq \left| \langle \nabla_{\mathbf{w}_{j,r}} \sigma(\langle \mathbf{w}_{j,r}^{(t)}, \boldsymbol{\xi}_i \rangle), \mathbf{w}_{j,r}^{(t+1)} - \mathbf{w}_{j,r}^{(t)} \rangle \right| + \widetilde{\Theta}(\eta^2 d) \\
&= \widetilde{\Theta}\big( \eta |\sigma'(\langle \mathbf{w}_{j,r}^{(t+1)}, \boldsymbol{\xi}_i \rangle)| \cdot \|\boldsymbol{\xi}\|_1 \big) + \widetilde{\Theta}(\eta^2 d) \\
&= \widetilde{\Theta}(\eta s \sigma_p + \eta^2 d).
\end{aligned}$$

These bounds further imply that

$$|\Delta F_{j,i}| \leq \widetilde{\Theta}\big( m \cdot (\eta s \sigma_p + \eta^2 d) \big) = \widetilde{\Theta}\big( \eta s \sigma_p + \eta^2 d \big). \quad \text{(C.24)}$$

Now we can plug (C.23) and (C.24) into (C.20) and get

$$\begin{aligned}
L_i(\mathbf{W}^{(t+1)}) - L_i(\mathbf{W}^{(t)}) &\leq \sum_j \frac{\partial L_i(\mathbf{W}^{(t)})}{\partial F_j(\mathbf{W}^{(t)}, \mathbf{x}_i)} \cdot \Delta F_{j,i} + \sum_j (\Delta F_{j,i})^2 \\
&\leq \sum_j \frac{\partial L_i(\mathbf{W}^{(t)})}{\partial F_j(\mathbf{W}^{(t)}, \mathbf{x}_i)} \cdot \langle \nabla_{\mathbf{W}} F_j(\mathbf{W}^{(t)}, \mathbf{x}_i), \mathbf{W}^{(t+1)} - \mathbf{W}^{(t)} \rangle \\
&\quad + \widetilde{\Theta}(\eta^2 d) + \widetilde{\Theta}\big( (\eta s \sigma_p + \eta^2 d)^2 \big) \\
&= \langle \nabla L_i(\mathbf{W}^{(t)}), \mathbf{W}^{(t+1)} - \mathbf{W}^{(t)} \rangle + \widetilde{\Theta}(\eta^2 d), \quad \text{(C.25)}
\end{aligned}$$

where in the second inequality we use the fact that $L_i(\mathbf{W})$ is 1-Lipschitz with respect to $F_j(\mathbf{W}, \mathbf{x}_i)$ and the last equation is due to our assumption that $\sigma_p = O(s^{-1/2})$ so that $\widetilde{\Theta}((\eta s \sigma_p + \eta^2 d)^2) = \widetilde{O}(\eta^2 d)$.

Now we are ready to characterize the behavior on the entire training objective $L(\mathbf{W}) = n^{-1} \sum_{i=1}^n L_i(\mathbf{W}) + \lambda \|\mathbf{W}\|_F^2$. Note that $\lambda \|\mathbf{W}\|_F^2$ is $2\lambda$-smoothness, where $\lambda = o(1)$. Then applying (C.25) for all $i \in [n]$ gives

$$\begin{aligned}
L(\mathbf{W}^{(t+1)}) - L(\mathbf{W}^{(t)}) &= \frac{1}{n} \sum_{i=1}^n \big[ L_i(\mathbf{W}^{(t+1)}) - L_i(\mathbf{W}^{(t)}) \big] + \lambda \big( \|\mathbf{W}^{(t+1)}\|_F^2 - \|\mathbf{W}^{(t)}\|_F^2 \big) \\
&\leq \langle \nabla L(\mathbf{W}^{(t)}), \mathbf{W}^{(t+1)} - \mathbf{W}^{(t)} \rangle + \widetilde{\Theta}(\eta^2 d),
\end{aligned}$$

where the second equation uses the fact that $\|\mathbf{W}^{(t+1)} - \mathbf{W}^{(t)}\|_F^2 = \widetilde{\Theta}(\eta^2 d)$. Recall that we have

$$\mathbf{w}_{j,r}^{(t+1)} - \mathbf{w}_{j,r}^{(t)} = -\eta \cdot \frac{\mathbf{m}_{j,r}^{(t)}}{\sqrt{\mathbf{v}_{j,r}^{(t)}}}.$$

Then by Lemma C.2, we know that $\mathbf{m}_{j,r}^{(t)}[k]/\sqrt{\mathbf{v}_{j,r}^{(t)}[k]}$ is close to sign gradient if $\nabla L(\mathbf{w}^{(t)})[k]$ is large. Then we have

$$\left\langle \nabla_{\mathbf{w}_{j,r}} L(\mathbf{W}^{(t)}), \frac{\mathbf{m}_{j,r}^{(t)}}{\sqrt{\mathbf{v}_{j,r}^{(t)}}} \right\rangle \geq \Theta\big(\big\|\nabla_{\mathbf{w}_{j,r}} L(\mathbf{W}^{(t)})\big\|_1\big) - \widetilde{\Theta}(d \cdot \eta) - \widetilde{\Theta}(ns \cdot \eta s \sigma_p)$$

$$\geq \Theta\big(\big\|\nabla_{\mathbf{w}_{j,r}} L(\mathbf{W}^{(t)})\big\|_1\big) - \widetilde{\Theta}(d\eta),$$

where the second and last terms on the R.H.S. of the first inequality are contributed by the small gradient coordinates $k \notin \cup_{i=1}^n \mathcal{B}_i$ and $k \in \cup_{i=1}^n \mathcal{B}_i$ respectively, and the last inequality is by the fact that $ns^2\sigma_p = O(d)$. Therefore, based on this fact (C.25) further leads to

$$L(\mathbf{W}^{(t+1)}) - L(\mathbf{W}^{(t)}) \leq -\eta\|\nabla L(\mathbf{W}^{(t)})\|_1 + \widetilde{\Theta}(\eta^2 d),$$

which completes the proof.

$\square$

**Lemma C.7** (Generalization Performance of Adam). Let

$$\mathbf{W}^* = \operatorname*{argmin}_{\mathbf{W} \in \{\mathbf{W}^{(1)}, \dots, \mathbf{W}^{(T)}\}} \|\nabla L(\mathbf{W})\|_1.$$

Then for all training data, we have

$$\frac{1}{n} \sum_{i=1}^n \mathbb{1}\left[F_{y_i}(\mathbf{W}^*, \mathbf{x}_i) \leq F_{-y_i}(\mathbf{W}^*, \mathbf{x}_i)\right] = 0.$$

Moreover, in terms of the test data $(\mathbf{x}, y) \sim \mathcal{D}$, we have

$$\mathbb{P}_{(\mathbf{x},y)\sim\mathcal{D}}\left[F_y(\mathbf{W}^*, \mathbf{x}) \leq F_{-y}(\mathbf{W}^*, \mathbf{x})\right] \geq \frac{1}{2}.$$

*Proof.* By Lemma C.6, we know that the algorithm will converge to a point with very small gradient (up to $O(\eta d)$ in $\ell_1$ norm). Then in terms of a noise vector $\boldsymbol{\xi}_i$, we have

$$\sum_{k \in \mathcal{B}_i} \big|\nabla_{\mathbf{w}_{y_i,r}} L(\mathbf{W}^*)[k]\big| \leq O(\eta d). \tag{C.26}$$

Note that

$$n\nabla_{\mathbf{w}_{y_i,r}} L(\mathbf{W}^*)[k] = \ell_{y_i,i}^* \sigma'(\langle \mathbf{w}_{y_i,r}^*, \boldsymbol{\xi}_i \rangle)\boldsymbol{\xi}_i[k] - n\lambda \mathbf{w}_{y_i,r}^*[k],$$

where $\ell_{y_i,i}^* = 1 - \operatorname{logit}_{y_i}(F^*, \mathbf{x}_i)$. Then by triangle inequality and (C.26), we have for any $r \in [m]$,

$$\left|\sum_{k \in \mathcal{B}_i} |\ell_{y_i,i}^*| \sigma'(\langle \mathbf{w}_{y_i,r}^*, \boldsymbol{\xi}_i \rangle)|\boldsymbol{\xi}_i[k]| - n\lambda \sum_{k \in \mathcal{B}_i} |\mathbf{w}_{y_i,r}^*[k]|\right| \leq n \sum_{k \in \mathcal{B}_i} \big|\nabla_{\mathbf{w}_{y_i,r}} L(\mathbf{W}^*)[k]\big| \leq O(n\eta d).$$

Then by Lemma C.5, let $r^* = \arg\max_{r \in [m]} \langle \mathbf{w}_{y_i,r}^*, \boldsymbol{\xi}_i \rangle$, we have $\langle \mathbf{w}_{y_i,r^*}, \boldsymbol{\xi}_i \rangle = \widetilde{\Theta}(1)$ and $\sum_{k \in \mathcal{B}_i} |\mathbf{w}_{y_i,r^*}^*[k]| \cdot |\boldsymbol{\xi}_i[k]| = \widetilde{\Theta}(1)$. Note that $|\boldsymbol{\xi}_i[k]| = \widetilde{O}(\sigma_p)$, we have $\sum_{k \in \mathcal{B}_i} |\mathbf{w}_{y_i,r^*}^*[k]| \geq \widetilde{\Theta}(1/\sigma_p)$. Then according to the inequality above, it holds that

$$|\ell_{y_i,i}^*| \cdot \widetilde{\Theta}(s\sigma_p) \geq \widetilde{\Theta}\left(n\lambda \sum_{k \in \mathcal{B}_i} |\mathbf{w}_{y_i,r}^*[k]| - n\eta d\right) \geq \widetilde{\Theta}\left(\frac{n\lambda}{\sigma_p}\right),$$

where the second inequality is due to our choice of $\eta$. This further implies that $|\ell_{y_i,i}^*| = |\ell_{-y_i,i}^*| = \widetilde{\Theta}\left(\frac{n\lambda}{s\sigma_p^2}\right)$ by combining the above results with (C.18). Then let us move to the gradient with respect to the first coordinate. In particular, since $\|\nabla L(\mathbf{W}^*)\|_1 \leq O(\eta d)$, we have

$$|n\nabla_{\mathbf{w}_{j,r}} L(\mathbf{W}^*)[1]| = \left|\sum_{i=1}^n y_i \ell_{j,i}^* \sigma'(\langle \mathbf{w}_{j,r}^*, y_i\mathbf{v}\rangle) - \alpha\sum_{i=1}^n y_i \ell_{j,i}^* \sigma'(\langle \mathbf{w}_{j,r}^*, \boldsymbol{\xi}_i\rangle) - n\lambda\mathbf{w}_{j,r}^*[1]\right|$$

$$\leq O(n\eta d). \tag{C.27}$$

Then note that $\mathrm{sgn}(y_i \ell_{j,i}^*) = \mathrm{sgn}(j)$, it is clear that $\mathbf{w}_{j,r^*}^*[1] \cdot j \leq 0$ since otherwise

$$\left| n \nabla_{\mathbf{w}_{j,r^*}} L(\mathbf{W}^*)[1] \right| \geq \left| \alpha \sum_{i=1}^{n} y_i \ell_{j,i}^* \left[ \sigma'(\langle \mathbf{w}_{j,r^*}^*, \boldsymbol{\xi}_i \rangle) - \sigma'(\langle \mathbf{w}_{j,r^*}^*, y_i \mathbf{v} \rangle) \right] \right| \geq \widetilde{\Theta}\left( \frac{\alpha n^2 \lambda}{s \sigma_p^2} \right) \geq \widetilde{\Omega}(n\eta d),$$

which contradicts (C.27). Therefore, using the fact that $\mathbf{w}_{j,r^*}^*[1] \cdot j \leq 0$, we have

$$\left| n \nabla_{\mathbf{w}_{j,r^*}} L(\mathbf{W}^*)[1] \right| = \left| \alpha \sum_{i:y_i=j}^{n} y_i \ell_{j,i}^* \sigma'(\langle \mathbf{w}_{j,r^*}^*, \boldsymbol{\xi}_i \rangle) - \sum_{i:y_i=-j}^{n} y_i \ell_{j,i}^* \sigma'(|\mathbf{w}_{j,r^*}^*[1]|) - n\lambda |\mathbf{w}_{j,r^*}^*[1]| \right|.$$

Then applying (C.27)and using the fact that $|\ell_{y_i,i}^*| = |\ell_{-y_i,i}^*| = \widetilde{\Theta}\left( \frac{n\lambda}{s\sigma_p^2} \right)$ for all $i \in [n]$, it is clear that

$$|\mathbf{w}_{j,r^*}^*[1]| \geq \widetilde{\Theta}\left( \alpha^{1/(q-1)} \wedge \frac{n\alpha}{s\sigma_p^2} \right) \geq \widetilde{\Theta}\left( \frac{n\alpha}{s\sigma_p^2} \right),$$

where the second equality is due to our choice of $\sigma_p$ and $\alpha$. Then combining with Lemma C.5 and the fact that $\mathbf{w}_{j,r^*}^*[1] \cdot j < 0$, we have

$$\mathbf{w}_{j,r^*}^*[1] \cdot j \leq -\widetilde{\Theta}\left( \frac{n\alpha}{s\sigma_p^2} \right).$$

Now we are ready to evaluate the training error and test error. In terms of training error, it is clear that by Lemma C.5, we have $\langle \mathbf{w}_{y_i,r^*}^*, \boldsymbol{\xi}_i \rangle \geq \widetilde{\Theta}(1)$, $\langle \mathbf{w}_{y_i,r}^*, \boldsymbol{\xi}_i \rangle \geq -o(1)$, and $|\langle \mathbf{w}_{y_i,r}^*, \mathbf{v} \rangle| = o(1)$, $|\langle \mathbf{w}_{-y_i,r}^*, \boldsymbol{\xi}_i \rangle| = o(1)$. Then we have for any training data $(\mathbf{x}_i, y_i)$,

$$F_{y_i}(\mathbf{W}^*, \mathbf{x}_i) = \sum_{r=1}^{m} \left[ \sigma(\langle \mathbf{w}_{y_i,r}^*, y_i \mathbf{v} \rangle) + \sigma(\langle \mathbf{w}_{y_i,r}^*, \boldsymbol{\xi}_i \rangle) \right] = \widetilde{\Theta}(1),$$

$$F_{-y_i}(\mathbf{W}^*, \mathbf{x}_i) = \sum_{r=1}^{m} \left[ \sigma(\langle \mathbf{w}_{-y_i,r}^*, -y_i \mathbf{v} \rangle) + \sigma(\langle \mathbf{w}_{-y_i,r}^*, \boldsymbol{\xi}_i \rangle) \right] = o(1),$$

which directly implies that the NN model $\mathbf{W}^*$ can correctly classify all training data and thus achieve zero training error.

In terms of the test data $(\mathbf{x}, y)$ where $\mathbf{x} = [y\mathbf{v}, \boldsymbol{\xi}]$, which is generated according to Definition 3.1. Note that for each neural, its weight $\mathbf{w}_{j,r}^*$ can be decomposed into two parts: the first coordinate and the rest $d-1$ coordinates. As previously discussed, for any $j \in [2]$ and $r = r^*$, we have $\mathrm{sgn}(j) \cdot \mathbf{w}_{j,r}^*[1] \leq -\widetilde{\Theta}(n\alpha/(s\sigma_p^2))$ and $\mathrm{sgn}(j) \cdot \mathbf{w}_{j,r}^*[1] \leq \widetilde{\Theta}(\lambda^{-1}\eta)$ for $r \neq r^*$. Therefore, using the fact that $\widetilde{\Theta}(n\alpha/(s\sigma_p^2)) = \omega(\lambda^{-1}\eta)$ and Lemma C.5, given the test data $(\mathbf{x}, y)$, we have

$$F_y(\mathbf{W}^*, \mathbf{x}) = \sum_{r=1}^{m} \left[ \sigma(\langle \mathbf{w}_{y,r}^*, y\mathbf{v} \rangle) + \sigma(\langle \mathbf{w}_{y,r}^*, \boldsymbol{\xi} \rangle) \right]$$

$$\leq \sum_{r=1}^{m} \widetilde{\Theta}\left( \left[ \alpha \cdot \frac{n\alpha}{s\sigma_p^2} + \zeta_{y,r} \right]_+^q \right),$$

$$F_{-y}(\mathbf{W}^*, \mathbf{x})) = \sum_{r=1}^{m} \left[ \sigma(\langle \mathbf{w}_{-y,r}^*, y\mathbf{v} \rangle) + \sigma(\langle \mathbf{w}_{-y,r}^*, \boldsymbol{\xi} \rangle) \right]$$

$$\geq \widetilde{\Theta}\left[ |\mathbf{w}_{-y,r^*}^*[1]|^q + [\zeta_{-y,r^*}]_+^q \right]$$

$$\geq \Theta\left( \left[ \frac{n\alpha}{s\sigma_p^2} \right]_+^q + [\zeta_{-y,r^*}]_+^q \right),$$

where the random variables $\zeta_{y,r}$ and $\zeta_{y,r}$ are symmetric and independent of $\mathbf{v}$. Besides, note that $\alpha = o(1)$, it can be clearly shown that $\alpha \cdot n\alpha/(s\sigma_p^2) \ll n\alpha/(s\sigma_p^2)$. Therefore, if the random noise $\zeta_{y,r}$ and $\zeta_{-y,r}$ are dominated by the feature noise term $\langle \mathbf{w}_{-y,r^*}^*, y\mathbf{v} \rangle$, we can directly get that $F_y(\mathbf{W}^*, \mathbf{x}) \leq F_{-y}(\mathbf{W}^*, \mathbf{x}))$ (recall that $m = \widetilde{\Theta}(1)$), which implies that the model has been biased by the feature noise and the true feature vector in the test dataset will not give any "positive" effect to the classification. Also note that $\zeta_y$ and $\zeta_{-y}$ are also independent of $\mathbf{v}$, which implies that if the random noise dominates the feature noise term, the model $\mathbf{W}^*$ will give at least $0.5$ error on test data. In sum, we can conclude that with probability at least $1/2$ it holds that $F_y(\mathbf{W}^*, \mathbf{x}) \leq F_{-y}(\mathbf{W}^*, \mathbf{x})$, which implies that the output of Adam achieves $1/2$ test error. $\qquad \square$

### C.3 PROOF FOR GRADIENT DESCENT

Recall the feature learning and noise memorization of gradient descent can be formulated by

$$
\langle \mathbf{w}_{j,r}^{(t+1)}, j \cdot \mathbf{v} \rangle = (1 - \eta\lambda) \cdot \langle \mathbf{w}_{j,r}^{(t)}, j \cdot \mathbf{v} \rangle
$$
$$
+ \frac{\eta}{n} \cdot j \cdot \left( \sum_{i=1}^{n} y_i \ell_{j,i}^{(t)} \sigma'(\langle \mathbf{w}_{j,r}^{(t)}, y_i \mathbf{v} \rangle) - \alpha \sum_{i=1}^{n} y_i \ell_{j,i}^{(t)} \sigma'(\langle \mathbf{w}_{j,r}^{(t)}, \boldsymbol{\xi}_i \rangle) \right),
$$
$$
\langle \mathbf{w}_{y_i,r}^{(t+1)}, \boldsymbol{\xi}_i \rangle = (1 - \eta\lambda) \cdot \langle \mathbf{w}_{y_i,r}^{(t)}, \boldsymbol{\xi}_i \rangle + \frac{\eta}{n} \cdot \sum_{k \in \mathcal{B}_i} \ell_{y_i,i}^{(t)} \sigma'(\langle \mathbf{w}_{y_i,r}^{(t)}, \boldsymbol{\xi}_i \rangle) \cdot \boldsymbol{\xi}_i[k]^2
$$
$$
+ \frac{\eta\alpha}{n} \cdot \left( \alpha \sum_{s=1}^{n} \ell_{y_i,s}^{(t)} \sigma'(\langle \mathbf{w}_{y_i,r}^{(t)}, \boldsymbol{\xi}_s \rangle) - \sum_{s=1}^{n} y_s \ell_{y_i,s}^{(t)} \sigma'(\langle \mathbf{w}_{y_i,r}^{(t)}, y_s \mathbf{v} \rangle) \right). \quad \text{(C.28)}
$$

Then similar to the analysis for Adam, we decompose the gradient descent process into multiple stages and characterize the algorithmic behaviors separately. The following lemma characterizes the first training stage, i.e., the stage where all outputs $F_j(\mathbf{W}^{(t)}, \mathbf{x}_i)$ remain in the constant level for all $j$ and $i$.

**Lemma C.8.** [Lemma 5.6, restated] Suppose the training data is generated according to Definition 3.1 and $\lambda = o(\sigma_0^{q-2}\sigma_p/n)$. Let $\Lambda_j^{(t)} = \max_{r \in [m]} \langle \mathbf{w}_{j,r}^{(t+1)}, j \cdot \mathbf{v} \rangle$, $\Gamma_{j,i}^{(t)} = \max_{r \in [m]} \langle \mathbf{w}_{j,r}^{(t)}, \boldsymbol{\xi}_i \rangle$, and $\Gamma_j^{(t)} = \max_{i:y_i=j} \Gamma_{j,i}^{(t)}$. Then let $T_j$ be the iteration number that $\Lambda_j^{(t)}$ reaches $\Theta(1/m)$, we have

$$
T_j = \widetilde{\Theta}(\sigma_0^{2-q}/\eta) \quad \text{for all } j \in \{-1, 1\}.
$$

Moreover, let $T_0 = \min_{j \in \{\pm 1\}} \{T_j\}$, then for all $t \leq T_0$ it holds that $\Gamma_j^{(t)} = \widetilde{O}(\sigma_0)$ for all $j \in \{-1, 1\}$.

We first provide the following useful lemma.

**Lemma C.9.** Let $\{x_t, y_t\}_{t=1,\dots}$ be two positive sequences that satisfy

$$
x_{t+1} \geq x_t + \eta \cdot A x_t^{q-1},
$$
$$
y_{t+1} \leq y_t + \eta \cdot B y_t^{q-1},
$$

for some $A = \Theta(1)$ and $B = o(1)$. Then for any $q \geq 3$ and suppose $y_0 = O(x_0)$ and $\eta < O(x_0)$, we have for every $C \in [x_0, O(1)]$, let $T_x$ be the first iteration such that $x_t \geq C$, then we have $T_x \eta = \Theta(x_0^{2-q})$ and

$$
y_{T_x} \leq O(x_0).
$$

*Proof.* By Claim C.20 in Allen-Zhu & Li (2020b), we have $T_x \eta = \Theta(x_0^{2-q})$. Then we will show

$$
y_t \leq 2x_0
$$

for all $t \leq T_x$. In particular, let $T_x \eta = C' x_0^{2-q}$ for some absolute constant $C'$ and assume $C' B 2^{q-1} < 1$ (this is true since $B = o(1)$), we first made the following induction hypothesis on $y_t$ for all $t \leq T_a$,

$$
y_t \leq y_0 + t\eta B'(2x_0)^{q-1}.
$$

Note that for any $t \leq T_0$, this hypothesis clearly implies that

$$
y_t \leq y_0 + T_x \eta B' 2^{q-1} x_0^{q-1} \leq x_0 + CB 2^{q-1} x_0^{2-q} \cdot x_0^{q-1} \leq 2x_0.
$$

Then we are able to verify the hypothesis at time $t+1$ based on the recursive upper bound of $y_t$, i.e.,

$$
y_{t+1} \leq y_t + \eta \cdot B y_t^{q-1}
$$
$$
\leq y_0 + t\eta B(2x_0)^{q-1} + \eta \cdot B y_t^{q-1}
$$
$$
\leq y_0 + (t+1)\eta B(2x_0)^{q-1}.
$$

Therefore, we can conclude that $y_t \leq 2x_0$ for all $t \leq T_x$. This completes the proof. $\square$

Now we are ready to complete the proof of Lemma C.8.

*Proof of Lemma C.8.* Note that at the initialization, we have $|\langle \mathbf{w}_{j,r}^{(0)}, \mathbf{v} \rangle| = \widetilde{\Theta}(\sigma_0)$ and $|\langle \mathbf{w}_{j,r}^{(0)}, \boldsymbol{\xi}_i \rangle| = \widetilde{\Theta}(s^{1/2}\sigma_p\sigma_0)$. Then based on the parameter scaling summarized in Appendix C.1, we have

$$F_j(\mathbf{W}^{(0)}, \mathbf{x}_i) = \sum_{r=1}^{m} \left[ \sigma(\langle \mathbf{w}_{j,r}^{(0)}, y_i\mathbf{v} \rangle) + \sigma(\langle \mathbf{w}_{j,r}^{(0)}, \boldsymbol{\xi}_i \rangle) \right] = o(1)$$

for all $j \in \{-1, 1\}$. Then we have

$$|\ell_{j,i}^{(0)}| \geq \min \left\{ \frac{e^{F_j(\mathbf{W}^{(0)}, \mathbf{x}_i)}}{\sum_j e^{F_{+1}(\mathbf{W}^{(0)}, \mathbf{x}_i)}}, \frac{e^{F_{-1}(\mathbf{W}^{(0)}, \mathbf{x}_i)}}{\sum_j e^{F_j(\mathbf{W}^{(0)}, \mathbf{x}_i)}} \right\} = \Theta(1).$$

Then we will consider the training period where $|\ell_{j,i}^{(t)}| = \Theta(1)$ for all $j$, $i$, and $t$. Besides, note that $\mathrm{sgn}(y_i\ell_{j,i}^{(t)}) = j$. Therefore, let $r^* = \arg\max_r \langle \mathbf{w}_{j,r}^{(t-1)}, j \cdot \mathbf{v} \rangle$, (C.28) implies that

$$
\begin{aligned}
\Lambda_j^{(t)} &= \langle \mathbf{w}_{j,r^*}^{(t-1)}, j \cdot \mathbf{v} \rangle \\
&= (1 - \eta\lambda) \cdot \langle \mathbf{w}_{j,r^*}^{(t-1)}, j \cdot \mathbf{v} \rangle \\
&\quad + \frac{\eta}{n} \cdot \left( \sum_{i=1}^{n} |\ell_{j,i}^{(t-1)}| \sigma'(\langle \mathbf{w}_{j,r^*}^{(t-1)}, y_i\mathbf{v} \rangle) - \alpha \sum_{i=1}^{n} |\ell_{j,i}^{(t-1)}| \sigma'(\langle \mathbf{w}_{j,r^*}^{(t-1)}, \boldsymbol{\xi}_i \rangle) \right) \\
&\geq (1 - \eta\lambda) \cdot \langle \mathbf{w}_{j,r^*}^{(t-1)}, j \cdot \mathbf{v} \rangle + \Theta(\eta) \cdot \left[ \sigma'(\langle \mathbf{w}_{j,r^*}^{(t-1)}, j \cdot \mathbf{v} \rangle) - \alpha\sigma'(\Gamma_j^{(t-1)}) \right] \\
&\geq (1 - \eta\lambda)\Lambda_j^{(t-1)} + \eta \cdot \Theta\left((\Lambda_j^{(t-1)})^{q-1}\right) - \eta \cdot \Theta\left(\alpha(\Gamma_j^{(t-1)})^{q-1}\right).
\end{aligned}
\tag{C.29}
$$

Similarly, let $r^* = \arg\max_r \langle \mathbf{w}_{y_i,r}^{(t)}, \boldsymbol{\xi}_i \rangle$, we also have the following according to (C.28)

$$
\begin{aligned}
\Gamma_{y_i,i}^{(t)} &= \langle \mathbf{w}_{y_i,r^*}^{(t)}, \boldsymbol{\xi}_i \rangle \\
&\leq (1 - \eta\lambda)\langle \mathbf{w}_{y_i,r^*}^{(t-1)}, \boldsymbol{\xi}_i \rangle + \widetilde{\Theta}\left(\frac{\eta s\sigma_p^2}{n}\right) \cdot \sigma'(\langle \mathbf{w}_{y_i,r^*}^{(t-1)}, \boldsymbol{\xi}_i \rangle) + \Theta\left(\frac{\eta\alpha^2}{n}\right) \cdot \sum_{s=1}^{n} \sigma'(\langle \mathbf{w}_{y_i,r^*}^{(t-1)}, \boldsymbol{\xi}_s \rangle) \\
&\leq \Gamma_{y_i,i}^{(t-1)} + \widetilde{\Theta}\left(\frac{\eta s\sigma_p^2(\Gamma_{y_i,i}^{(t-1)})^{q-1}}{n}\right) + \Theta\left(\frac{\eta\alpha^2}{n} \cdot \sum_{s=1}^{n} (\Gamma_{y_i,s}^{(t-1)})^{q-1}\right).
\end{aligned}
$$

Then by our definition of $\Gamma_j^{(t)} = \max_{i \in [n]} \Gamma_{j,i}^{(t)}$, we further get the following for all $j \in \{-1, 1\}$,

$$\Gamma_j^{(t)} \leq \Gamma_j^{(t-1)} + \widetilde{\Theta}\left(\frac{\eta s\sigma_p^2 + n\eta\alpha^2}{n} \cdot (\Gamma_j^{(t-1)})^{q-1}\right) = \Gamma_j^{(t-1)} + \Theta\left(\frac{\eta s\sigma_p^2}{n} \cdot (\Gamma_j^{(t-1)})^{q-1}\right),$$
$$\tag{C.30}$$

where the last equation is by our assumption that $\alpha = \widetilde{O}(s\sigma_p^2/n)$.

Then we will prove the main argument for general $t$, which is based on the following two induction hypotheses

$$\Lambda_j^{(t)} \geq \Lambda_j^{(t-1)} + \eta \cdot \Theta\left((\Lambda_j^{(t-1)})^{q-1}\right), \tag{C.31}$$

$$\Gamma_j^{(t)} \leq \Gamma_j^{(t-1)} + \Theta\left(\frac{\eta s\sigma_p^2}{n} \cdot (\Gamma_j^{(t-1)})^{q-1}\right). \tag{C.32}$$

Note that when $t = 0$, we have already verified these two hypotheses in (C.29) and (C.30), where we use the fact that $\lambda = o(\sigma_0^{q-2}\sigma_p/n) \leq (\Lambda_j^{(0)})^{q-2}$ and $\alpha = o(1)$. Suppose that (C.29) and (C.30) hold for iterations $\tau \leq t$. At time $t + 1$, for all $\tau \leq t$, we have

$$\Gamma_j^{(\tau)} \leq O(\Lambda_j^{(\tau)}),$$

as $s\sigma^2/n = o(1)$ and $\Lambda_j^{(t)}$ increases faster than $\Gamma_j^{(t)}$. Besides, we can also show that $\lambda\Gamma_j^{(t)} \leq (\Gamma_j^{(t)})^{q-1}$, which has been verified at time $t = 0$, since $\Gamma_j^{(t)}$ keeps increasing. Therefore, we have

$$\lambda\Gamma_j^{(t)} \leq (\Gamma_j^{(t)})^{q-1} \leq O((\Lambda_j^{(t)})^{q-1}),$$

and hence (C.29) implies

$$\Lambda_j^{(t+1)} \geq (1 - \eta\lambda)\Lambda_j^{(t)} + \eta \cdot \Theta((\Lambda_j^{(t)})^{q-1}) - \eta \cdot \Theta(\alpha(\Gamma_j^{(t)})^{q-1})$$
$$\geq \Lambda_j^{(t)} + \eta \cdot \Theta((\Lambda_j^{(t)})^{q-1}),$$

which verifies Hypothesis (C.31) at $t + 1$. Additionally, (C.30) implies

$$\Gamma_j^{(t+1)} \leq \Gamma_j^{(t)} + \Theta\left(\frac{\eta s\sigma_p^2}{n} \cdot (\Gamma_j^{(t)})^{q-1}\right),$$

which verifies Hypothesis (C.32) at $t + 1$. Then by Lemma C.9, we have that $\Lambda_j^{(t)} = \widetilde{O}(1)$ for all $t \leq T_0 = \widetilde{\Theta}((\Lambda_j^{(0)})^{2-q}/\eta) = \widetilde{\Theta}(\sigma_0^{2-q}/\eta)$. Moreover, Lemma C.9 also shows that $\Gamma_j^{(t+1)} = O(\Lambda_j^{(0)}) = \widetilde{O}(\sigma_0)$. This completes the proof. $\qquad\square$

**Lemma C.10.** For all $i \in [n]$ and $t \leq T_{-y_i}$, it holds that $\langle \mathbf{w}_{-y_i,r}^{(t)}, \boldsymbol{\xi}_i \rangle \leq \widetilde{\Theta}(\alpha)$.

*Proof.* First of all, for $j \in \{\pm 1\}$, by the definition of $T_j$, we have

$$\langle \mathbf{w}_{j,r}^{(t)}, j \cdot \mathbf{v} \rangle \leq \widetilde{\Theta}(1).$$

Moreover, with the same proof as Lemma C.8, it is clear that $-\langle \mathbf{w}_{j,r}^{(t)}, j \cdot \mathbf{v} \rangle$ is decreasing in $t$ for $t \leq T_j$. Therefore, by the fact that $|\langle \mathbf{w}_{j,r}^{(0)}, \mathbf{v} \rangle| \leq \widetilde{\Theta}(1)$, we have

$$|\langle \mathbf{w}_{j,r}^{(t)}, \mathbf{v} \rangle| \leq \widetilde{\Theta}(1) \tag{C.33}$$

for all $t \leq T_j$.

Now by the update form of GD, we have for any $k \in \mathcal{B}_i$,

$$\mathbf{w}_{-y_i,r}^{(t+1)}[k] \cdot \boldsymbol{\xi}_i[k] = (1 - \eta\lambda) \cdot \mathbf{w}_{-y_i,r}^{(t)}[k] \cdot \boldsymbol{\xi}_i[k] + \frac{\eta}{n} \cdot \sum_{k \in \mathcal{B}_i} \ell_{-y_i,i}^{(t)} \sigma'(\langle \mathbf{w}_{-y_i,r}^{(t)}, \boldsymbol{\xi}_i \rangle) \cdot \boldsymbol{\xi}_i[k]^2.$$

Note that $\ell_{-y_i,i}^{(t)} \sigma'(\langle \mathbf{w}_{-y_i,r}^{(t)}, \boldsymbol{\xi}_i \rangle) < 0$, which implies that $\mathbf{w}_{-y_i,r}^{(t)}[k] \cdot \boldsymbol{\xi}_i[k]$ is decreasing in $t$. Therefore, for all $r$ and $i$, we have

$$\langle \mathbf{w}_{-y_i,r}^{(t)}, \boldsymbol{\xi}_i \rangle = \mathbf{w}_{-y_i,r}^{(t)}[1] \cdot \boldsymbol{\xi}_i[1] + \sum_{k \in \mathcal{B}_i} \mathbf{w}_{-y_i,r}^{(t)}[k]\boldsymbol{\xi}_i[k]$$
$$\leq \mathbf{w}_{-y_i,r}^{(t)}[1] \cdot \boldsymbol{\xi}_i[1] + \sum_{k \in \mathcal{B}_i} \mathbf{w}_{-y_i,r}^{(0)}[k]\boldsymbol{\xi}_i[k]$$
$$\leq |\mathbf{w}_{-y_i,r}^{(t)}[1] \cdot \boldsymbol{\xi}_i[1]| + \left|\sum_{k \in \mathcal{B}_i} \mathbf{w}_{-y_i,r}^{(0)}[k]\boldsymbol{\xi}_i[k]\right|$$
$$\leq \widetilde{\Theta}(\alpha) + \widetilde{\Theta}(\sigma_0\sigma_p s^{1/2})$$
$$= \widetilde{\Theta}(\alpha),$$

where the third inequality follows by (C.33). This completes the proof. $\qquad\square$

Note that for different $j$, the iteration numbers when $\Lambda_j^{(t)}$ reaches $\widetilde{\Theta}(1/m)$ are different. Without loss of generality, we can assume $T_1 \leq T_{-1}$. Lemma C.8 has provided a clear understanding about how $\Lambda_j^{(t)}$ varies within the iteration range $[0, T_j]$. However, it remains unclear how $\Gamma_1^{(t)}$ varies within the iteration range $[T_1, T_{-1}]$ since in this period we no longer have $|\ell_{j,i}^{(t)}| = \Theta(1)$ and the effect of gradient descent on the feature learning (i.e., increase of $\langle \mathbf{w}_{j,r}, j \cdot \mathbf{v} \rangle$) becomes weaker. In the following lemma we give a characterization of $\Lambda_1^{(t)}$ for every $t \in [T_1, T_{-1}]$.

**Lemma C.11** (Stage I of GD: part II). Without loss of generality assuming $T_1 < T_{-1}$. Then it holds that $\Lambda_1^{(t)} = \widetilde{\Theta}(1)$ for all $t \in [T_1, T_{-1}]$.

*Proof.* Recall from (C.29) that we have the following general lower bound for the increase of $\Lambda_j^{(t)}$

$$\Lambda_j^{(t+1)} \geq (1 - \eta\lambda) \cdot \langle \mathbf{w}_{j,r^*}^{(t)}, j \cdot \mathbf{v} \rangle + \frac{\eta}{n} \cdot \left( \sum_{i=1}^{n} |\ell_{j,i}^{(t)}| \sigma'(\langle \mathbf{w}_{j,r^*}^{(t)}, y_i \mathbf{v} \rangle) - \alpha \sum_{i=1}^{n} |\ell_{j,i}^{(t)}| \sigma'(\langle \mathbf{w}_{j,r^*}^{(t)}, \boldsymbol{\xi}_i \rangle) \right)$$

$$\geq (1 - \eta\lambda)\Lambda_j^{(t)} + \Theta\left(\frac{\eta}{n}\right) \cdot \sum_{i:y_i=j} |\ell_{j,i}^{(t)}| \cdot \left(\Lambda_j^{(t)}\right)^{q-1} - \Theta(\alpha\eta) \cdot \left(\Gamma_j^{(t)} \vee \widetilde{\Theta}(\alpha)\right)^{q-1}, \quad (C.34)$$

where the last inequality is by Lemma C.10. Note that by Lemma C.8, we have $\Gamma_j^{(t)} = \widetilde{O}(\sigma_0)$ for all $t \leq T_{-1}$ and . Then the above inequality leads to

$$\Lambda_j^{(t+1)} \geq (1 - \eta\lambda)\Lambda_j^{(t)} + \Theta\left(\frac{\eta}{n}\right) \cdot \sum_{i:y_i=j} |\ell_{j,i}^{(t)}| \cdot \left(\Lambda_j^{(t)}\right)^{q-1} - \Theta(\alpha^q\eta), \quad (C.35)$$

where we use the fact that $\alpha = \omega(\sigma_0)$. The the remaining proof consists of two parts: (1) proving $\Lambda_j^{(t)} \geq \Theta(1/m) = \widetilde{\Theta}(1)$ and (2) $\Lambda_j^{(t)} \leq \Theta(\log(1/\lambda))$.

Without loss of generality we consider $j = 1$. Regarding the first part, we first note that Lemma C.8 implies that $\Lambda_1^{(T_1)} \geq \Theta(1/m)$. Then we consider the case when $\Lambda_1^{(t)} \leq \Theta(\log(1/\alpha)/m)$, it holds that for all $y_i = 1$,

$$\ell_{1,i}^{(t)} = \frac{e^{F_{-1}(\mathbf{W}^{(t)}, \mathbf{x}_i)}}{\sum_{j \in \{-1,1\}} e^{F_j(\mathbf{W}^{(t)}, \mathbf{x}_i)}}$$

$$= \exp\left(\Theta\left(\sum_{r=1}^{m} \left[\sigma(\langle \mathbf{w}_{-1,r}^{(t)}, y_i\mathbf{v}\rangle) + \sigma(\langle \mathbf{w}_{-1,r}^{(t)}, \boldsymbol{\xi}_i\rangle)\right] - \sum_{r=1}^{m} \left[\sigma(\langle \mathbf{w}_{1,r}^{(t)}, y_i\mathbf{v}\rangle) + \sigma(\langle \mathbf{w}_{1,r}^{(t)}, \boldsymbol{\xi}_i\rangle)\right]\right)\right)$$

$$\geq \exp\left(-\Theta(m\Lambda_1^{(t)})\right)$$

$$\geq \exp(-\Theta(\log(1/\alpha)))$$

$$= \widetilde{\Theta}(\alpha).$$

Then (C.35) implies that if $\Gamma_1^{(t)} \leq \Theta(\log(1/\sigma_0)/m)$, we have

$$\Lambda_1^{(t+1)} \geq (1 - \eta\lambda)\Lambda_1^{(t)} + \Theta(\eta\alpha) \cdot \Lambda_1^{(t)} - \Theta(\alpha^q\eta) \geq \Lambda_1^{(t)} + \Theta(\eta\alpha) \cdot \Lambda_1^{(t)} \geq \Lambda_1^{(t)},$$

where the second inequality is due to $\lambda = o(\alpha)$. This implies that $\Lambda_1^{(t)}$ will keep increases in this case so that it is impossible that $\Lambda_1^{(t)} \leq \Theta(1/m)$, which completes the proof of the first part.

For the second part, (C.28) implies that

$$\Lambda_1^{(t+1)} \leq (1 - \eta\lambda)\Lambda_1^{(t)} + \Theta\left(\frac{\eta}{n}\right) \cdot \sum_{i:y_i=1} |\ell_{1,i}^{(t)}| \cdot \left(\Lambda_1^{(t)}\right)^{q-1}. \quad (C.36)$$

Consider the case when $\Gamma_1^{(t)} \geq \Theta(\log(d))$, then for all $y_i = 1$,

$$\ell_{1,i}^{(t)} = \frac{e^{F_{-1}(\mathbf{W}^{(t)}, \mathbf{x}_i)}}{\sum_{j \in \{-1,1\}} e^{F_j(\mathbf{W}^{(t)}, \mathbf{x}_i)}}$$

$$= \exp\left(\Theta\left(\sum_{r=1}^{m} \left[\sigma(\langle \mathbf{w}_{-1,r}^{(t)}, y_i\mathbf{v}\rangle) + \sigma(\langle \mathbf{w}_{-1,r}^{(t)}, \boldsymbol{\xi}_i\rangle)\right] - \sum_{r=1}^{m} \left[\sigma(\langle \mathbf{w}_{1,r}^{(t)}, y_i\mathbf{v}\rangle) + \sigma(\langle \mathbf{w}_{1,r}^{(t)}, \boldsymbol{\xi}_i\rangle)\right]\right)\right)$$

$$\leq \exp\left(-\Theta(\Lambda_1^{(t)})\right)$$

$$\leq \exp(-\Theta(\log(1/\lambda))$$

$$= \widetilde{\Theta}(\text{poly}(\lambda)).$$

Then (C.36) further implies that

$$\Lambda_1^{(t+1)} \le (1 - \eta\lambda)\Lambda_1^{(t)} + \Theta\left(\frac{\eta}{\text{poly}(d)}\right) \cdot \left(\Lambda_1^{(t)}\right)^{q-1}$$

$$\le \Lambda_1^{(t)} - \Theta\left(\eta\Lambda_1^{(t)}\right) \cdot \left(\lambda - \text{poly}(\lambda) \cdot \left(\Lambda_1^{(t)}\right)^{q-2}\right) \le \Lambda_1^{(t)},$$

which implies that $\Lambda_1^{(t)}$ will decrease. As a result, we can conclude that $\lambda_1^{(t)}$ will not exceed $\Theta(\log(1/\lambda))$, this completes the proof of the second part.

$\square$

**Lemma C.12** (Lemma 5.7, restated). *If $\eta \le O(\sigma_0)$, it holds that $\Lambda_j^{(t)} = \widetilde{\Theta}(1)$ and $\Gamma_j^{(t)} = \widetilde{O}(\sigma_0)$ for all $t \in [T_{-1}, T]$.*

*Proof.* We will prove the desired argument based on the following three induction hypothesis:

$$\Lambda_j^{(t+1)} \ge (1 - \lambda\eta)\Lambda_j^{(t)} + \widetilde{\Theta}\left(\frac{\eta}{n}\right) \sum_{i:y_i=j} |\ell_{j,i}^{(t)}| - \widetilde{\Theta}(\alpha^q\eta) \cdot \frac{1}{n}\sum_{i=1}^{n} |\ell_{j,r}^{(t)}|, \qquad \text{(C.37)}$$

$$\Gamma_j^{(t)} = \widetilde{O}(\sigma_0), \qquad \text{(C.38)}$$

$$\Lambda_j^{(t)} = \widetilde{\Theta}(1). \qquad \text{(C.39)}$$

In terms of Hypothesis (C.37), we can apply Hypothesis (C.38) and (C.39) to (C.34) and get that

$$\Lambda_j^{(t+1)} \ge (1 - \eta\lambda)\Lambda_j^{(t)} + \Theta\left(\frac{\eta}{n}\right) \cdot \sum_{i:y_i=j} |\ell_{j,i}^{(t)}| \cdot \left(\Lambda_j^{(t)}\right)^{q-1} - \Theta(\alpha\eta) \cdot \left(\Gamma_j^{(t)} \vee \widetilde{\Theta}(\alpha)\right)^{q-1} \cdot \frac{1}{n}\sum_{i=1}^{n} |\ell_{j,r}^{(t)}|$$

$$\ge (1 - \lambda\eta)\Lambda_j^{(t)} + \widetilde{\Theta}\left(\frac{\eta}{n}\right) \sum_{i:y_i=j} |\ell_{j,i}^{(t)}| - \widetilde{\Theta}(\alpha^q\eta) \cdot \frac{1}{n}\sum_{i=1}^{n} |\ell_{j,r}^{(t)}|.$$

where the last inequality we use the fact that $\alpha \ge \sigma_0$. This verifies Hypothesis (C.37).

In order to verify Hypothesis (C.38), we have the following according to (C.37),

$$\sum_{j\in\{-1,1\}} \Lambda_j^{(t+1)} \ge (1 - \lambda\eta) \sum_{j\in\{-1,1\}} \left[\Lambda_j^{(t)} + \widetilde{\Theta}\left(\frac{\eta}{n}\right)\sum_{i=1}^{n} |\ell_{j,i}^{(t)}| - \widetilde{\Theta}(\alpha^q\eta) \cdot \frac{1}{n}\sum_{i=1}^{n} |\ell_{j,r}^{(t)}|\right]$$

$$= (1 - \lambda\eta) \sum_{j\in\{-1,1\}} \left[\Lambda_j^{(t)} + \widetilde{\Theta}\left(\frac{\eta}{n}\right)\sum_{i=1}^{n} |\ell_{j,i}^{(t)}|\right],$$

where the last equality holds since $\alpha = o(1)$. Recursively applying the above inequality from $T_{-1}$ to $t$ gives

$$\sum_{j\in\{-1,1\}} \Lambda_j^{(t)} \ge (1 - \lambda\eta)^{t-T_{-1}} \sum_{j\in\{-1,1\}} \left[\Lambda_j^{(T_{-1})} + \widetilde{\Theta}\left(\frac{\eta}{n}\right) \cdot \sum_{\tau=0}^{t-T_{-1}-1} (1 - \lambda\eta)^\tau \sum_{i=1}^{n} |\ell_{j,i}^{(t-1-\tau)}|\right].$$

Then by Hypothesis (C.39) we have

$$\widetilde{\Theta}\left(\frac{\eta}{n}\right) \cdot \sum_{\tau=0}^{t-T_{-1}-1} (1 - \lambda\eta)^\tau \sum_{i=1}^{n} |\ell_{j,i}^{(t-1-\tau)}| \le \widetilde{\Theta}(1).$$

Now let us look at the rate of memorizing noises. By (C.28) and use the fact that $\alpha^2 \leq O(s\sigma_p^2/n)$, we have

$$
\begin{aligned}
\Gamma_j^{(t)} &\leq (1-\eta\lambda)\Gamma_j^{(t-1)} + \widetilde{\Theta}\left(\frac{\eta s\sigma_p^2}{n}\right) \cdot \sum_{i=1} |\ell_{j,i}| \cdot \left(\Gamma_j^{(t-1)}\right)^{q-1} \\
&\leq (1-\eta\lambda)\Gamma_j^{(t-1)} + \widetilde{\Theta}\left(\frac{\eta s\sigma_p^2 \sigma_0^{q-1}}{n}\right) \cdot \sum_{i=1} |\ell_{j,i}| \\
&\leq \Gamma_j^{(T_{-1})} + \widetilde{\Theta}\left(\frac{\eta s\sigma_p^2 \sigma_0^{q-1}}{n}\right) \cdot \sum_{\tau=0}^{t-T_{-1}-1} (1-\lambda\eta)^\tau \sum_{i=1}^n |\ell_{j,i}^{(t-1-\tau)}| \\
&\leq \widetilde{\Theta}\left(\sigma_0 + s\sigma_p^2 \sigma_0^{q-1}\right) \\
&\leq \widetilde{\Theta}(\sigma_0),
\end{aligned}
$$

which verifies Hypothesis (C.38).

Given Hypothesis (C.37) and (C.38), the verification of (C.39) is straightforward by applying the same proof technique of Lemma C.11 and thus we omit it here. $\qquad\square$

**Lemma C.13** (Lemma 5.8, restated). If the step size satisfies, then for any $t \geq T_{-1}$ it holds that

$$
L(\mathbf{W}^{(t+1)}) - L(\mathbf{W}^{(t)}) \leq -\frac{\eta}{2}\|\nabla L(\mathbf{W}^{(t)})\|_F^2.
$$

*Proof.* The proof of this lemma is similar to that of Lemma C.6, which is basically relying the smoothness property of the loss function $L(\mathbf{W})$ given certain constraints on the inner products $\langle \mathbf{w}_{j,r}, \mathbf{v}\rangle$ and $\langle \mathbf{w}_{j,r}, \boldsymbol{\xi}_i\rangle$.

Let $\Delta F_{j,i} = F_j(\mathbf{W}^{(t+1)}, \mathbf{x}_i) - F_j(\mathbf{W}^{(t)}, \mathbf{x}_i)$, we can get the following Taylor expansion on the loss function $L_i(\mathbf{W}^{(t+1)})$,

$$
L_i(\mathbf{W}^{(t+1)}) - L_i(\mathbf{W}^{(t)}) \leq \sum_j \frac{\partial L_i(\mathbf{W}^{(t)})}{\partial F_j(\mathbf{W}^{(t)}, \mathbf{x}_i)} \cdot \Delta F_{j,i} + \sum_j (\Delta F_{j,i})^2. \tag{C.40}
$$

In particular, by Lemma C.12, we know that $\langle \mathbf{w}_{j,r}^{(t)}, y_i\mathbf{v}\rangle \leq \widetilde{\Theta}(1)$ and $\langle \mathbf{w}_{j,r}^{(t)}, \boldsymbol{\xi}_i\rangle \leq \widetilde{\Theta}(\sigma_0) \leq \widetilde{\Theta}(1)$. Then similar to (C.21), we can apply first-order Taylor expansion to $F_j(\mathbf{W}^{(t+1)}, \mathbf{x}_i)$, which requires to characterize the second-order error of the Taylor expansions on $\sigma(\langle \mathbf{w}_{j,r}^{(t+1)}, y_i\mathbf{v}\rangle)$ and $\sigma(\langle \mathbf{w}_{j,r}^{(t+1)}, \boldsymbol{\xi}_i\rangle)$,

$$
\begin{aligned}
&\left|\sigma(\langle \mathbf{w}_{j,r}^{(t+1)}, y_i\mathbf{v}\rangle) - \sigma(\langle \mathbf{w}_{j,r}^{(t)}, y_i\mathbf{v}\rangle) - \langle \nabla_{\mathbf{w}_{j,r}}\sigma(\langle \mathbf{w}_{j,r}^{(t)}, y_i\mathbf{v}\rangle), \mathbf{w}_{j,r}^{(t+1)} - \mathbf{w}_{j,r}^{(t)}\rangle\right| \\
&\leq \widetilde{\Theta}\left(\|\mathbf{w}_{j,r}^{(t+1)} - \mathbf{w}_{j,r}^{(t)}\|_2^2\right) = \widetilde{\Theta}(\eta^2 \|\nabla_{\mathbf{w}_{j,r}}L(\mathbf{W}^{(t)})\|_2^2), \\
&\left|\sigma(\langle \mathbf{w}_{j,r}^{(t+1)}, \boldsymbol{\xi}_i\rangle) - \sigma(\langle \mathbf{w}_{j,r}^{(t)}, \boldsymbol{\xi}_i\rangle) - \langle \nabla_{\mathbf{w}_{j,r}}\sigma(\langle \mathbf{w}_{j,r}^{(t)}, \boldsymbol{\xi}_i\rangle), \mathbf{w}_{j,r}^{(t+1)} - \mathbf{w}_{j,r}^{(t)}\rangle\right| \\
&\leq \widetilde{\Theta}\left(\|\mathbf{w}_{j,r}^{(t+1)} - \mathbf{w}_{j,r}^{(t)}\|_2^2\right) = \widetilde{\Theta}(\eta^2 \|\nabla_{\mathbf{w}_{j,r}}L(\mathbf{W}^{(t)})\|_2^2). \tag{C.41}
\end{aligned}
$$

Then combining the above bounds for every $r \in [m]$, we can get the following bound for $\Delta F_{j,i}$

$$
\begin{aligned}
\left|\Delta F_{j,i} - \langle \nabla_{\mathbf{W}} F_j(\mathbf{W}^{(t)}, \mathbf{x}_i), \mathbf{W}^{(t+1)} - \mathbf{W}^{(t)}\rangle\right| &\leq \widetilde{\Theta}\left(\eta^2 \sum_{r\in[m]} \|\nabla_{\mathbf{w}_{j,r}}L(\mathbf{W}^{(t)})\|_2^2\right) \\
&= \widetilde{\Theta}\left(\eta^2 \|\nabla L(\mathbf{W}^{(t)})\|_F^2\right). \tag{C.42}
\end{aligned}
$$

Moreover, since $\langle \mathbf{w}_{j,r}^{(t)}, y_i\mathbf{v}\rangle \leq \widetilde{\Theta}(1)$ and $\langle \mathbf{w}_{j,r}^{(t)}, \boldsymbol{\xi}_i\rangle \leq \widetilde{\Theta}(1)$ and $\sigma(\cdot)$ is convex, then we have

$$
\begin{aligned}
|\sigma(\langle \mathbf{w}_{j,r}^{(t+1)}, y_i\mathbf{v}\rangle) - \sigma(\langle \mathbf{w}_{j,r}^{(t)}, y_i\mathbf{v}\rangle)| &\leq \max\left\{|\sigma'(\langle \mathbf{w}_{j,r}^{(t+1)}, y_i\mathbf{v}\rangle)|, |\sigma'(\langle \mathbf{w}_{j,r}^{(t)}, y_i\mathbf{v}\rangle)|\right\} \cdot |\langle \mathbf{v}, \mathbf{w}_{j,r}^{(t+1)} - \mathbf{w}_{j,r}^{(t)}\rangle| \\
&\leq \widetilde{\Theta}\left(\|\mathbf{w}_{j,r}^{(t+1)} - \mathbf{w}_{j,r}^{(t)}\|_2\right).
\end{aligned}
$$

Similarly we also have

$$|\sigma(\langle \mathbf{w}_{j,r}^{(t+1)}, \boldsymbol{\xi}_i \rangle) - \sigma(\langle \mathbf{w}_{j,r}^{(t)}, \boldsymbol{\xi}_i \rangle)| \leq \widetilde{\Theta}(\|\mathbf{w}_{j,r}^{(t+1)} - \mathbf{w}_{j,r}^{(t)}\|_2).$$

Combining the above inequalities for every $r \in [m]$, we have

$$|\Delta F_{j,i}|^2 \leq \widetilde{\Theta}\left(\left[\sum_{r \in [m]} \|\mathbf{w}_{j,r}^{(t+1)} - \mathbf{w}_{j,r}^{(t)}\|_2\right]^2\right) \leq \widetilde{\Theta}(m\eta^2 \|\nabla L(\mathbf{W}^{(t)})\|_F^2) = \widetilde{\Theta}(\eta^2 \|\nabla L(\mathbf{W}^{(t)})\|_F^2). \tag{C.43}$$

Now we can plug (C.42) and (C.43) into (C.40), which gives

$$L_i(\mathbf{W}^{(t+1)}) - L_i(\mathbf{W}^{(t)}) \leq \sum_j \frac{\partial L_i(\mathbf{W}^{(t)})}{\partial F_j(\mathbf{W}^{(t)}, \mathbf{x}_i)} \cdot \Delta F_{j,i} + \sum_j (\Delta F_{j,i})^2$$

$$= \langle \nabla L_i(\mathbf{W}^{(t)}), \mathbf{W}^{(t+1)} - \mathbf{W}^{(t)} \rangle + \widetilde{\Theta}(\eta^2 \|\nabla L(\mathbf{W}^{(t)})\|_F^2). \tag{C.44}$$

Taking sum over $i \in [n]$ and applying the smoothness property of the regularization function $\lambda \|\mathbf{W}\|_F^2$, we can get

$$L(\mathbf{W}^{(t+1)}) - L(\mathbf{W}^{(t)}) = \frac{1}{n} \sum_{i=1}^n \left[L_i(\mathbf{W}^{(t+1)}) - L_i(\mathbf{W}^{(t)})\right] + \lambda\left(\|\mathbf{W}^{(t+1)}\|_F^2 - \|\mathbf{W}^{(t)}\|_F^2\right)$$

$$\leq \langle \nabla L(\mathbf{W}^{(t)}), \mathbf{W}^{(t+1)} - \mathbf{W}^{(t)} \rangle + \widetilde{\Theta}(\eta^2 \|\nabla L(\mathbf{W}^{(t)})\|_F^2)$$

$$= -\left(\eta - \widetilde{\Theta}(\eta^2)\right) \cdot \|\nabla L(\mathbf{W}^{(t)})\|_F^2$$

$$\leq -\frac{\eta}{2} \|\nabla L(\mathbf{W}^{(t)})\|_F^2,$$

where the last inequality is due to our choice of step size $\eta = o(1)$ so that gives $\eta - \widetilde{\Theta}(\eta^2) \geq \eta/2$. This completes the proof. $\qquad\square$

**Lemma C.14** (Generalization Performance of GD). Let

$$\mathbf{W}^* = \arg\min_{\{\mathbf{W}^{(1)}, \ldots, \mathbf{W}^{(T)}\}} \|\nabla L(\mathbf{W}^{(t)})\|_F.$$

Then for all training data, we have

$$\frac{1}{n} \sum_{i=1}^n \mathbb{1}\left[F_{y_i}(\mathbf{W}^*, \mathbf{x}_i) \leq F_{-y_i}(\mathbf{W}^*, \mathbf{x}_i)\right] = 0.$$

Moreover, in terms of the test data $(\mathbf{x}, y) \sim \mathcal{D}$, we have

$$\mathbb{P}_{(\mathbf{x},y)\sim\mathcal{D}}\left[F_y(\mathbf{W}^*, \mathbf{x}) \leq F_{-y}(\mathbf{W}^*, \mathbf{x})\right] = o(1).$$

*Proof.* By Lemma C.12 it is clear that all training data can be correctly classified so that the training error is zero. Besides, for test data $(\mathbf{x}, y)$ with $\mathbf{x} = [y\mathbf{v}^\top, \boldsymbol{\xi}^\top]^\top$, it is clear that with high probability $\langle \mathbf{w}_{y,r}^*, y\mathbf{v} \rangle = \widetilde{\Theta}(1)$ and $[\langle \mathbf{w}_{y,r}^*, \boldsymbol{\xi} \rangle]_+ \leq \widetilde{O}(\sigma_0)$, then

$$F_y(\mathbf{W}^*, \mathbf{x}) = \sum_{r=1}^m \left[\sigma(\langle \mathbf{w}_{y,r}^*, y\mathbf{v} \rangle) + \sigma(\langle \mathbf{w}_{y,r}^*, \boldsymbol{\xi} \rangle)\right] \geq \widetilde{\Omega}(1).$$

If $j = -y$, we have with probability at least $1 - 1/\text{poly}(n)$, $\langle \mathbf{w}_{-y,r}^*, y\mathbf{v} \rangle \leq 0$ and $[\mathbf{w}_{-y,r}^*, \boldsymbol{\xi} \rangle]_+ \leq \widetilde{O}(\alpha)$, which leads to

$$F_{-y}(\mathbf{W}^*, \mathbf{x}) = \sum_{r=1}^m \left[\sigma(\langle \mathbf{w}_{-y,r}^*, y\mathbf{v} \rangle) + \sigma(\langle \mathbf{w}_{-y,r}^*, \boldsymbol{\xi} \rangle)\right] \leq \widetilde{O}(m\alpha^q) = \widetilde{O}(\alpha^q) = o(1).$$

This implies that GD can also achieve nearly at most $1/\text{poly}(n)$ test error. This completes the proof. $\qquad\square$

## D  PROOF OF THEOREM 4.2: CONVEX CASE

**Theorem D.1** (Convex setting, restated)**.** Assume the model is over-parameterized. Then for any convex and smooth training objective with positive regularization parameter $\lambda$, suppose we run **Adam** and **gradient descent** for $T = \frac{\text{poly}(n)}{\eta}$ iterations, then with probability at least $1 - n^{-1}$, the obtained parameters $\mathbf{W}^*_{\text{Adam}}$ and $\mathbf{W}^*_{\text{GD}}$ satisfy that $\|\nabla L(\mathbf{W}^*_{\text{Adam}})\|_1 \leq \frac{1}{T\eta}$ and $\|\nabla L(\mathbf{W}^*_{\text{Adam}})\|^2_2 \leq \frac{1}{T\eta}$ respectively. Moreover, it holds that:

- Training errors are the same:

$$\frac{1}{n} \sum_{i=1}^n \mathbb{1}\left[\text{sgn}\big(F(\mathbf{W}^*_{\text{Adam}}, \mathbf{x}_i)\big) \neq y_i\right] = \frac{1}{n} \sum_{i=1}^n \mathbb{1}\left[\text{sgn}\big(F(\mathbf{W}^*_{\text{GD}}, \mathbf{x}_i)\big) \neq y_i\right].$$

- Test errors are nearly the same:

$$\mathbb{P}_{(\mathbf{x},y)\sim\mathcal{D}}\left[\text{sgn}\big(F(\mathbf{W}^*_{\text{Adam}}, \mathbf{x}_i)\big) \neq y\right] = \mathbb{P}_{(\mathbf{x},y)\sim\mathcal{D}}\left[\text{sgn}\big(F(\mathbf{W}^*_{\text{GD}}, \mathbf{x})\big) \neq y\right] \pm o(1).$$

*Proof.* The proof is straightforward by applying the same proof technique used for Lemmas C.6 and C.13, where we only need to use the smoothness property of the loss function. Then it is clear that both Adam and GD can provably find a point with a sufficiently small gradient. Note that the training objective becomes strongly convex when adding weight decay regularization, implying that the entire training objective only has one stationary point, i.e., point with a sufficiently small gradient. This further implies that the points found by Adam and GD must be exactly the same and thus GD and Adam must have nearly the same training and test performance.

Besides, when the problem is sufficiently over-parameterized, with proper regularization (feasibly small), we can still guarantee zero training errors. $\qquad\square$

## E  DISCUSSION ON THE DATA MODELS IN WILSON ET AL. (2017); REDDI ET AL. (2018)

**Data model in Wilson et al. (2017).** In particular, given the binary label $y_i \in \{-1, 1\}$, the feature vector $\mathbf{x}_i$ is set as

$$\mathbf{x}_i[j] = \begin{cases} y_i, & j = 1 \\ 1, & j = 2, 3 \\ 1, & j = 4 + 5(i-1), \ldots, 4 + 5(i-1) + 2(1 - y_i) \\ 0, & \text{otherwise.} \end{cases}$$

**Data model in Reddi et al. (2018).** In particular, Reddi et al. (2018) considers a one-dimensional optimization objective. Besides, in each iteration of Adam, the stochastic gradient is taken based on the function $f_t(x)$ defined as follows:

$$f_t(x) = \begin{cases} Cx, & t \mod 3 = 1 \\ -x, & \text{otherwise.} \end{cases}$$

Then it can be seen that in these two prior works, each coordinate of the feature vector (or the objective function) is hard coded. In contrast, our data model allows randomness in the data generation process. This implies that our theory can hold for the data points generated from a certain distribution, while these prior works can only cover one particular data or optimization objective.

