# OpenReview forum: "Understanding the Generalization of Adam in Learning Neural Networks with Proper Regularization"
_ICLR.cc/2023/Conference — ICLR 2023 poster_

### Official Review · Reviewer_52yW · 2022-10-22

**Confidence:** 2
**Correctness:** 4
**Technical Novelty And Significance:** 3
**Empirical Novelty And Significance:** 1
**Recommendation:** 6

**Clarity, Quality, Novelty And Reproducibility:**

The presentation of the paper is pretty clean to me. The overall quality of the draft is great.


**Strength And Weaknesses:**

Strength: It looks like the first work to show the finite sample generalization guarantee on the local optima of a CNN trained using ADAM as compared to GD. I guess one reason we haven’t seen such results is that in general it is not easy to give rigorous finite sample guarantees on ADAM’s generalization errors for the non-convex neural networks. In that sense, this paper has its unique value.

Weakness: Even though I understand without the assumptions it is hard to provide rigorous proof, I would still say the assumptions and simplifications made in the paper are too much. Those include
1) two layer CNN but the second layer is fixed to all 1s
2) the simplifications on the input vectors seem weird. 1-sparse looks over simplified to me, and in general the noise comes together with the input signal instead of coming from a totally different dimension.
3) the width of the network is assumed to be polylogarithmic in the training sample size. Even though over-parameterization is used a lot, usually people do not assume the model’s width grows with the sample size in applications
4) The work only gives results on full batch ADAM and full batch GD. However in real applications full batch optimization is rarely used.


**Summary Of The Paper:**

This paper proves under some particular situation, e.g. the input vector is the concatenation of a 1-sparse feature vector and a noise vector, under certain assumptions, the two layer CNN (whose second layer is assumed to be fixed as all 1’s), converges to “bad” local optima with high probability when trained using ADAM. However under the same setting, when trained using gradient descent, with high probability the final convergence is a relatively “good” local optimum. The authors also show for the convex settings such as linear models + logistic loss, both ADAM and gradient descent converge to “good” local optima with high probability.


**Summary Of The Review:**

I have mixed feelings about this paper. On one hand it is a solid move towards understanding the generalization capability of ADAM and GD. On the other hand, the assumptions and simplifications made in the paper have made me less excited. The models and data discussed in the draft are too far away from the real models and data used in applications, so that I am hesitating to use the insights from the paper in any real applications.

---

> ### Author Response · Authors · 2022-11-15
> **Response to Reviewer 52yW**
>
> Thank you for your positive comments.
>
> **Q1:** “two layer CNN but the second layer is fixed to all 1s”
>
> **A1:** As we have claimed in Section 2, this assumption is made for the simplicity of analysis, which can be relaxed with extra efforts. Besides, we set the second layer to be in the same order as this helps better characterize the learning dynamics of the hidden layer weights.
>
> ---
>
> **Q2:** "the simplifications on the input vectors seem weird. 1-sparse looks over simplified to me, and in general the noise comes together with the input signal instead of coming from a totally different dimension."
>
> **A2:** While 1-sparsity is indeed an over-simplified case, it does not lose any generality. In fact, this assumption can be relaxed as we only need to guarantee a separation between Adam and GD in terms of feature learning and noise memorization. In detail, this can be achieved if the feature vector has a smaller $\ell_1$ norm and larger $\ell_2$ norm than noise vectors, which can be held as long as the feature vector is sparser than the noise vectors (i.e., $k = o(s/\mathrm{polylog}(n))$, where $k$ and $s$ are the sparsity of feature and noise vectors respectively).
> Besides, we also allow the noise to come together with the feature, as long as the overlapped component is sufficiently small compared to the remaining noise components (that are not overlapped with the feature).
>
> ---
>
> **Q3:** “the width of the network is assumed to be polylogarithmic in the training sample size. Even though over-parameterization is used a lot, usually people do not assume the model’s width grows with the sample size in applications
>
> **A3:** We would like to clarify that we use polylogarithmic width in order to guarantee in the initialization, with high probability (greater than $1-\exp(-\Theta(m))=1-1/\mathrm{poly}(n)$), some of the neurons can be positively correlated with the feature vector $v$, so that the feature vector can be learned during the later training stage. We can also set the width to be some large enough constant at the price of deriving a constant probability guarantee, e.g., $>1-e^{-1}$ instead of a high probability guarantee.
>
> ---
>
> **Q4:** "The work only gives results on full batch ADAM and full batch GD. However in real applications full batch optimization is rarely used.’"
>
> **A4:** We have discussed how to extend our theoretical analysis to stochastic gradient Adam and SGD in Appendix B, where we show that (1) the feature learning and noise memorization of SGD will be similar to GD; and (2) the feature learning of stochastic gradient Adam will be similar to full-batch Adam, while the noise memorization speed will be slowed down by a factor of $n/B$, where $B$ is the minibatch size. Thus, based on Lemmas B.1 \& B.2 (feature learning and noise memorization of SGD and stochastic gradient Adam in Stage I), we can still show the separation between SGD and stochastic gradient Adam if the dimension is large enough (i.e., requiring $s\sigma_p = \Theta(d^{1/4}/(n\mathrm{polylog}(n)))\gg n/B$).

---

> ### Author Response · Authors · 2022-11-18
> **Follow up with Reviewer 52yW**
>
> Dear reviewer, since the paper revision phase is ending soon, we would like to follow up with you to see if our response has addressed your concerns. We have also revised Appendix B (which discusses how to extend the results to stochastic gradient Adam and SGD) by summarizing the feature learning and noise memorization results of stochastic gradient Adam and SGD in stage I. Based on these results, we also show that the separation between SGD and stochastic gradient Adam still exists under a relatively stronger over-parameterization condition.
>
> We look forward to your feedback and would be happy to address any remaining concerns. Thank you!

---

### Official Review · Reviewer_oAui · 2022-10-24

**Confidence:** 3
**Correctness:** 3
**Technical Novelty And Significance:** 3
**Empirical Novelty And Significance:** 2
**Recommendation:** 6

**Clarity, Quality, Novelty And Reproducibility:**

For writing, most parts of this work are well written and clear.

For the contribution parts,  since there are gaps on both network architectures, data assumptions, and algorithms, the analysis of this work actually does not match the practical setting and may not well reflect the truth of practical applications.


**Strength And Weaknesses:**

Strengths:
There are main three contributions in this work.
1) On a well-designed two-layer network,  Adam often converges to a bad minimum while GD can converge to a good solution in terms of generalization error.


2) For convex problem, with the weight decay, Adam and GD converge to the same solution.

3)The above two results show the different behaviors of Adam and GD and may provide some insights for their deep learning application.

Weaknesses:
1)The analysis in this work heavily depends on the well-designed network and data assumptions. For network, this work designs a special two-layered network which is shallow and also uses a rarely used  polynomial RELU. This network has a big gap with the real network. For data, it assumes the feature is 1 sparse while the noise is also s-sparse. For feature, it is too sparse to hold in practice. Though the authors claim their results still hold when feature is k-sparse, as long as the sparsity gap between feature and noises exit. However, they did not explain more how sparsity gap is required and whether it is reasonable in practice setting.  So it is a big concern whether this analysis with these well-designed network and data can really reflect the performance for network training in practical applications.


2)In deep learning, one often SGD and stochastic Adam to train network. But this work analyzes GD and  deterministic Adam which yields a gap. Though the authors also say their results can be extended to stochastic setting but did not give many details. It is better to directly put the results under the stochastic setting in the paper, if stochastic setting can be well handled, since one often cares more about the real settings.


**Summary Of The Paper:**

This work mainly focuses on the generalization comparison between Adam and GD. Specifically,   for a well-designed two-layer network,  Adam often converges to a bad minimum while GD can converge to a good solution in terms of generalization error, even with weight decay regularization. For convex problem, with the weight decay, Adam and GD converge to the same solution.

**Summary Of The Review:**

Overall, this work provides some good initial results. But its assumptions may not be consistent with the real settings.

---

> ### Author Response · Authors · 2022-11-15
> **Response to Reviewer oAui**
>
> Thanks for your helpful comments.
>
> ---
>
> **Q1:** “For network, this work designs a special two-layered network which is shallow and also uses a rarely used polynomial RELU.”
>
> **A1:** Two-layer networks have been widely studied in the related literature, including but not limited to [Du et al., 2019b, Bai and Lee, 2019, Ji and Telgarsky, 2020]. It is believed that a comprehensive understanding of the two-layer network serves as the first step for studying deeper neural networks.
>
> Besides, the polynomial ReLU (or smoothed ReLU, which, as mentioned in our paper, is also applicable in our setting)  activation functions have also been widely considered to study the generalization performance of two-layer neural networks. For example, [Frei et al., 2022] studies the benign overfitting of two-layer neural network with a (quadratic function enabled) smoothed leaky ReLU activation; the same topic has also been studied in [Cao et al., 2022] on a two-layer CNN with the exactly same activation function considered in our paper; [Shen et al., 2022] studies the role of data augmentation in deep learning based on a two-layer CNN with a ($q$-degree polynomial based) smoothed symmetrized ReLU activation; [Chen et al., 2022] studies the performance of mixture-of-experts in deep learning based on a two-layer CNN with the cubic activation function.
>
> - Frei et al., Benign overfitting without linearity: Neural network classifiers trained by gradient descent for noisy linear data. COLT 2022
> - Shen et al., Data augmentation as feature manipulation. ICML 2022
> - Cao et al., Benign overfitting in two-layer convolutional neural networks. NeurIPS 2022
> - Chen et al., Towards understanding mixture of experts in deep learning. NeurIPS 2022
>
>
> Due to the above reasons, we believe a two-layer network with the polynomial ReLU is a meaningful setting and good starting point to understand the generalization performance of Adam vs GD. We have also added these related works in Section 3 to justify the usage of polynomial ReLU activations.
>
> ---
>
> **Q2:** For data, it assumes the feature is 1 sparse while the noise is also s-sparse. ... they did not explain more how sparsity gap is required and whether it is reasonable in practice setting.
>
> **A2:** Technically, we only need the feature vector to have a smaller $\ell_1$ norm than that of the noise vectors to make our results hold. In particular, the inequalities in Lemma 5.2, which characterize the speed of feature learning and noise memorization of Adam in Stage I, can also be written as
> $$
> \langle w_{j,r}^{(t+1)}, j\cdot v\rangle\le \langle w_{j,r}^{(t)}, j\cdot v\rangle + \eta\cdot \Theta(\\|v\\|_1)\\
> $$
>
> $$
> \langle w_{j,r}^{(t+1)}, j\cdot \xi_s\rangle\le \langle w_{j,r}^{(t)}, j\cdot \xi_s\rangle + \eta\cdot \Theta(\\|\xi_s\\|_1).
> $$
>
> Adam will still prefer noise memorization as long as $\\|v\\|_1 \ll \\|\xi_s\\|_1$. Back to the sparsity of the feature (i.e., $k$), we can guarantee $\\|v\\|_1 \ll \\|\xi_s\\|_1$ if $k=o(s/\mathrm{polylog}(n))$ with $\\| v\\|_2=1$. To show this, we have $\\|v\\|_1\le k^{1/2}\\|v\\|_2 = o(s^{1/2}/\mathrm{polylog}(n))$. Then note that $\\|\xi_s\\|_1 = \Theta(s\cdot\sigma_p) = \Theta(s^{1/2}/\mathrm{polylog})(n)$, according to our setup in Definition 2.1, we can immediately get that $\\|v\\|_1 \ll \\|\xi_s\\|_1$ and our current theoretical results can still hold. Therefore, the $k=1$ sparsity of the feature vector can be largely relaxed.
>
> ---
>
> **Q3:** “In deep learning, one often SGD and stochastic Adam to train network. ... if stochastic setting can be well handled, since one often cares more about the real settings.”
>
> **A3:** We have added more details about the stochastic gradient setting in the revision. In particular, we have summarized the feature learning and noise memorization dynamics of SGD and stochastic gradient Adam in Stage 1 in Lemma B.1 and Lemma B.2 respectively. It can be seen that within each epoch ($n/B$ iterations, where $B$ is the batch size), stochastic gradient Adam will perform $O(\eta n/B)$ feature learning and $\tilde \Theta(\eta s\sigma_p)$ noise memorization, while full-batch Adam will perform $O(\eta n/B)$ feature learning and $\tilde\Theta(\eta ns\sigma_p/B)$ noise memorization according to Lemma 5.2. Then stochastic gradient Adam will still tend to memorize the noise rather than learning the feature vector if $s\sigma_p \gg n/B$, which can be satisfied if the problem dimension is sufficiently high. Besides, by comparing Lemma B.1 to (C.31) and (C.32), we can get that the noise memorization and feature learning of SGD will still be similar to those of full-batch GD after each $n/B$ iterations. Therefore, our current theoretical results for GD will still hold.
>
>  To this end, we believe that studying the full-batch version of GD and Adam will be more suitable as a starting point, as the difference in terms of the algorithm biases and the effect of regularization can be demonstrated in a clearer manner.

---

> > ### Comment · Reviewer_oAui · 2022-11-30
> > **Thanks for Response**
> >
> > Thanks for the response. After reading it, my biggest concerns have been resolved. So I raise the score from 5 to 6.

---

> ### Author Response · Authors · 2022-11-18
> **Follow up with Reviewer oAui**
>
> Dear reviewer, since the paper revision phase is ending soon, we would like to follow up with you to see if our response has addressed your concerns. We have also updated the paper as follows:
>
> - We have added more related works to clarify the usage of the polynomial ReLU activation function.
> - We have summarized the feature learning and noise memorization results in stage I for SGD and stochastic gradient Adam in Appendix B (Lemma B.2).
>
> We look forward to your feedback and would be happy to address any remaining concerns. Thank you!

---

### Official Review · Reviewer_1Tav · 2022-10-25

**Confidence:** 4
**Correctness:** 4
**Technical Novelty And Significance:** 3
**Empirical Novelty And Significance:** Not applicable
**Recommendation:** 6

**Clarity, Quality, Novelty And Reproducibility:**

I have commented on these aspects in the previous section about strengths and weakness, but here is a summary:

* Clarity: the paper is well-written in general, and I especially appreciate that the text nicely supports the mathematical derivations.
* Quality: without having reviewed in detail the whole appendix, the theoretical analysis seemed sound and solid to me, and the conclusions reflect the results.
* Novelty: I am not an expert in the theoretical analysis of Adam, but I suspect that some of the conclusions might have been reached and discussed before, perhaps by different means, since they seem intuitive to me. However, I am not providing a large weight to novelty in my review.
* Reproducibility: the paper contains only a small set of experimental results. Some of them contain a sufficient level of detail (as in Appendix A), but some lack sufficient details (Figure 1). I am not aware of any code accompanying this submission.

**Strength And Weaknesses:**

### Strengths

As main strength, I highlight that this paper is technically solid, mostly well-written and easy to follow. I value that introduction is clearly written, it includes motivations for the work and a concise review of the relevant related work. Further, both the methodology and the main results are discussed with sufficient detail, and I value that the significance of the mathematical results are unpacked nicely in the text, for example in Section 5. Finally, I also appreciate that, in the last section of the paper, some of the limitations of the paper are transparently mentioned as future work, for instance the limitation of the analysis to a two-layer neural network.

### Weaknesses

In my opinion, the main weakness of this paper is that the limitations adopted to enable a solid and conclusive analysis may impact the relevance and generalisation of the conclusions for practical applications. In particular, I identify two main limitations:

* The use of a highly synthetic data model (almost as simple as it can get), crafted to highlight the disadvantages of Adam with respect to gradient descent.
* The dependence of the results on a two-layer neural network.

The first limitation is directly connected with the data. Data sets used in practical applications vastly differ from the data used to derive the theoretical results in this paper, which consists of the combination of a 1-sparse feature vector and noise vector. This choice highly simplifies the analysis, but it leaves doubts to whether the conclusions can be extended to more complex data. As a matter of fact, the authors point out that their "theoretical analysis can lead to an opposite conclusion on the generalization comparison between Adam and GD if the noise is sparse and feature is denser"

The second limitation is also important, as it restricts the relevance of the conclusions to shallow models (2-layer neural networks), whereas Adam was born as an optimisation algorithm for deep neural networks and it is in this realm where it has been found to be superior--- according to certain aspects such as convergence rate, lower dependence on hyper-parameter optimisation, as well as generalisation in some cases---to gradient descent.

While I value the contribution of an analysis with a set of limitations to enable detailed conclusions, I think it is fair to highlight that the significance of the results are limited by such assumptions.

Some more minor weaknesses I would like to point out are the following:

* One of the main conclusions highlighted by the authors in the paper is that Adam and SGD converge to the same solution in the convex case, but differ in the non-convex case. That both algorithms are able to find the global optimum in convex optimisation (with weight decay) is intuitive (the differences lie in the convergence rate), and that the two algorithms are most different with a non-convex objective is also expected. Therefore, it is unclear how much technical novelty this conclusion is able to offer.
* Figure 1 shows visualisations of the features learnt by the first layer of AlexNet trained with Adam and SGD on CIFAR-10. However, no further details are provided beyond those mentioned in the capture of the figure. This visualisation, in and on itself without further details and analysis, seems to me like a rather superficial result, disconnected from the rest of the paper which is focused on theoretical results.
* In Section 3, the authors claim that the data model used in the paper is "more practical" than that in Wilson et al. (2017) and Reddi  et al. (2018). In this regard, I missed a more detailed discussion of the justification of the data model used in this paper, as well as the differences with models used in previous works analysing Adam. Furthermore, it is unclear to me how the data model in this paper can be "more practical", while it is an extreme case where the feature feature is a 1-sparse vector.
* The last item in the review of related work (Feature learning by neural networks) lacks diversity, since only works from the same group of authors is cited. As a matter of fact, citations of Allen-Zhu & Li are over-represented in the list of references.
* While the paper is well-written in general, some aspects could be improved in my opinion:
    * Some sentences contain long list of citations, which makes them unreadable. For instance, see the paragraph "Optimization and generalization guarantees in deep learning"
    * Typos or writing issues:
        * "In this section, we discuss the works that are mostly related to our paper": the word "mostly" sounds strange to me in this sentence.
        * The word "data" is used several times in singular, where I would instead use "data point". For instance: Definition 3.1, after Equation 3.2, second paragraph in Section 5.1
        * patter -> pattern (in last paragraph (Proof outline) of the introduction of Section 5)

**Summary Of The Paper:**

This paper studies the generalisation properties of Adam compared to gradient descent (GD). In order to simplify the analysis and isolate some distinguishing hypothesised differences between the two optimisation variants, the authors make use of a data model consisting of simple, sparse feature vectors and noise. The paper then provides a theoretical analysis in terms of convergence, generalisation, feature learning and noise memorisation, using a 2-layer neural network trained with weight decay regularisation. The main results are that both Adam and SGD behave similar in the convex case, but in the convex case they converge to different global solutions, Adam tending to fit the noise in the data.

**Summary Of The Review:**

My overall impression of this paper is positive, since it is a clearly written, solid analysis of relevant optimisation algorithms such as gradient descent and Adam. While I think the set of assumptions to carry out the analysis (shallow network and simple data model) limit the relevance of the results, I still think that the contributions in this paper advance our understanding of Adam and gradient descent.

---

> ### Author Response · Authors · 2022-11-15
> **Response to Reviewer 1Tav**
>
> Thank you for your constructive and valuable comments.
>
> ---
>
> **Q1:** “The first limitation is directly connected with the data. Data sets used in practical applications vastly differ from the data used to derive the theoretical results in this paper, ...  the authors point out that their "theoretical analysis can lead to an opposite conclusion on the generalization comparison between Adam and GD if the noise is sparse and feature is denser”
>
> **A1:** First of all, we consider data-dependent analysis an advantage instead of a disadvantage of our work. The performance of Adam indeed highly depends on the datasets, and this is even true for most deep learning models and algorithms. So there is no hope to understand deep learning without considering the underlying data. Second, as you mentioned, we consider the $1$-sparse feature vector for simplifying the analysis. In fact, this assumption can be relaxed as long as the feature vector is sparser than noise vectors. Generally speaking, Adam will tend to learn dense components, thus it will still prefer memorizing noise as long as the noise is denser. While the sparsity of feature and noise is flipped, Adam will tend to learn the feature rather than the noise, and this is why such an opposite conclusion can be drawn in Section 6.
>
> ---
>
> **Q2:** “The second limitation is also important, as it restricts the relevance of the conclusions to shallow models (2-layer neural networks), …”
>
> **A2:** We agree that studying the generalization of Adam and GD for training DNNs is closer to the practical setting. However, even in a simpler 2-layer NN setting, there is no such kind of comparison before our work. Without a comprehensive understanding on a simpler model — 2-layer NNs, it seems rather challenging or even impossible to perform a rigorous comparison between Adam and GD for deeper neural network models. Our paper is the first to theoretically compare the generalization performance of Adam v.s. GD in training neural networks, which we believe is an important step along this research direction.
>
> ---
>
> **Q3:** “One of the main conclusions highlighted by the authors in the paper is that Adam and SGD converge to the same solution in the convex case, but differ in the non-convex case. …”
>
> **A3:** While it is intuitive to imagine that different algorithms can find different solutions for nonconvex optimization, our paper provides a nontrivial comparison in this aspect by showing that these two different solutions have different generalization performances: the solution found by GD is well aligned with the feature while the solution found by Adam is the composition of noise vectors of the training data. We believe this is one of the key messages that need to be highlighted.
>
> ---
>
> **Q4:** “Figure 1 shows visualisations of the features learnt by the first layer of AlexNet trained with Adam and SGD on CIFAR-10. However, no further details are provided beyond those mentioned in the capture of the figure. This visualisation, in and on itself without further details and analysis, seems to me like a rather superficial result, disconnected from the rest of the paper which is focused on theoretical results.”
>
> **A4:** Thanks for pointing this out. In fact, the experiments in Figure 1 are performed to motivate us to study the separation between Adam and GD from a feature learning vs. noise memorization perspective. Particularly, we gain two main observations from Figure 1: (1) Adam and SGD tend to find different types of solutions; (2) the solution found by Adam is more “noisy” than that found by SGD. This actually motivates (a) the construction of the data model (composition of feature and noise); and (b) careful characterization of the learning dynamics of Adam and SGD. We have revised the introduction section to emphasize this.
>
> ---
>
> **Q5:** Comparison to Wilson et al. (2017) and Reddi et al. (2018).
>
> **A5:** In particular, [Wilson et al., 2017] considered a convex linear regression problem with artificially defined coordinates of the feature vectors; [Reddi et al., 2018] considered the sum of a set of one-dimensional linear functions. Each coordinate of the feature vector or the individual optimization objective is hard coded in these two prior works, while our data are randomly generated from some distribution. We have formally stated the data models considered in these prior works in the revision (Appendix E) along with a comparison with their data.
>
> ---
>
> **Q6:** The last item in the review of related work (Feature learning by neural networks) lacks diversity.
>
> **A6:** We have enriched that part with more related work on feature learning by different groups.
>
> ---
>
> **Q7:** long list of citations, typos
>
> **A7:** Thanks for your suggestions. We have revised the paper accordingly.
>
> ---
>
> **Q8:** Figure 1 lacks sufficient details:
>
> **A8:** We have added a subsection in Appendix A to provide the experiment details for Figure 1.

---

> ### Author Response · Authors · 2022-11-18
> **Follow up with Reviewer 1Tav**
>
> Dear reviewer, since the paper revision phase is ending soon, we would like to follow up with you to see if our response has addressed your concerns. We have also updated the paper as follows:
>
> - We have revised the introduction to clarify that Figure 1 is used to motivate the study of the generalization gap between Adam and GD from a “feature learning vs noise memorization” perspective. Besides, the experimental details of Figure 1 are provided in Appendix A.
> - We have added a new section (Appendix E) in the appendix to discuss the data models used in Wilson et al. (2017) and Reddi et al. (2018).
> - We have revised the related work section and enriched the part  “Feature learning by neural networks” with more related work on feature learning by different groups.
>
> We look forward to your feedback and would be happy to address any remaining concerns. Thank you!

---

### Official Review · Reviewer_zveC · 2022-10-26

**Confidence:** 3
**Correctness:** 3
**Technical Novelty And Significance:** 3
**Empirical Novelty And Significance:** Not applicable
**Recommendation:** 6

**Clarity, Quality, Novelty And Reproducibility:**

Although there are assumptions about the data, this paper considers the problem in a new setting (i.e. non-asymptotic and non-convex), so the novelty seems sufficient.

**Strength And Weaknesses:**

####  Strength
- This submission is well-written and the key motivation is well-discussed.
- It reveals the difference w.r.t. generalization and convergence speed between Adam and GD on a non-convex problem under the non-asymptotic setting, which was an open problem before.



#### Weaknesses
- The submission does not consider the de-bias term in Adam; hence, it should specify the initialize condition for $m_{j,r}$ and $v_{j,r}$ in Eq. (3.3) and Eq. (3.4), respectively.
- The noise vector's norm is much smaller than that of the feature part.
- The stochastic gradient complexity to find an $\epsilon$-approximate first-order stationary point on the general non-convex problem (non-stochastic) for GD is $\epsilon^{-2}$. Namely, $T = O(\epsilon^{-2})$ in Thm.4.1 for GD, which is right. Not the complexity's lower bound for the first-order optimation methods on the non-stochastic non-convex problem is $\epsilon^{-1.75}$. The complexity for Adam in Thm.4.1 is $\epsilon^{-1}$, which obviously violates the known lower bound. I know some assumptions exist on the data and model while the low bound is proved in the worst case, but the author should also discuss the part against a common guarantee.
- It is strange that the results for Adam have no connection to the moving average coefficients $\beta_1$ and $\beta_2$.
- As the author also claimed, Adam is only similar to signGD when  $\beta_1$ and $\beta_2$ are small. However, in practice and as pointed out by the author, we usually set  $\beta_1=0.9$ and $\beta_2=0.99$, which distinguish signGD and Adam. Moreover, small  $\beta$'s may make Adam diverge on some examples, but large $\beta$'s will not. Hence I quite doubt the proof routine that extends the results on signGD to Adam.
- For Lemma 5.2, $\eta s \sigma_p$ seems much smaller than $\eta$ due to $s = O(1)$ and $\sigma_p = O((poly log (n))^{-1})$, why to claim that $\eta s \sigma_p$ increases. much faster?

**Summary Of The Paper:**

On the specified data distribution, this submission shows that Adam and GD, starting from the same initialization, can converge to different solutions with significantly various generalization errors, even with proper regularization.

**Summary Of The Review:**

Although there are some limitations and issues in the theoretical part, this submission attempts to explain an interesting experimental observation from the theoretical perspective, so I have a positive score on it.

---

> ### Author Response · Authors · 2022-11-15
> **Response to Reviewer zveC**
>
> Thank you for your positive and helpful comments!
>
> ---
>
> **Q1:** "The submission does not consider the de-bias term in Adam"
>
> **A1:** Indeed, we do not consider bias correction in our paper. We set $m_{j,r}^{(0)}=\nabla L(W^{(0)})$ and $v_{j,r}^{(0)}=[\nabla L(W^{(0)})]^2$ as the gradient (and square of the gradient) at the initialization. We have clarified this in the revision.
>
> ---
>
> **Q2:** “The noise vector's norm is much smaller than that of the feature part”.
>
> **A2:**  In our setting. the noise vector’s norm is smaller than the feature vector norm by a factor of $\mathrm{polylog}(n)$ (the feature norm is $1$ and the noise norm is in the order of $\tilde\Theta(s\sigma_p^2)=1/\mathrm{polylog}(n))$. It serves as the purpose to demonstrate the difference between GD and Adam. Specifically, we show that even in the case where the noise vector has a smaller norm than the feature vector, Adam still fails to find a solution with good generalization. Therefore, we believe this is not a weakness of our setting as we demonstrate that Adam has issues even when learning such a relatively easy problem.
>
> ---
>
> **Q3:** Lower bound results of finding first-order stationary on the general non-convex problem.
>
> **A3:** Thanks for bringing it up. First of all, we would like to clarify that in our result, in order to achieve a convergence to a point $W_{\mathrm{Adam}}^*$ with $\\|\nabla L(W_{\mathrm{Adam}}^*)\\|\_F\le \epsilon$ for some $\epsilon > 0$, we in fact need to use a learning rate $\eta < \epsilon$. Therefore, reaching such a $W_{\mathrm{Adam}}^{*}$ requires $T = \Theta(1/(\epsilon \cdot \eta)) = \Theta(1 / \epsilon^2 )$ iterations. We have added a characterization of the iteration complexities of Adam and GD in the revision.
>
> Besides, you are also correct that we only consider a specific optimization problem (with certain data distribution, model structure, and loss function). Therefore, it is also possible that some other optimization algorithms can achieve a faster convergence than the minimax rate, which is established in the worst case.
>
> ---
>
> **Q4:** It is strange that the results for Adam have no connection to the moving average coefficients $\beta_1$ and $\beta_2$.
>
> **A4:** In our proof, we treat $\beta_1$ and $\beta_2$ as constants so their dependencies are hidden in the big-O notations. We have added this clarification in Section 3.
>
> ---
>
> **Q5:** “Adam is only similar to signGD when $\beta_1$ and $\beta_2$ are small”
>
> **A5:** In fact, instead of using small $\beta_1$ and $\beta_2$, we have mentioned in Section 5.1  that Adam can also be approximated by signGD if using a sufficiently small learning rate, which is the setting considered in our paper. In particular, we have rigorously proven this in Lemma C.2: considering constant $\beta_1$, and $\beta_2$,  we show that when the gradient coordinates are greater than $\Theta(\eta)$ or other related quantities depending on the learning rate, the Adam update of these coordinates will be similar to signGD. This demonstrates that the majority coordinates of the Adam update can be well approximated by signGD and our proof in Section 5 holds for Adam. To avoid confusion, we have revised Section 5.1 to emphasize that we consider a sufficiently small learning rate and highlight Lemma C.2 to demonstrate the close connection between Adam and signGD.
>
> ---
>
> **Q6:** “For Lemma 5.2, $\eta s \sigma_p$ seems much smaller than $\eta$ due to $ s=O(1)$ and $\sigma_p=O((\mathrm{polylog}(n))^{−1})$, why to claim that $\eta s \sigma_p$  increases. much faster?”
>
> **A6:** In fact $s$ is defined as the number of nonzero entries rather than its ratio (so $s$ is an integer). Then by Definition 3.1, we actually have $s=\Theta(d^{1/2}/n)=\omega(1)$ (note that $d=\omega(n^4)$) and $\sigma^2_p = \Theta\big(1/(s \mathrm{polylog}(n))\big)$. This implies that $\eta s\sigma_p = \Theta(\eta s^{1/2}/\mathrm{polylog}(n))=\omega(\eta)$, which is faster than the feature learning rate, i.e., $\Theta(\eta)$.

---

> ### Author Response · Authors · 2022-11-18
> **Follow up with Reviewer zveC**
>
> Dear reviewer, since the paper revision phase is ending soon, we would like to follow up with you to see if our response has addressed your concerns. We have also updated the paper as follows:
>
> - We have highlighted that the Adam parameters $\beta_1$ and $\beta_2$ are treated as constants in our paper.
> - We have also emphasized (in Section 5.1) that Adam can be well approximated by SignGD when using a small learning rate, which has been rigorously justified in Lemma C.2.
>
> We look forward to your feedback and would be happy to address any remaining concerns. Thank you!

---

### Decision · Program_Chairs · 2023-01-20

**Decision:**

Accept: poster

**Justification For Why Not Higher Score:**

Strong assumptions on the data and model limit the scope of the results.

**Justification For Why Not Lower Score:**

Interesting extension to our understanding of SGD vs Adam.

**Metareview: Summary, Strengths And Weaknesses:**

All reviewers agree that this paper makes important contributions towards extending existing theoretical analysis comparing SGD and Adam optimizers for the non convex setting and non asymptotic settings. However they also note several limitations.
- Analysis uses restricted data model.
- Analysis limited to 2 layer ReLU networks.
- Analysis does not consider the de-bias term in Adam.
- No results for mini batch setting.

Author's acknowledged the limitations in the response and provide some additional results during the discussion phase. Overall I think this is a good contribution and suggest acceptance.  I encourage authors to adopt all the reviewers suggestions in the final version. Importantly, some of the comparisons to existing works (Wilson et al. (2017) and Reddi et al. (2018)), which are currently in the appendix, should be moved to main paper. Perhaps author's can also include  comments on generalizing their results to data generated using standard distributions.


**Note From Pc:**

if the above contains the word "oral" or "spotlight" please see: "oral" presentation means -> notable-top-5% and "spotlight" means -> notable-top-25%. As stated in our emails, we are disassociating presentation type from AC recommendations